

# Ocean acidification dampens warming and contamination effects on the physiological stress response of a commercially important fish

Eduardo Sampaio[1#*], Ana R Lopes[1,2#], Sofia Francisco[1], Jose R Paula[1], Marta Pimentel[1], Ana L Maulvault[1,3,4], Tiago Repolho[1], Tiago F Grilo[1], Pedro Pousão-Ferreira[3], António Marques[3,4], Rui Rosa[1]

[1]MARE- Marine Environmental Sciences Centre & Laboratório Marítimo da Guia, Faculdade de Ciências, Universidade de Lisboa, Av. Nossa Senhora do Cabo 939, Cascais 2750-374, Portugal
[2]UCIBIO, REQUIMTE, Departamento de Química, Faculdade de Ciências e Tecnologia, Universidade Nova de Lisboa, Quinta da Torre, 2829-516 Caparica, Portugal
[3]Divisão de Aquacultura e Valorização (DivAV), Instituto Português do Mar e a Atmosfera (IPMA, I.P.), Av. Brasília, Lisboa 1449-006, Portugal
[4]Interdisciplinary Centre of Marine and Environmental Research (CIIMAR), University of Porto, Rua das Bragas, 289, 4050-123 Porto, Portugal

*Correspondence to*: Eduardo Sampaio (easampaio@fc.ul.pt)

# equally contributed

**Abstract.** Increases in carbon dioxide ($CO_2$) and other greenhouse gases emissions are leading to changes in ocean temperature and carbonate chemistry, the so-called ocean warming and acidification phenomena, respectively. Methylmercury (MeHg) is the most abundant form of mercury (Hg), well-known for its toxic effects on biota and environmental persistency. Despite more than likely co-occurrence in future oceans, the interactive effects of these stressors are largely unknown. Here we assessed organ-dependent Hg accumulation (gills, liver and muscle) within a warming ($\Delta T = 4$ ºC) and acidification ($\Delta pCO_2 = 1100$ µatm) context, and the respective phenotypic responses of molecular chaperone and antioxidant enzymatic machineries, in a commercially important fish (the meagre *Argyrosomus regius*). After 30 days of exposure, although no mortalities were observed in any treatments, Hg concentration was significantly enhanced under warming conditions, significantly more so in the liver. On the other hand, increased $CO_2$ decreased Hg accumulation and, despite negative effects prompted as a sole stressor, consistently elicited an antagonistic effect with temperature and contamination on oxidative stress (catalase, superoxide dismutase and glutathione-S-tranferase activities) and heat shock (Hsp70 levels) responses. We argue that the mechanistic interactions are grounded on simultaneous increase in excessive hydrogen ($H^+$) and reactive oxygen species (e.g. $O_2^-$) free radicals, and subsequent chemical reaction equilibrium balancing. Additional multi-stressor experiments are needed to understand such biochemical mechanism and further disentangle interactive (additive, synergistic or antagonistic) stressor effects on fish ecophysiology in the oceans of tomorrow.



# 1 Introduction

Atmospheric carbon dioxide ($CO_2$) concentrations have been increasing since the preindustrial era ($\simeq$400 CO2 µatm nowadays), and are expected to reach approximately 1000 CO2 µatm by the year 2100 (IPCC, 2014). Increased $CO_2$, along with other gases, triggers a continuous rise in mean ocean temperatures due to greenhouse effect, and predictions point to a

5 global temperature increase between 0.3 °C to 4.8 °C by the end of the century (IPCC, 2014). Simultaneously, atmospheric $CO_2$ dissolves in the ocean, altering seawater carbonate chemistry. Carbon dioxide uptake increases hydrogen ion ($H^+$) availability, leading to a concomitant decrease of 0.13-0.42 units in mean ocean pH by the year 2100, i.e. ocean acidification (IPCC, 2014). Due to naturally frequent variations in seawater physicochemical properties (e.g. upwelling events, significant carbon input from river basins), a more accentuated $CO_2$ input will occur in coastal areas, easily reaching $pCO_2$ values beyond

1500 µatm (Melzner, 2013). The combined occurrence of ocean warming and acidification imposes ecophysiological challenges to marine organisms, eliciting interactive negative effects on survival, growth and overall physiological fitness (Harvey et al., 2013; Kroeker et al., 2010; Pimentel et al., 2015).

In addition to global warming and ocean acidification, marine biota will also deal with an additional major stressor: contamination. One of the most concerning and persistent metal contaminants is mercury (Hg) and its ubiquitous

environmental compound, methylmercury (MeHg) (Korbas et al., 2011). Inorganic mercury is methylated into organic MeHg by bacteria present in the sediment of estuaries and coastal areas (Dijkstra et al., 2013), augmenting Hg bioavailability, bioaccumulation and biomagnification in marine organisms throughout the food web (Campbell et al., 2005; Evers et al., 2011). In teleost fish, MeHg accumulates preferentially in organ tissue, producing site-specific structural and functional damage (Gonzalez et al., 2005), and comprises around 90–95% of total mercury (HgT) (Burger et al., 2003; Gray et al., 2000).

Mercury accumulation can cause deleterious effects, such as physiological distress, i.e. activation of antioxidant and xenobiotic defense (Gonzalez et al., 2005; Mieiro et al., 2010), behavioural and organ functionality impairments (Berntssen et al., 2003; Sampaio et al., 2016), ultimately, mortality (Coccini et al., 2000).

Contaminant uptake and its impacts are potentially shaped by increased temperature or $CO_2$ and vice-versa (Noyes et al., 2009). Specifically, interactions between temperature and heavy metal contamination influence the physiological tolerance to

25 both stress factors (Sokolova and Lannig, 2008) while exacerbating biological responses (Dorts et al., 2014; Lapointe et al., 2011; Sappal et al., 2014). Consequently, MeHg accumulation is augmented and propagation throughout the food chain is strengthened, until metabolic thresholds are reached (Dijkstra et al., 2013). In parallel, severe acidification (pH < 7) increases metal availability (Wiener et al., 1990) and toxicity (Han et al., 2014). However, this effect may be offset by $CO_2$-linked metabolism decrease, leading to lower mercury accumulation via feeding (Sampaio et al., 2016; Schiedek et al., 2007). Under

30 environmental stressor exposure, a general deleterious biochemical pathway triggered is the formation of oxygen reactive species (ROS) in the organism's cells. Although there is some proof linking ROS production to hypercapnic scenarios (Pimentel et al., 2015), such is particularly true for increased temperature and mercury contamination (Berntssen et al., 2003;



Portner, 2002). Increasing ROS concentrations cause protein damage and lipid peroxidation, i.e. oxidative stress, cascading in augmented malondialdehyde content (MDA), one of the final products of lipid peroxidation (Lesser, 2006). As a physiological defense response, ROS production elicits antioxidant activity in the organism. Specifically, a battery of enzymes are activated to eliminate ROS and prevent MDA build-up: superoxide dismutase (SOD), which converts superoxide ($O_2^-$) into hydrogen

peroxide ($H_2O_2$); catalase (CAT) which converts $H_2O_2$ into water ($H_2O$) and oxygen ($O_2$); and glutathione S-transferase (GST), which is involved in the protection against xenobiotics and linked to antioxidant defense (Lesser, 2006; Wang et al., 2000). Moreover, tissue-specific heat shock proteins (Hsp70) production are also correlated with thermal stress, i.e. high temperatures (Repolho et al., 2014; Rosa et al., 2012, 2014a) and metal contamination (Rajeshkumar and Munuswamy, 2011; Williams et al., 1996). Heat shock proteins help repair, refold and eliminate damaged or denatured proteins, as well as protect

and control ROS formation (Sokolova et al., 2011). Given their wide scope, these constituents of the antioxidant enzymatic and protein chaperone machineries are widely used as biomarkers in ecotoxicology to assess fish physiological stress response (e.g. Anacleto et al., 2014; Fonseca et al., 2011; Rosa et al., 2014b).

Despite the inevitability of marine organisms having to cope with simultaneous effects of ocean warming, acidification and persistent contamination (MeHg), no studies have focused on how the interactive effects between these three stressors will

challenge teleost fish ecophysiology. Due to its coastal distribution, the meagre *(Argyrosomus regius)* is particularly susceptible to MeHg accumulation, especially when they migrate towards the estuaries to spawn (Durrieu et al., 2005). Understanding how this commercially important species will deal with the predicted climate change scenarios may provide valuable information on future stock population conditions and potential impacts on coastal food-webs. Within this context, here we performed a 30-day acclimation experiment to investigate organ-dependent Hg accumulation (gills, liver and muscle)

under a warming ($\Delta T = 4$ ºC) and acidification ($\Delta CO_2 = 1100$ µatm) context, as well as the respective phenotypic responses of molecular chaperone (Hsp70) and antioxidant enzymatic (SOD, CAT and GST) machineries, in commercially important fish (*A. regius*). The direct consequences at organism (survival rates and condition index) and cellular (lipid peroxidation, MDA) levels were also evaluated.

## 2 Material and Methods

### 2.1 Experimental setup and incubation

Juvenile *Argyrosomus regius* (n $\simeq$ 100; Fig. 5) (mean ± SD; total weight: 4.26 ± 2.8 g; total length: 6.30 ± 1.2 cm) from EPPO - IPMA (Estação Piloto de Piscicultura de Olhão - Instituto Português do Mar e da Atmosfera, Portugal) were transported to the facilities of Laboratório Marítimo da Guia (LMG, MARE, Faculdade de Ciências, Universidade de Lisboa) in August 2014. Fish were randomly placed in twenty-four 50 l tanks (n = 3-4 per tank) with individual recirculating aquaculture systems

(RAS) equipped with glass wool (physical filtration), bio-balls (Fernando Ribeiro Lda) and protein skimmers (biological



filtration, ReefSkimPro 850, TMC Iberia), as well as additional UV disinfection (Vecton 120, TMC Iberia) to maintain superior water quality. Natural seawater was pumped directly from the ocean into an 8 m3 storage tank, and subsequently filtered (0.35 µm filters, Fernando Ribeiro Lda) and UV-sterilized (Vecton600, TMC Iberia), before pumping into mixing (n = 24) and respective experimental (n = 24, 50 l) tanks/RAS. Each RAS worked as a semi-closed system, with constant low flow external

water input. Ammonia, nitrate and nitrite were regularly monitored and kept within recommended levels (Aquamerk). Salinity was kept at $35.0 \pm 1.0$ gl-1 and the photoperiod was fixed to 12 h light: 12 h dark. Temperature, salinity and pH (multiparametric probe, Multi3420 SET G, WTW) were daily measured directly in the holding tanks.

As per experimental conditions, temperature in the tanks was down-regulated using chillers ($\pm 0.1$ ℃, Frimar, Fernando Ribeiro Lda), and up-regulated by submerged 200 W heaters (V2Therm, TMCIberia). Seawater carbonate chemistry was altered

through $CO_2$-enriched air input, with pH (8.0 and 7.5) used as proxy measurement. As pH controller, we used a Profilux system ($\pm 0.1$, Profilux 3.1N, GHL) connected to each tank by individual pH probes. Within each RAS, pH was down-regulated by injection of the certified $CO_2$-enriched air (Air Liquide), and up-regulated by injection of atmospheric air. Seawater carbonate system speciation (Table S1) was calculated once every week from total alkalinity according to Sarazin et al. (1999). Total dissolved inorganic carbon (CT), $pCO_2$ and aragonite saturation were calculated using CO2SYS software (Lewis and

Wallace, 1998), with dissociation constants from (Mehrbach et al., 1973) as refitted by Dickson and Millero (1987). The non-contaminated and contaminated fish were fed similar diets, differing only on MeHg content. Contaminated diet was fortified with MeHg (inserted in the form of MeHg (II) chloride, $CH_3ClHg$, 99.8 %, Sigma-Aldrich, solubilized previously in ethanol). The pellet given to the fish in the contaminated treatment had approximately $8.02 \pm 0.01$ mg kg$^{-1}$ dw of MeHg and $8.28 \pm 0.01$ mg kg$^{-1}$ dw of HgT. An ecologically relevant concentration was chosen, indicated by previous studies on contaminated coastal

areas (Nunes et al., 2008). MeHg exposure occurred via feed intake. Fish were fed two to three times a day and food amount provided per day was approximately 1% of animal weight.

After 15 days of lab acclimation (control conditions: 19 ℃, $CO_2 \simeq 400$ µatm), fish were kept during 30 days under crossed-treatments of ocean warming ($\Delta T = 4$ °C), acidification ($\Delta pH = 0.5$ units, i.e. $\Delta pCO_2 = 1100$ µatm) and MeHg contamination (contaminated and non-contaminated) in a full-factorial design, simulating predicted "business-as-usual" scenarios for the year

2100 (IPCC, 2014; Melzner et al., 2013; Schiedek et al., 2007).The experimental setup mimicked the design elaborated by Cornwall and Hurd (Fig. 3d, 2015). More specifically, the setup was divided in eight treatments (n = 3 tanks per treatment): i) 19 °C, 400 $pCO_2$ µatm (control conditions) and non-contaminated feed (MeHg: 0.06 mg kg$^{-1}$; HgT: 0.07 mg kg$^{-1}$), ii) 19 °C, 400 $pCO_2$ µatm and contaminated feed (MeHg: 8.02 mg kg$^{-1}$;HgT: 8.28 mg kg$^{-1}$), iii) 19 °C, 1500 $pCO_2$ µatm (control temperature and hypercapnic scenario) and non-contaminated feed (MeHg: 0.06 mg kg$^{-1}$; HgT: 0.07 mg kg$^{-1}$ ), iv) 19 °C, 400

$pCO_2$ µatm and contaminated feed (MeHg: 8.02 mg kg$^{-1}$; HgT: 8.28 mg kg$^{-1}$); v) 23 °C, 400 $pCO_2$ µatm (warming and normocapnic scenario) and non-contaminated feed (MeHg: 0.06 mg kg$^{-1}$; HgT: 0.07 mg kg$^{-1}$); vi) 23 °C, 400 $pCO_2$ µatm and contaminated feed (MeHg: 8.02 mg kg$^{-1}$; HgT: 8.28 mg kg$^{-1}$); vii) 23 °C, 1500 $pCO_2$ µatm (warming and hypercapnic





scenario) and non-contaminated feed (MeHg: 0.06 mg kg⁻¹; HgT: 0.07 mg kg⁻¹); and viii) 23 °C, 1500 pCO$_2$ µatm and contaminated feed (MeHg: 8.02 mg kg⁻¹; HgT: 8.28 mg kg⁻¹).

Survival rates were monitored throughout the experiment, and after 30 days fish were measured (total length) and weighed. Health status was assessed through the widely used Fulton's condition factor K (n = 6-8 per treatment), described by the following formula: K = 100 x (Weight / Length3). Individuals were anesthetized with MS-222 and euthanized by swift spinal cord severing. Sample tissues of three organs (muscle, liver and gills) were harvested for further analysis.

## 2.2 Total mercury and Methylmercury accumulation

Methylmercury extraction from samples (fish and different feeds, n = 3-6) was performed as described by Scerbo and Barghigiani (1998), i.e. freeze-dried samples (≃200 mg) were hydrolyzed in 10 ml of hydrobromic acid (47 % w/w, Merck), following addition of 35 ml toluene (99.8 % w/w, Merck) to allow MeHg extraction and removal with 6 ml cysteine solution (1 % L-cysteinium chloride in 12.5 % anhydrous sodium sulfate and 0.775 % sodium acetate, Merck). Afterwards, HgT and MeHg were determined in all samples (10-15 mg for solids or 100-200 µl for liquids) by atomic absorption spectrometry (AAS), following EPA (2007) by means of an automatic Hg analyser (AMA 254, LECO, USA) with a detection threshold of 0.005 mg kg-1, wet weight (ww). Mercury concentrations were calculated through linear calibration (using > 5 standard concentrations), with a Hg(II) nitrate standard solution (1000 mg l-1, Merck) dissolved in nitric acid (0.5 mol l−1, Merck). Accuracy was checked by also analyzing certified reference material DORM-4, and framing results obtained within the certified range of values. A minimum of three measurements were performed per sample. Blanks were always tested in the same conditions as the samples and measurements were taken in triplicate. All laboratory ware was previously cleaned using with nitric acid (20 % v/v) for 24h and ultrapure water, in that order. All standards and reagents were of analytical (pro analysis) or superior grade.

## 2.3 Enzymatic assays

### 2.3.1 Preparation of tissue extracts

Muscle, liver and gills samples (n = 4-6 per tank) were homogenized (Ultra-Turrax, Staufen, Germany) in accordance to body mass of each sample in homogenization buffer, 300 mg tissue per 1 ml phosphate buffered saline solution (PBS, pH 7.4): 0.14 M NaCl, 2.7 mMKCl, 8.1 mM Na$_2$HP0$_4$, 1.47 mM KH$_2$P0$_4$. Posteriorly, homogenates were centrifuged (20 min at 14000 rpm at 4 °C) and antioxidant enzyme activities, as well as lipid peroxidation and heat shock response concentrations were quantified in the supernatant fraction. All enzyme assays were tested with commercial enzymes obtained from Sigma-Aldrich (St. Louis, USA), and each sample was run in triplicate (technical replicates). The enzyme results were normalized with total protein content following the Bradford method (Bradford, 1976).



### 2.3.2 Lipid peroxides assay (malondialdehyde concentration)

As an end-product of oxidative stress, malondialdehyde (MDA) concentration was used as a proxy to assess extent of lipid peroxidation. We used the thiobarbituric acid reactive substances (TBARS) protocol described by Uchiyama and Mihara, (1978). A total of 10µl of each sample were added to 45 µl of 50 mM monobasic sodium phosphate buffer, followed by addition

of 12.5 µl of sodium dodecyl sulfate (8.1%), 93.5 µl of trichloroacetic acid (20%, pH = 3.5) and 93.5 µl of thiobarbituric acid (1%) to each microtube. Then, 50.5 µl of ultrapure water were added to this mixture and placed in a vortex for 30 s. A needle was used to puncture the lids and microtubes were incubated in boiling water (10 min) followed by ice cooling. Subsequently, 62.5 µl of ultrapure water and 312.5 µl of n-butanol pyridine (15:1, v/v) (Sigma-Aldrich, Hamburg, Germany) were added and microtubes centrifuged (5000 x g; 5 min.). 150 µl of the supernatant's reaction were introduced into a 96-well microplate in

duplicate and absorbance was read at 530 nm. Lipid peroxides (i.e., MDA concentration) were determined using malondialdehyde (dimethylacetal) (MDA) (Merck, Switzerland) standards in an eight-point calibration curve (0–0.3 µM TBARS). Results were expressed in relation to the sample total protein (nmol mg-1 total protein).

### 2.3.3 Catalase (CAT) activity

Catalase activity was assessed through an adaptation of the method described by Johansson and Borg (1988). In this assay, 20

µl of each sample, 100 µl of 100 mM potassium phosphate and 30 µl of methanol were added to a 96-well microplate, which was promptly shaken and incubated for 20 minutes. Afterwards, 30 µl of potassium hydroxide (10 M KOH) and 30 µl of purpald (34.2 mM in 0.5 M HCl) were added to each well, and the plate shaken and incubated for another 10 minutes. Subsequently, 10 µl of potassium per iodate (65.2 mM in 0.5 M KOH) was added to each well and a final incubation was performed for 5 minutes. Using a microplate reader (Asys UVM 340, Biochrom, USA), enzymatic activity was determined

spectrophotometrically at 540 nm. Formaldehyde concentration of the samples was calculated based on a calibration curve (from 0 to 75 µM formaldehyde), followed by the calculation of CAT activity for each sample, where one unit of CAT is defined as the amount that will cause the formation of 1.0 nmol of formaldehyde per minute at 25 ºC. The results are expressed in relation to total protein content (nmol min mg-2 protein).

### 2.3.4 Superoxide Dismutase (SOD) activity

SOD activity was determined following the nitro blue tetrazolium (NBT) method adapted from Sun et al. (1988).Superoxide radicals ($O_2^-$) are generated by xanthine oxidation, and simultaneous reduction of NBT to formazan. SOD competes with NBT for the dismutation of $O_2^-$ into hydrogen peroxide ($H_2O_2$) and molecular oxygen, and this is used to determine enzyme activity. Briefly, the assay was performed using a 96-well microplate (Nunc-Roskilde), adding to each well 200 µl of 50 mM phosphate buffer (pH 8.0) (Sigma-Aldrich), 10 µl of 3 mM EDTA (Riedel-de Haën, Seelze, Germany), 10 µl of 3 mM xanthine (Sigma-

Aldrich), 10 µl of 0.75 mM NBT (Sigma-Aldrich) and 10 µl of SOD standard or sample. Reaction began by adding 10µl of



100 mU xanthine-oxidase (XOD, Sigma-Aldrich) and absorbance (560 nm) was recorded every 5 minutes for 25 minutes, using a plate reader (Asys UVM 340, Biochrom, USA). SOD from bovine erythrocytes (Sigma-Aldrich) was used as standard and positive control and a negative control included all components except SOD or sample. The latter yielded a maximum threshold in absorbance, which allowed the assessment of inhibition percentage per minute, which is caused by SOD activity.

Thus, SOD activity percentage was expressed in % inhibition mg⁻¹ of total protein.

### 2.3.5 Gluthationse S-Tranferase (GST) activity

GST total activity was determined according to the procedure described by Habig et al., (1974) and optimized for 96-well microplate (Sigma Technical Bulletin, GST Assay Kit CS0410). 1-Chloro-2,4-dinitrobenzene (CDNB) is used as substrate and, upon conjugation of the thiol group of glutathione to the CDNB, absorbance is increased and enzymatic activity can be

determined spectrophotometrically. The assay included 200 mM L-glutathione (reduced), 100 mM 1-chloro-2,4-dinitrobenzene (CDNB) solution and Dulbecco's PBS. Equine liver GST (Sigma-Aldrich) was used as positive control to validate the assay. 180 µl of substrate solution were added to 20 µl sample in each well of a 96-well microplate (Nunc-Roskilde) and 340 nm absorbance was registered every minute during 6 minutes, through a plate reader (Asys UVM 340, Biochrom, USA). Finally, GST activity was calculated using a molar extinction coefficient for CDNB of 5.3 εmM (Sigma Technical

Bulletin, CS0410), as follows: GST activity = (ΔA340min / 0.0053) x (Total volume / Sample volume) x dilution factor. Results were expressed in relation to total protein of the sample (nmol min mg-2 total protein).

### 2.3.6 Heat shock proteins

Heat shock protein (Hsp70/Hsc70) content was assessed by Enzyme-Linked Immunoabsorbent Assay (ELISA) protocol adapted from Njemini et al. (2005).10 µl of the supernatant was diluted in 990µl of PBS and 50 µl of that sample were added

to a 96-well microplates (Microloan 600, Greiner) and allowed to incubate overnight at 4 °C. On the next day, microplates were washed (three times) in 0.05 % PBS-Tween-20. 100µl of blocking solution (1 % bovine serum albumin (BSA) Sigma-Aldrich) were added to each well and left to incubate for 2 h at room temperature. After washing the 96-well plates, we introduced 50 µl of 5 µg ml-1 primary antibody (anti-Hsp70/Hsc70, Acris, San Diego, CA, USA),and again left incubating overnight at 4 °C. According to manufacturer details, the primary antibody Hsp70/Hsc70 (AM12032PU-N) possesses broad

range reactivity, e.g. in varied fish species, making it suitable for our analysis. On the next day, the non-linked antibody was removed by washing the microplates, and 50 µl of 1 µg ml⁻¹ of the secondary antibody, antimouse IgC, Fab specific, alkaline phosphatase conjugate (Sigma-Aldrich) were added and incubated for 2 h at room temperature. After three additional washes, 100 µl of substrate (SIGMA FASTTM p-Nitrophenyl Phosphate Tablets, Sigma-Aldrich) was added to each well and incubated 10-30 minutes at room temperature. Stop solution (50 µl; 3 N NaOH) was added in each well, and absorbance was read at 405

30 nm in a 96-well microplate reader (Asys UVM 340, Biochrom, USA). The amount of Hsp70/Hsc70 present in the samples





was calculated from an absorbance/concentration calibration curve based on serial dilutions of purified Hsp70 active protein (Acris), ranging from 0 to 2000 ng ml⁻¹. Results were expressed in relation to the sample total protein (ng mg⁻¹ total protein).

## 2.4 Statistics

All statistical analysis were performed on R Studio (R Development Core Team, 2016). We used Generalized Linear Models (GLM) analysis to infer significant differences between sampled groups (please see R script provided in the Supplemental Data for a step-by-step protocol). Mix models, e.g. tank as random factor, were ruled unnecessary as previous analysis (using 'lme4' and 'nlme' packages) showed no significant differences between tanks, within each group treatment, for all variables used. Best model selection fit our data was found using the Akaike Information Criterion (AIC), a widespread indicator that balances model complexity with model quality of fitness (Quinn and Keough, 2002). Thus, models were simplified and factors that did not influence data variation were removed. Data was fitted using gaussian family models, and model residuals were checked for homogeneity of variances, independence and leverage were used to perform model validation. When assumptions were not met, we turned to gamma family models to fit our data, and model validation was assessed following the same procedure. Temperature (T, 2 levels: 19 ℃, 23 ℃) $CO_2$ ($CO_2$, 2 levels: 400 µatm, 1500 µatm), MeHg exposure (MeHg, 2 levels: Non-contaminated, 0.06 mg kg⁻¹; Contaminated, 8.02 mg kg⁻¹) and organ tissue sampled (Tissue, 3 levels: Muscle, Gills, Liver) were generally used as explanatory variables or factors, according to each specific dependent variable.

## 3 Results

After 30 days of exposure, no mortalities were registered in any treatment. Fulton condition (K) did not show any significant differences between treatments (MeHg, $p > 0.05$, GLM analysis in Table 1). Significant differences were found in total mercury concentrations between contaminated and non-contaminated scenarios (GLM analysis, $t = 9.079$, $p < 0.001$, see Supplemental Data) and also between tissues analyzed (ANOVA F test, $F = 14.015$, $p < 0.001$, see Supplemental Data). Hg concentration was lower in the muscle (Muscle & Liver/ Muscle & Gills, $p < 0.001$, GLM Analysis) and higher in the liver (Liver & Muscle, $p < 0.001$, GLM Analysis in Table 1, Figure 1a). Within each tissue, temperature and $CO_2$ interacted significantly (T x $CO_2$, $p < 0.001$ for all tissues, GLM analysis in Table 2, Figure 1b-d) affecting MeHg accumulation. In other words, temperature increased Hg accumulation and such effect was counter-balanced by elevated $CO_2$.

A significant antagonistic effect was detected between increasing temperature and MeHg contamination on MDA build-up (T x MeHg, $p < 0.05$, GLM analysis in Table 3, Figure 2). Isolated stressors increased MDA production, however this effect was annulled when both stressors were present. Regarding the antioxidant enzyme machinery, CAT activity was positively affected by MeHg contamination (MeHg, $p < 0.05$, GLM analysis in Table 4, Figure 3a). On the other hand, elevated $CO_2$ increased SOD activity as a single stressor; yet, when combined with warming (T x $CO_2$, $p < 0.001$, GLM analysis in Table 4, Figure




3b), the effect was reversed. GST activity was modelled by two interactions between $CO_2$ and temperature (T x $CO_2$, $p < 0.01$, GLM analysis in Table 4, Figure 4a), and between $CO_2$ and MeHg contamination ($CO_2$ x MeHg, $p < 0.01$, GLM analysis in Table 4, Figure 4b). Not reporting strong effects as a sole stressor, increased $CO_2$ inhibited GST activity when combined with warming (Figure 4a) or MeHg contamination (Figure 4b).

5 Concerning heat shock response, Hsp70 production varied between the analyzed organs (ANOVA F test, F = 11.732, $p < 0.001$, see Supplemental Data), reporting higher concentrations in the liver and lower in the gills (liver > muscle > gills; see GLM analysis in Table 5 for p values, Figure 5a). Within the gills, Hsp70 concentration was positively affected by MeHg contamination (Gills, $p < 0.05$, GLM analysis in Table 5, Figure 5b). On the other hand, in the muscle, temperature and $CO_2$ modulated Hsp70 production (T x $CO_2$, $p < 0.001$, GLM analysis in Table 5, Figure 5c). While isolated, elevated $CO_2$ increased

10 Hsp70 production, but under simultaneous warming, heat shock response was significantly decreased. Concomitantly, temperature-driven Hsp70 increase was also dampened by hypercapnia. Similarly, in the liver, Hsp70 concentration MeHg contamination increased Hsp70 production, but this effect was countered by increased $CO_2$ ($CO_2$ x MeHg, $p < 0.01$, GLM analysis in Table 5, Figure 5b).

## 4 Discussion

### 4.1 Non-lethal preferential accumulation

The present study showed that Hg contamination, ocean warming and acidification interactively affected fish physiological condition at non-lethal levels, i.e. zero mortality was registered. However, our AIC-chosen best model indicated that mercury may diminish organism Fulton condition, which is in agreement with previous results obtained in river fish populations (Pyle et al., 2005). The fact that the meagre (A. regius) is a very resilient species and easily adapts to environmental alterations

20 (Monfort, 2010) may explain the absence of deleterious effects after 30 days of exposure at an organism level.

Affinity for metal accumulation varied between fish tissues with increasing Hg accumulation as follows: muscle < gills < liver. These results are supported by previous reports on mercury tissue preferential accumulation. The muscle is an organ tissue generally characterized for its low metal affinity (Jezierska and Witeska, 2006) compared to, e.g. the liver, where metals accumulate at higher levels, due to its key role in metal accumulation and detoxification (Gbem et al., 2001; Wagner and

25 Boman, 2003). Furthermore, as a result of increased blood supply, gills are organs likewise known to possess higher Hg affinity than the muscle (Jezierska and Witeska, 2006; Vergilio et al., 2012).

### 4.2 Environmental influence on mercury accumulation

Mercury accumulation in fish is known to depend on the water physicochemical properties (e.g. temperature, pH, alkalinity) (Harris and Bodaly, 1998; Ponce and Bloom, 1991; Wren et al., 1991). Indeed, we also showed a consistent increase in Hg



accumulation under the warming scenario. However, when both temperature and $CO_2$ stressors were present, Hg accumulation was notoriously decreased. Temperature increases Hg bioaccumulation in fish due to enhanced metabolism and consequent higher intake of MeHg-contaminated prey (Dijkstra et al., 2013; MacLeod and Pessah, 1973). Despite previous evidence that lowered pH (< 7.0 units) increases Hg accumulation in freshwater fish (Haines et al., 1992; Ponce and Bloom, 1991), the

5 current findings do not reflect this pattern, arguably due to the magnitude of pH decrease (here we used pH 7.5). Instead, our results support other reports demonstrating that fish exposed to hypercapnia may display metabolic decrease due to prioritization of $CO_2$-excretory physiological processes (Perry et al., 1988; Sampaio et al., 2016). Thus, taking also into account that occurrence of both stressors lowers physiological (and consequently metabolic) thresholds (Harvey et al., 2013; Rosa et al., 2013), it is likely that a certain degree of metabolic arrest played a key role on HgT concentration decrease.

## 4.3 Oxidative stress under a multi-stressor environment

Exposure to MeHg contamination, ocean warming and acidification potentiated significant changes in meagre physiology. As expected, lipid peroxidation and consequent MDA build-up was higher under MeHg contamination (Berntssen et al., 2003; Vieira et al., 2009).The fact that contamination and warming per se elicited only small MDA build-up, is likely due to the fact

that A. regius is a highly resilient estuarine species, i.e. great tolerance to environmental stressors (Monfort, 2010). To cope with oxidative stress, A. regius displayed enhanced CAT, SOD and GST activities under contaminated and warming scenarios. While it is worth mentioning that increased $CO_2$ played a minor role in CAT activity (non-significant, $p = 0.116$), regarding the other enzymes, hypercapnia as a sole stressor significantly augmented antioxidant activity. However when combined with other stressors, elevated $CO_2$ antagonized the co-ocurring stressor's effect (i.e., contamination and/or warming). We argue that

such can be explained by the dramatic increase of H+ ion concentrations in the blood and cellular surroundings stemming from increased $CO_2$ (Michaelidis et al., 2007). By itself, the presence of excessive $H^+$ ions activates free radical neutralizing defenses (Tiedke et al., 2013), however the production of $O_2^-$ and further complementary ROS free radicals (e.g., $OH^-$) by other stressors may result in facilitated $H_2O$ and $H_2O_2$ formation, due to chemical reactions balancing equilibrium (e.g. $H^+ + OH^- \rightleftharpoons H2O$), thus eliminating free radicals and decreasing activity of antioxidant enzymes to basal standards.

## 4.4 Protein chaperone functioning under a multi-stressor environment

Hsp70 response was tissue-dependent, showing a pattern similar to HgT tissue preferential accumulation (see first section). Higher liver expression is not unexpected given the fact that this organ plays a key role in metal accumulation and detoxification (Gbem et al., 2001; Wagner and Boman, 2003). More importantly, as observed in antioxidant stress enzymatic

machinery, hypercapnia revealed the same antagonistic relationship with other stressor's effects: increased $CO_2$ down-



regulated heat shock response in the livers of contaminated fish and in the muscle of fish under warming. As such, this study confirms that Hsp70 expression is closely correlated with other forms of antioxidant response, such as CAT, SOD and GST (Iwama et al., 1998; Rosa et al., 2012, 2014a). Moreover, given that Hsp70 production can be stimulated by extreme ionic (e.g. $H^+$) concentrations (Feder and Hofmann, 1999), we speculate that the mechanism by which hypercapnia modulates heat shock

response expression is likely similar to oxidative stress enzymatic machinery modulation. Enhanced $CO_2$ leads to increased $H^+$ concentration triggering physiological stress responses, while the facilitated conversion of free ions and radicals ($H^+$ and O-associated molecules) into $H_2O$ and $H_2O_2$ leads to reduced stress input by warming, contamination (and hypercapnia itself).

## 5 Conclusions

In this study, we verified that sub-lethal MeHg contamination is organ selective (accumulating to higher levels in the liver)

and found that future abiotic conditions modulate its accumulation throughout the organism. In general, warming conditions enhanced MeHg accumulation but $CO_2$-linked metabolic reductions countered this effect. Moreover, despite negative effects prompted as a sole stressor, acidification consistently elicited antagonistic responses to temperature and contamination effects on oxidative stress and heat shock responses. Thus, we argue that the mechanistic interactions found are underpinned by the coinciding increase of excessive hydrogen ($H^+$) and radical reactive oxygen species (e.g. $O_2^-$, $OH^-$), which subsequently nullify

each other due to the spontaneous equilibrium of chemical reactions (e.g. $H^+ + OH^- \rightleftharpoons H2O$).

In the future, it is important to deepen our understanding on this mechanism and evaluate if this antagonistic relationship is conservative throughout other less-resilient species (e.g. non-estuarine ones). Further knowledge on climate change and contamination impacts on fish ecophysiology (and biochemical stress-coping mechanisms) will help towards better comprehend the future health condition of coastal fish populations and consequently forecast socio-ecological consequences

in the oceans of tomorrow.

## 6 Code availability

R code used in the analysis is available as Supplemental material.

## 7 Data availability

The full dataset is made available as Supplemental material.



## 8 Competing interests

The authors declare no conflict of interest.

## 9 Author contributions

ES, ARL, SF, AM, PP and RR designed the study. JRP, MP, TR and TFG assisted during the experiment and sampling. AL and SF quantified HgT accumulation. ARL, SF, MP and JRP quantified the enzymes. ES, TR, TFG and RR performed the statistical analysis. ES and ARL wrote the paper, for which all authors contributed with discussion and earlier drafts.

## Acknowledgments

We thank Kenneth Storey for the helpful discussion and IPMA-Olhão for providing juvenile meagre specimens for the trials. This work was supported by Fundação para a Ciência e Tecnologia (FCT): PhD (ARL, SFRH/BD/97070/2013; JRP, SFRH/BD/111153/2015; ALM, SFRH/BD/103569/2014) and post-doctoral (TFG, SFRH/BPD/98590/2013; TR, SFRH/BPD/98590/2013) scholarships, as well as RR and AM in the framework of the IF 2013 and IF2014 programs.

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

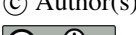




**Figure 1. Total mercury (HgT) accumulation (mean ± SE) in A. regius: a) Differences among tissues (muscle, gills and liver); and shaped by interactions between temperature (19 and 23 ºC) and CO2 (400 and 1500 µatm) within b) muscle, c) gills and d) liver, respectively. Graphs were plotted according to significant factors yielded by GLM analysis described in Table 1 and 2, respectively.**





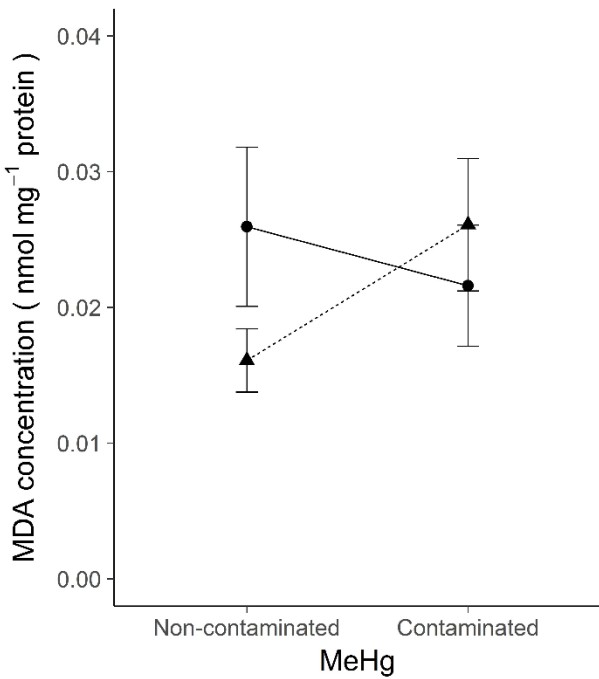

5  **Figure 2. Malondialdehyde (MDA) build-up concentrations (mean ± SE) in A. regius driven by an interaction between MeHg contamination (Non-contaminated and contaminated) and temperature (19 and 23 ºC). Graphs were plotted according to significant factors yielded by GLM analysis described in Table 3 and 4, respectively.**





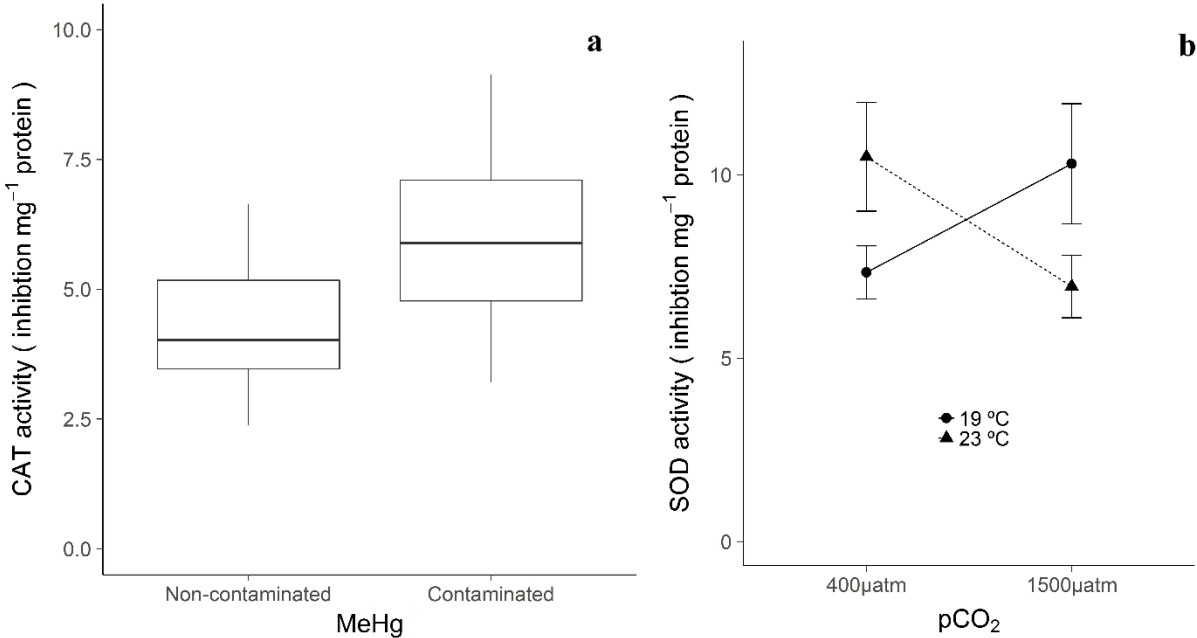

5   **Figure 3. a) Catalase (CAT) enzyme activities (mean ± SE) driven by MeHg contamination (Non-contaminated and Contaminated). b) Superoxide dismutase (SOD) activities (mean ± SE) in A. regius driven by an interaction temperature (19 and 23 ºC) and CO2 (400 and 1500 µatm). Graphs were plotted according to significant factors yielded by GLM analysis described in Table 4.**




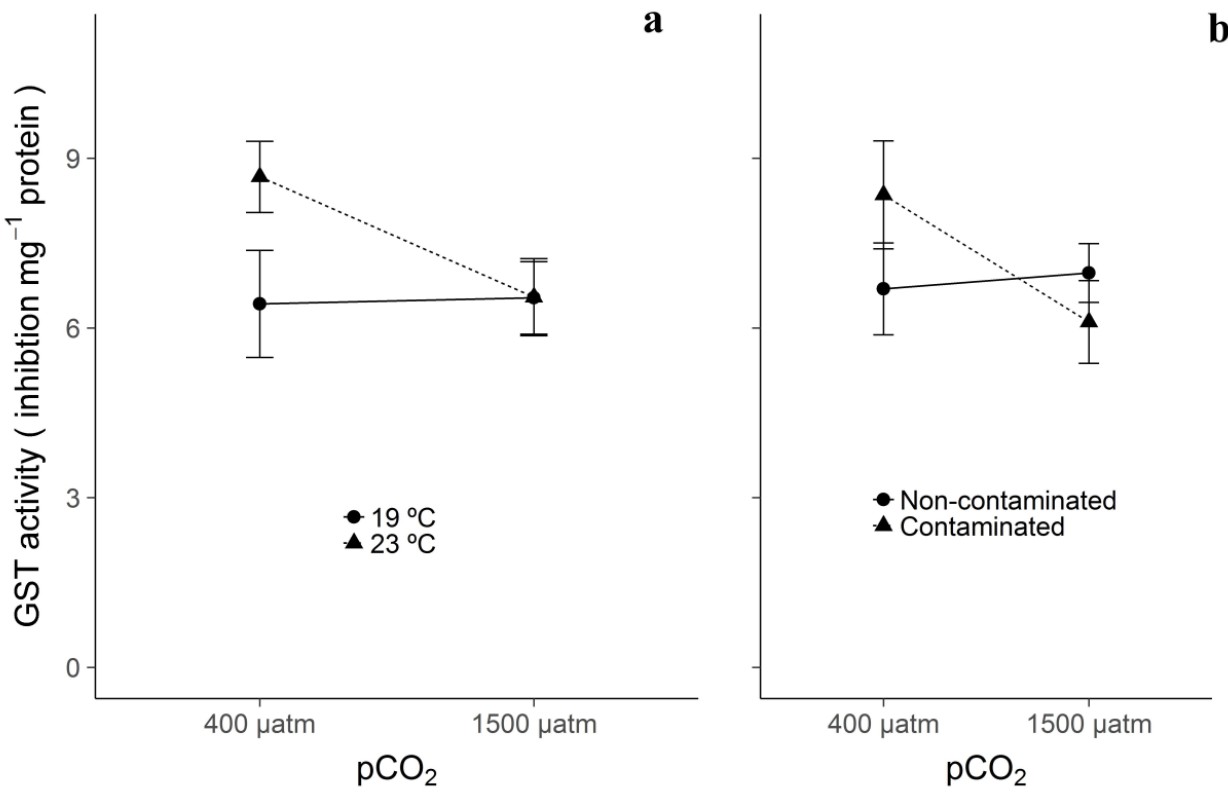

**Figure 4. Glutathione S-Transferase (GST) activities (mean ± SE) in A. regius driven by: a) an interaction between temperature (19 and 23 ºC) and CO2 (400 and 1500 µatm); and b) an interaction between MeHg contamination (Non-contaminated and contaminated) and CO2 (400 and 1500 µatm). Graphs were plotted according to significant factors yielded by GLM analysis (triple interaction) described in Table 4.**






**Figure 5. Heat shock protein70 (Hsp70) concentrations (mean ± SE) in A. regius: a) tissues; d) in the gills liver shaped by MeHg contamination (Non-contaminated and Contaminated) and CO2 (400 and 1500 µatm); in the c) muscle shaped by an interaction between temperature (19 and 23 ºC) and CO2 (400 and 1500 µatm); and in the d) liver shaped by an interaction between MeHg contamination (Non-contaminated and Contaminated) and CO2 (400 and 1500 µatm). Graphs were plotted according to significant factors yielded by GLM analysis described in Table 5.**





**Table 1. GLM analysis of A. regius Fulton's K and total mercury (HgT) concentrarion in tissues (3 levels within contaminated treatments: liver, muscle and gills) exposed to MeHg contamination (2 levels: non-contaminated and contaminated) for 30 days. Model formula on top, family and respective model AIC in the bottom. Est – Estimates; Std Error – Standard Error. Bold values indicate p < 0.05. For more details please see the R script in Supplemental Data.**

*GLM: Fulton's K in function of MeHg*

|  | Est | Std Error | t value | p value |
|---|---|---|---|---|
| (Intercept) | 1.602 | 0.041 | 39.09 | **< 0.001** |
| MeHg | -0.072 | 0.057 | 0.057 | 0.213 |

| Family = Gaussian | AIC = -8.6 |
|---|---|

*GLM: HgT in function of MeHg * Tissues*

|  | Est | Std Error | t value | p value |
|---|---|---|---|---|
| (Intercept) | 1.576 | 0.082 | 19.11 | **< 0.001** |
| Muscle & Gills | 0.470 | 0.128 | 3.665 | **< 0.001** |
| Muscle & Liver | 0.660 | 0.130 | 5.063 | **< 0.001** |
| Gills & Liver | 0.191 | 0.141 | 1.355 | 0.181 |

| Family = Gamma | AIC = 270.3 |
|---|---|





*GLM: Liver HgT in function of T * CO₂*  *GLM: Muscle HgT in function of T * CO₂*  *GLM: Gills HgT in function of T * CO₂*

| | Est | Std Error | t value | p value | Est | Std Error | t value | p value | Est | Std Error | t value | p value |
|---|---|---|---|---|---|---|---|---|---|---|---|---|
| (Intercept) | 2.287 | 0.151 | 15.18 | **< 0.001** | 1.520 | 0.063 | 24.31 | **< 0.001** | 2.059 | 0.125 | 16.74 | **< 0.001** |
| T | 0.794 | 0.261 | -3.043 | **0.010** | 0.579 | 0.088 | 6.551 | **< 0.001** | -0.917 | 0.191 | -4.792 | **< 0.001** |
| $CO_2$ | -0.295 | 0.195 | -1.514 | 0.156 | -0.201 | 0.088 | -2.268 | **0.035** | -0.157 | 0.162 | -0.970 | 0.350 |
| T * $CO_2$ | 1.468 | 0.326 | 4.508 | **< 0.001** | 0.627 | 0.125 | 5.017 | **< 0.001** | 1.452 | 0.251 | 5.799 | **< 0.001** |

Family = Gamma (all)          AIC = 82.0                    AIC =59.8                    AIC = 73.3

Table 2. GLM analysis of total mercury concentrarion (HgT) within each sampled tissue (liver, muscle and gills) of A. regius exposed to MeHg for 30 days, under crossed treatments of temperature (T, 2 levels: 19 ºC and 23 ºC) and CO2 (CO2, 2 levels: 400 µatm and 1500 µatm). Model formula on top, family and respective model AIC in the bottom. Est – Estimates; Std Error – Standard Error. Bold values indicate p < 0.05. For more details please see the R script in Supplemental Data.





### GLM: MDA in function of T * MeHg

|  | Est | Std Error | t value | p |
|---|---|---|---|---|
| (Intercept) | 0.026 | 0.003 | 8.055 | **< 0.001** |
| T | -0.010 | 0.005 | -2.163 | **0.036** |
| MeHg | -0.004 | 0.005 | -0.954 | 0.345 |
| T * MeHg | 0.014 | 0.007 | 2.174 | **0.035** |

Family = Gaussian                                        AIC = -277.2

**Table 3. GLM analysis of malondialdehyde (MDA) build-up in A. regius after 30 days exposed to crossed treatments of MeHg contamination (MeHg, 2 levels, non-contaminated and contaminated) and temperature (T, 2 levels: 19 ºC and 23 ºC). Model formula on top, family and respective model AIC in the bottom. Est – Estimates; Std Error – Standard Error. Bold values indicate $p < 0.05$. For more details please see the R script in Supplemental Data.**





### GLM: CAT in function of $CO_2$ * MeHg

|            | Est    | Std Error | t value | *p*       |
|------------|--------|-----------|---------|-----------|
| (Intercept) | 4.375  | 0.399     | 10.96   | **< 0.001** |
| $CO_2$      | -0.454 | 0.564     | -0.804  | 0.426     |
| MeHg        | 1.482  | 0.564     | 2.625   | **0.012** |
| $CO_2$ * MeHg | 1.313  | 0.818     | 1.605   | 0.116     |

Family = Gaussian                                    AIC = 166.2

### GLM: SOD in function of $CO_2$ * T + $CO_2$ * MeHg

|            | Est    | Std Error | t value | *p*       |
|------------|--------|-----------|---------|-----------|
| (Intercept) | 9.496  | 1.040     | 9.135   | **< 0.001** |
| T           | 3.346  | 1.200     | -2.787  | **0.008** |
| $CO_2$      | -1.264 | 1.484     | -0.852  | 0.399     |
| MeHg        | 1.614  | 1.200     | 1.344   | 0.186     |
| T * $CO_2$  | 6.319  | 1.744     | 3.623   | **< 0.001** |
| $CO_2$ * MeHg | -3.399 | 1.744   | -1.949  | 0.058     |

Family = Gaussian                                    AIC = 237.3

### GLM: GST in function of T * $CO_2$ * MeHg

|            | Est    | Std Error | t value | *p*       |
|------------|--------|-----------|---------|-----------|
| (Intercept) | 7.561  | 0.676     | 11.19   | **< 0.001** |
| T           | -1.174 | 0.955     | -1.229  | 0.227     |
| $CO_2$      | -2.320 | 0.955     | -2.428  | **0.020** |
| MeHg        | -2.054 | 0.955     | -2.150  | **0.038** |
| T * $CO_2$  | 4.076  | 1.351     | 3.017   | **0.005** |
| T * MeHg    | 2.375  | 1.351     | 1.758   | 0.087     |
| $CO2$ * MeHg | 4.427  | 1.351    | 3.277   | **0.002** |
| T * $CO_2$ * MeHg | -3.422 | 1.970 | -1.737  | 0.090     |

Family = Gaussian                                    AIC = 186.1

**Table 4. GLM analysis of oxidative stress response (CAT, SOD and GST) in A. regius after 30 days exposed to crossed treatments of MeHg exposure (MeHg, 2 levels: non-contaminated and contaminated), temperature (T, 2 levels: 19ºC and 23ºC) and CO2 (CO2, 2 levels: 400 µatm and 1500 µatm). Model formula on top, family and respective model AIC in the bottom. Est – Estimates; Std Error – Standard Error. Bold values indicate p < 0.05. For more details please see the R script in Supplemental Data**





### GLM: Hsp70 in function of Tissues

|  | Est | Std Error | t value | p |
|---|---|---|---|---|
| (Intercept) | 3.605 | 0.235 | 15.32 | **< 0.001** |
| Gills & Liver | 1.607 | 0.335 | 4.804 | **< 0.001** |
| Gills & Muscle | 0.975 | 0.333 | 2.929 | **0.004** |
| Muscle & Liver | 0.633 | 0.335 | 1.890 | 0.061 |

Family = Gaussian                               AIC = 481.5

### GLM: Gills Hsp70 in function of T + MeHg

| | | | | |
|---|---|---|---|---|
| (Intercept) | 3.561 | 0.255 | 13.99 | **< 0.001** |
| T | -0.530 | 0.299 | -1.775 | 0.083 |
| MeHg | 0.622 | 0.299 | 2.085 | **0.043** |

Family = Gaussian                               AIC = 146.4

### GLM: Muscle Hsp70 in function of T * $CO_2$

| | | | | |
|---|---|---|---|---|
| (Intercept) | 4.671 | 0.291 | 16.07 | **< 0.001** |
| T | -0.294 | 0.411 | -0.715 | 0.479 |
| $CO_2$ | -0.955 | 0.411 | -2.323 | **0.025** |
| T * $CO_2$ | -2.331 | 0.596 | 3.913 | **< 0.001** |

Family = Gaussian                               AIC = 137.0

### GLM: Liver Hsp70 in function of T + $CO_2$

| | | | | |
|---|---|---|---|---|
| (Intercept) | 5.376 | 0.593 | 9.064 | **< 0.001** |
| $CO_2$ | -0.588 | 0.839 | -0.702 | 0.487 |
| MeHg | 1.315 | 0.839 | -1.567 | 0.125 |
| $CO_2$ * MeHg | 3.627 | 1.235 | 2.938 | **0.005** |

Family = Gaussian                               AIC = 73.3

**Table 5. GLM analysis of heat shock protein 70 (Hsp70) production in A. regius tissues (gills, muscle and liver) and, posteriorly within tissues, under crossed treatments of MeHg exposure (MeHg, 2 levels, non-contaminated and contaminated), temperature (T, 2 levels: 19ºC and 23ºC) and CO2 (CO2, 2 levels: 400 µatm and 1500 µatm). Model formula on top, family and respective model AIC in the bottom. Est – Estimates; Std Error – Standard Error. Bold values indicate p < 0.05. For more details please see the R script in Supplemental Data.**