# Peer review of "Ocean acidification dampens warming and contamination effects on the physiological stress response of a commercially important fish"

_Biogeosciences, 2017_

## Referee Comment (RC1) · Anonymous Referee #1 · 14 Jun 2017

**General comments:**

This manuscript aims to address a topic of significant importance, namely the interaction between climate change stressors and contamination in coastal regions, and particularly its impact on species of commercial importance. This is certainly a topic of great interest and an area that has been identified as a significant knowledge gap in the field at present. Despite this potential and the undoubted requirement for such a study within the field, regrettably the manuscript presented here does not adequately address this question. As it stands there is insufficient detail presented throughout the

methods to adequately appraise what has been done, there appear to be a number of methodological oversights that hamper the interpretation of the results and this has, to a large extent, led to many of the conclusions drawn not being supported by the data. Based on these factors I believe the manuscript at least requires major revisions to include this required detail, as well as restructure the conclusions to match what has actually been undertaken. It would then require re-review to appraise the manuscript in its new form. If it is not possible to include this required detail in full, in its current state the manuscript is not of sufficient quality to be published.

**Specific comments:**

aĂć Abstract, discussion and conclusions – Throughout the manuscript the authors suggest the reduced accumulation of mercury in tissues under combined exposure is due to metabolic depression, and a subsequent reduced apatite/ingestion of food, initiated by elevated CO2. However, the authors do not measure any parameters in the current study that could confirm or counter this suggestion. There is no indication that these fish ingested less food so the conclusions, certainly as they are presented, are unfounded. Reduced accumulation could in fact be caused by a number of different mechanisms in the organism by which elevated COAn2 augmented Hg accumulation, either by metabolic depression, reduced appetite (could be caused by alternative mechanism), reduced digestive efficiency, reduced uptake across the gut epithelium, greater egestion of Hg or impacts of Hg transport and complexation in plasma to reduce delivery to measured tissues. All are potentially feasible and at present insufficient information is known about this to surmise it is metabolic depression. It is vital to indicate that whilst altered accumulation is noted, which differs between specific tissues, the mechanism is not known. Following this point, the authors have not cited two key references on ocean acidification and mercury contamination recently published (Li et al Scientific Reports 7;324 2017; and Wang et al ES&T 51:5820 2017). It is possible these were published after initial submission of the current manuscript, but in light of altered mercury accumulation under elevated CO2 these two manuscripts are key as

**BGD**
they support the current finding.

ć Abstract, discussion and conclusions – Similarly to the point above, the authors repeatedly suggest elevated H+ impacts mercury accumulation/toxicity at a molecular level, but no acid base measures were made. Also it is a common misunderstanding that elevated CO2 results in chronic acidosis in fish plasma, this is not the case. Elevated CO2 results in acute acidosis which is rapidly compensated for by an elevation in bicarbonate, returning the plasma H+ to normal levels. Therefore the suggestion that elevated H+ impacts on mercury toxicity/accumulation is not supported, especially as acid-base parameters are not presented that counter this common response noted in acid-base compensating species such as fish. The authors need to again re-interpret data and re-write conclusions to better reflect the demonstrated results and not make broad unsupported conclusions, pinned loosely on previously published literature that has been misinterpreted/misunderstood.

ć Discussion – The authors suggest the Fulton condition may diminish under mercury contamination. Whilst AIC indicates the best fit model as slightly negative the statistic (p-Value) clearly indicates no significant effect and therefore suggesting this is not the case may mislead readers to interpret a result that is not supported statistically, even if it may support a previous publication.

åÅć Intro and methods - The justification of mercury, and methylmercury, in fish from coastal regions is insufficient. There is no quantification of levels within the environment from different regions globally, how coastal compares to open ocean and how this then translates into a burden for fish populations. As it stands this is not adequate for a contaminant manuscript, and belies the statement that an environmentally relevant concentration was used, as stated in the methods. What is an environmentally relevant concentration, where does the level chosen fit with measured environmental levels from different regions globally, and even just within the region the study was undertaken in. Finally, if the route of uptake is solely dietary for fish then how do environmental levels correspond to burdens in prey species and thus exposure in the experimental organ-
ism? Is the level chosen a typical contaminant level in prey species in an impacted environment or the level in water/sediment? This needs clarifying, and fully justifying in relation to existing literature and levels previously used.

ć Methods - Following this, the total amount of mercury (mg per kg of food), is higher than the content of methylmercury added as an additive (8.02 MeHg, 8.28 HgT). This is not possible. Also how were these levels measured (or is it nominal)?

 $\hat{a}A\dot{c}$  Methods (page 4 lines 1-7) - The description of the conditions, and particularly their maintenance is not sufficient. It states ammonia, nitrate and nitrite were regularly monitored and kept within recommended levels. How was this tested, what were the accepted levels and what were the levels measured within the experiment? Also how were high levels mitigated against and how often? Furthermore, it mentions salinity was kept at  $35.0 \pm 1.0$  g/l NaCl? The probe listed is a conductivity probe so does not measure in g/l of NaCl but gives a conductivity measure or salinity as a psu. Also how was salinity maintained? i.e. is this addition of deionised water to compensate for evaporation? Or additional NaCL addition. Any further addition of NaCl would significantly alter osmolality thus this needs clarifying to explain if input water fluctuated in salinity. A better description of this process is therefore required.

ć Methods - There is no measure (or data presented) of methylmercury or total mercury in experimental water. This is a major omission, and gives no indication as to what proportion of the contaminant leaches from food into water, particularly if any food remains uneaten and in the tank for any time. It also prevents the discussion of amounts of methylmercury that egested immediately into the water by this fish, not being taken up or bioaccumulated.

ć Throughout - Given the commercial importance of the species, one surprising oversight is the fact that no discussion on different tissue burdens were made with respect to human consumption and climate change impacts. The only place this is
alluded to is in the title! This is particularly relevant given the possibility that elevated CO2 reduces Hg accumulation possibly reducing transfer of hg into humans directly via consumption of muscle tissue, which could be an important result. This would provide some wider context in which to place the importance of this study generally, as well as contaminant/climate changes studies more generally.

Technical corrections:

Page 1, Line 18-19 – Sentence beginning "Despite the more than likely cooccurrence..." is weak and doesn't read well. Needs stronger justification (see above) to enable stronger conviction in abstract, as well as explicitly highlight that contaminant/climate change stressor interactions are largely overlooked, rather than just "these stressors".

Page 1, Line 29 – should read mechanisms not mechanism

Page 2, Line 2 (and throughout) – should be CO2 sub-scripted, this error occurs in a number of positions throughout manuscript, also sometimes is sub-scripted so inconsistent.

Page 2, Line 4-5 –I would argue greenhouse gas effect is increasing global temperatures, and this is resulting in projected further increase (already increased by  $0.76^{\circ}$ C from pre-industrial) in surface ocean temperature of ... by end of the century.

Page 2, Line 22 – Should read "...Sampaio et al., 2016) and ultimately mortality (Coccini et al., 2000)."

Page 3, Line 7 - protein not proteins

Page 3, line 11 – responses not response

Page 3, Line 16 - remove the before estuaries

Page 4, Line 10 - should be pH controllers not controller

BGD
Page 5, Line 5 - Length3 should be super-scripted

Page 5, Line 19 - remove with before nitric acid

Page 5, Line 23 – should be gill not gills

Page 5, Line 25 - remove posteriorly

Page 5, Line 26 - rewrite as "..response concentrations, quantified" removing were

Page 6, Line 18 – assume is potassium periodate not potassium per iodate

Page 6, Line 23 (and page 7, line 16) – mg-2 needs super-scripting

Page 6, Line 25 – insert space before Superoxide

Page 7, Line 5 – is the % inhibition of SOD activity calculated as maximum inhibition, average inhibition at each 5 minute time point or from initial and final, just measured every 5 minutes over 25 minutes so potentially have different rates of inhibition and total overall inhibition over this time course

Page 7, Line 23 – insert space before and

Page 8, Line 2 - insert space in mg-1total

Page 9, Line 19 (and other places) – A. regius needs italicising

Page 10, Line 2 – notoriously is an odd choice of words, suggest just removing as reads fine without replacing

Page 10, Line 23 (and page 11, line 15) – H20 needs subscripting

Page 18 - Why is the x-axis reversed on figure 1, d, compared to b and c. This confuses comparisons.

---

## Referee Comment (RC2) · M. Orte (Referee) · 27 Jun 2017

**General comments**

The interactive effects of acidification, warming and the presence of the metal Hg was assessed in the Fish Argyrosomus regius. Bioaccumulation of Hg was measured in different organs of the fish and sublethal toxic responses were also analyzed by the use of biomarkers. The topic is highly relevant since research regarding global change issues should preferably focus on a multi-stressors approach. Furthermore, mercury is an

important persistent contaminant found in coastal environments around the world and information regarding its interactive toxicological effects with other parameters such as acidification and warming are of great value. In general, the writing is clear and the data obtained is interesting. However, some issues regarding the methodological approach used are not well explained and there are some information at the results and discussion section that should be included. Therefore I recommend that the authors perform the suggested corrections before the article is published.

Specific comments

Introduction and discussion

The focus of the study is the evaluation of toxic responses of the metal Hg in a global change scenario. It is mentioned that concentration of Hg was chosen according to environmental measurements, however data on the range of toxic concentrations of this metal to this species or other fish species is not included. Considering that the article uses an ecotoxicological approach and therefore it is based on dose-response concentration it is crucial that more details on this subject is included, such as values of toxicity for fishes and environmental values within contaminated and non contaminated areas, especially in the area where the study was conducted.

In the discussion section, comparative results of mercury accumulation and biomarker response are missing. The study of Biomarkers is quite complex as responses can be influenced by many parameters. In this sense, there are several studies on biomarker response to mercury in the literature. Such studies should also be mentioned to provide information on the sensitivity of this species comparing to others, as well as to know the relevance of the used Hg concentration.

In the abstract, (page 1 line 20), introduction (page 3 line 20) and methodology (page 4 line 23) pCO2 concentration is given as 1100 $\mu$atm, while the actual value used was 1500 $\mu$atm. Please correct.

Methodology

The fishes were taken from an aquaculture station. Were the physico-chemical parameters measured at the station? This is relevant to known the levels of pH and temperature that organisms were acclimated at the long-term.

Page 4 Line 5- Ammonia levels is an important issue at toxicity tests, especially with fishes, as it can interfere on the toxic responses. Authors mention that ammonia (along with nitrate and nitrite) levels were kept within recommended levels. How was this performed? What are the recommended levels? Please give more details.

Salinity should be given as psu or without unit.

Page 4 line 13- Please give more details on alkalinity measurements, such as the equipment used, storage of samples, the use of certified materials. . ..

Page 4 Line 20- The method for mercury contamination is confusing. MeHg exposure was performed by food intake and fished were fed two to three times a day. How was the difference between food intakes measured? Authors states that ingestion decreased due to changes in metabolism, but how was this measured? Where is this result? How much mercury was given as total in the experiment? How much of this metal remain dissolved in the water column?

In the experimental set-up, the setup "IV" is the same as the setup "II", 19 °C, 400 pCO2 $\mu$atm and contaminated feed (MeHg: 8.02 mg kgÂź; HgT: 8.28 mg kgÂź). Setup IV should be 19 °C , 1500$\mu$atm and contaminated feed.

Results

In the methodology section, it is mentioned that Reference material was also used to validate measurements of metal content. However, results of recovery percentage in not given. Please include this data as it validates the measurements.

Page8line20-25 concentration of Hg was lower in muscle but concentration in liver and

gills was actually the same considering error between replicates.

Figure 1d the 400 and 1500 $\mu$atm are inverted.

Page 8 line27- As expected, catalase activity was affected by mercury contamination, but was this biomarker affected by pCO2 also? What about warming? This is briefly mentioned in the discussion section, but the results are not given.

While the values for Hsp70 are given in each organ analyzed, the results for the other biomarkers are not specified. Were they measured only in the liver or other parts? Please include this information in the results and also in the methodology.

Discussion

Page 9 lines 15-20 the information "However, our AIC-chosen best model indicated that mercury may diminish organism Fulton condition" is contradictory to what is mentioned on the results : "Fulton condition (K) did not show any significant differences between treatments (MeHg, p > 0.05, GLM analysis in Table 1)."

Technical corrections

Page 2 Lines 1-2: CO2 should be subscript Page 4 Line 2: m3 should be superscript Pag 4 Line 10: CO2 should be subscript Pag 5 Line 5: lenght3 check type error Page 6 Line 12: mg-1 should be superscript Page 6 Line 23: mg-2 should be superscript, Pag 10 Line 20: H+ should be superscript Page 9 line 17: the word non-lethal could be replaced by sublethal, which is more often used in toxicity studies Page 9 line19: A. regious should be written in italic

---

## Author Comment (AC1) · 4 Jul 2017

General comments

"This manuscript aims to address a topic of significant importance, namely the inter-action between climate change stressors and contamination in coastal regions, and particularly its impact on species of commercial importance. This is certainly a topic of great interest and an area that has been identified as a significant knowledge gap in

the field at present. Despite this potential and the undoubted requirement for such a study within the field, regrettably the manuscript presented here does not adequately address this question. As it stands there is insufficient detail presented throughout the methods to adequately appraise what has been done, there appear to be a number of methodological oversights that hamper the interpretation of the results and this has, to a large extent, led to many of the conclusions drawn not being supported by the data. Based on these factors I believe the manuscript at least requires major revisions to include this required detail, as well as restructure the conclusions to match what has actually been undertaken. It would then require re-review to appraise the manuscript in its new form. If it is not possible to include this required detail in full, in its current state the manuscript is not of sufficient quality to be published."

Response: We thank the referee for his suggestions which have served to greatly improve the manuscript. We hope we have now provided sufficient detail on the methodologies employed in this work. We also hope to have clarified some misinterpretations throughout the text. Below, we reply to each comment in a point-by-point manner. Please note that Page and Line numbers now correspond to the marked up version of the manuscript.

Specific comments

Comment #1: Abstract, discussion and conclusions – Throughout the manuscript the authors suggest the reduced accumulation of mercury in tissues under combined exposure is due to metabolic depression, and a subsequent reduced apatite/ingestion of food, initiated by elevated $CO_2$. However, the authors do not measure any parameters in the current study that could confirm or counter this suggestion. There is no indication that these fish ingested less food so the conclusions, certainly as they are presented, are unfounded. Reduced accumulation could in fact be caused by a number of different mechanisms in the organism by which elevated $CO_2$ augmented Hg accumulation, either by metabolic depression, reduced appetite (could be caused by alternative mechanism), reduced digestive efficiency, reduced uptake across the gut epithelium,

greater egestion of Hg or impacts of Hg transport and complexation in plasma to re-duce delivery to measured tissues. All are potentially feasible and at present insuffi-cient information is known about this to surmise it is metabolic depression. It is vital to indicate that whilst altered accumulation is noted, which differs between specific tis-sues, the mechanism is not known. Following this point, the authors have not cited two key references on ocean acidification and mercury contamination recently published (Li et al Scientific Reports 7;324 2017; and Wang et al ES&T 51:5820 2017). It is possible these were published after initial submission of the current manuscript, but in light of altered mercury accumulation under elevated CO2 these two manuscripts are key as they support the current finding.

Response: The authors acknowledge that no additional parameters were measured to validate the conclusion that lower Hg accumulation under increased CO2 was due to metabolic depression. Nonetheless, based on previous studies, there are reasons to believe this is the case. A wide range of organisms show metabolic decrease in response to increased extracellular acid–base stress(Kroeker et al., 2010), and es-pecially to simultaneous occurrence of warming and acidification (Harley et al., 2006; Harvey et al., 2013; Rosa et al., 2013; Rosa and Seibel, 2008). Concerning CO2, theoretically, the prioritization of acid–base regulation and ion regulatory enzyme ma-chinery for CO2 excretion (e.g. pyruvate kinase) may lead to lower metabolic activity in other enzymes (energy reallocation),as reported in other fish (Perry et al., 1988). As MeHg accumulation rates are positively correlated with metabolic rates (Dijkstra et al., 2013), these results would support the claim that acidification affects toxic compound accumulation rates (Schiedek et al., 2007). Given our simultaneous exposure to both warming and acidification, which has been shown to undeniably suppress metabolic rates directly (Christensen et al., 2011; Harley et al., 2006; Rosa et al., 2013; Rosa and Seibel, 2008; see also Harvey et al., 2013 and Kroeker et al, 2010) we still hold the conviction that metabolic processes may be at play. However, the authors also acknowledge that they were not aware of the recent research pointed out by the ref-eree, which thoroughly picks apart the causes of behind these mechanisms. We thank

the referee for this useful comment and we have altered our interpretation, changing the text: "However, such effect may be offset by $CO_2$-linked decreases in mercury accumulation (Sampaio et al., 2016; Schiedek et al., 2007; Wang et al., 2017)" (Page 2, Lines 29-31) "Instead, our results support recent studies demonstrating that hypercapnia dampens Hg accumulation in marine organisms (Li et al., 2017; Sampaio et al., 2016; Wang et al., 2017). There are several possible reasons which may underpin such an interaction, encompassing digestive (reduced digestive efficiency, reduced uptake through the gut membrane, reduced appetite, increased Hg depuration) and molecular (competition between Hg and $H+$ ions for binding sites, impacts on Hg plasma transport, lower phospholipidic membrane permeability) mechanisms (Li et al., 2017). A recent study has also found that the lysosome-autophagy pathway was upregulated by combined exposure to Hg and increased $CO_2$, enabling better animal fitness which may potentially reduce Hg accumulation and toxicity (Wang et al., 2017). In addition, taking into account that the occurrence of both warming and acidification changes physiological thresholds (Christensen et al., 2011; Harley et al., 2006; Rosa et al., 2013; Rosa and Seibel, 2008), a degree of metabolic depression may also play a role on decreasing HgT accumulation (Dijkstra et al., 2013; Sampaio et al., 2016)." (Page 11, Lines 4-16) "In general, warming conditions enhanced MeHg accumulation but $CO_2$-linked impacts countered this effect." (Page 12, Lines 28-29)

References Dijkstra, J. A., Buckman, K. L., Ward, D., Evans, D. W., Dionne, M. and Chen, C. Y.: Experimental and Natural Warming Elevates Mercury Concentrations in Estuarine Fish, PLoS One, 8(3), 1–9, doi:10.1371/journal.pone.0058401, 2013. Harley, C. D. G., Hughes, A. R., Kristin, M., Miner, B. G., Sorte, C. J. B. and Carol, S.: The impacts of climate change in coastal marine systems, Ecol. Lett., 9, 228–241, 2006. Harvey, B. P., Gwynn-Jones, D. and Moore, P. J.: Meta-analysis reveals complex marine biological responses to the interactive effects of ocean acidification and warming., Ecol. Evol., 3(4), 1016–1030, doi:10.1002/ece3.516, 2013. Kroeker, K. J., Kordas, R. L., Crim, R. N. and Singh, G. G.: Meta-analysis reveals negative yet variable effects of ocean acidification on marine organisms., Ecol. Lett.,

13(11), 1419–34, doi:10.1111/j.1461-0248.2010.01518.x, 2010. Li, Y., Wang, W.-X. and Wang, M.: Alleviation of mercury toxicity to a marine copepod under multigenerational exposure by ocean acidification, Sci. Rep., 7(1), 324, doi:10.1038/s41598-017-00423-1, 2017. Perry, S. F., Walsh, P. J., Mommsen, T. P. and Moon, T. W.: Metabolic consequences of hypercapnia in the rainbow trout Salmo airdneri : B-adrenergic effects, Gen. Comp. Endocrinol., 69, 439–447, 1988. Rosa, R. and Seibel, B. A.: Synergistic effects of climate-related variables suggest future physiological impairment in a top oceanic predator, Proc. Natl. Acad. Sci., 105(52), 20776–20780, doi:10.1073/pnas.0806886105, 2008. Rosa, R., Trübenbach, K., Repolho, T., Pimentel, M., Faleiro, F., Boavida-Portugal, J., Baptista, M., Lopes, V. M., Dionísio, G., Leal, M. C., Calado, R. and Pörtner, H. O.: Lower hypoxia thresholds of cuttlefish early life stages living in a warm acidified ocean., Proc. Biol. Sci., 280(1768), 20131695, doi:10.1098/rspb.2013.1695, 2013. Sampaio, E., Maulvault, A. L., Lopes, V. M., Paula, J. R., Barbosa, V., Alves, R., Pousão-Ferreira, P., Repolho, T., Marques, A. and Rosa, R.: Habitat selection disruption and lateralization impairment of cryptic flatfish in a warm, acid, and contaminated ocean, Mar. Biol., 163(10), 217, doi:10.1007/s00227-016-2994-8, 2016. Schiedek, D., Sundelin, B., Readman, J. W. and Macdonald, R. W.: Interactions between climate change and contaminants, Mar. Pollut. Bull., 54, 1845–1856, doi:10.1016/j.marpolbul.2007.09.020, 2007. Wang, M., Lee, J.-S. and Li, Y.: Global Proteome Profiling of a Marine Copepod and the Mitigating Effect of Ocean Acidification on Mercury Toxicity after Multigenerational Exposure, Environ. Sci. Technol., 51, 5820–5831, doi:10.1021/acs.est.7b01832, 2017a.

Comment #2: Abstract, discussion and conclusions – Similarly to the point above, the authors repeatedly suggest elevated H+ impacts mercury accumulation/toxicity at a molecular level, but no acid base measures were made. Also it is a common misunderstanding that elevated CO2 results in chronic acidosis in fish plasma, this is not the case. Elevated CO2 results in acute acidosis which is rapidly compensated for by an elevation in bicarbonate, returning the plasma H+ to normal levels. Therefore the suggestion that elevated H+ impacts on mercury toxicity/accumulation is not supported,

especially as acid-base parameters are not presented that counter this common response noted in acid-base compensating species such as fish. The authors need to again re-interpret data and re-write conclusions to better reflect the demonstrated results and not make broad unsupported conclusions, pinned loosely on previously published literature that has been misinterpreted/misunderstood.

Response: We would like to point out that we never said that acidosis was present in a long-term perspective, nor did we assume that fish are not able to acid-base compensate, a mechanism that is already extensively described (Brauner and Baker, 2009; Heuer and Grosell, 2014; Michaelidis et al., 2007; among many more). In fact, besides some logistical and time constrains, that was the main reason why no acid-base measurements were performed. Having said that, acid-base compensation occurs mainly by increasing bicarbonate ($HCO_3-$) levels in both blood and cellular, which in turn leads to a normalization of intracellular and extracellular pH (Heuer and Grosell, 2014; Michaelidis et al., 2007). The chemical equation that underpins this reaction is as follows: $CO_2 + H_2O$ ⇌ $H_2CO_3$ ⇌ $H+ + HCO_3-$

Thus, despite pH being normalized by balancing the ratio between $H+ + HCO_3-$ and $H_2CO_3$ (it generally stabilizes at ∼0.05/0.1 units lower than in normocapic conditions), it is important to note that H+ levels in the organism are still increased relatively to basal levels, especially in long-term acclimations to hypercapnia- where there is a constant influx of H+ ions (Heuer and Grosell, 2014; Michaelidis et al., 2007). Moreover, due to cell prioritization, intracellular and extracellular pH often display significantly different values: the former is up-regulated to normocapnic levels or higher, while the latter generally stabilizes at lower pH (ΔpHcan reach∼0.3-0.7) (Brauner and Baker, 2009; Heuer and Grosell, 2014). This also partially contributes to increased H+ levels. As our reasoning is grounded on molecular interactions (both oxidative stress-inducing and ROS-mitigating) of increased H+ chemical reactions (see also Dean, 2010), it does not imply for fish acid-base compensation to fail. In light of the new recent studies mentioned by the referee, we have introduced some new considerations to our Abstract/Discussion/Conclusion. The reason we do not believe that the lysozyme-autophagy pathway (Wang et al., 2017) is solely responsible for the antagonistic relationship between stressors is that it does not account for hypercapnia-induced oxidative stress and chaperone activation. Within this context, we have rephrased our interpretations and changed the text accordingly:

In the Abstract: "Together with $CO_2$-promoted removal of damaged proteins and enzymes, we argue that simultaneous increase in hydrogen ($H^+$) and reactive oxygen species (e.g. $O_2^-$) radicals is partially compensated through chemical reaction equilibrium balancing." (Page 1, Lines 26-29)

In the Discussion: "Increased $CO_2$ (co-occuring with Hg contamination) may elicit the up-regulation of the lysosome-autophagy pathway, which is responsible for removing damaged proteins and organelles, effectively reducing oxidative stress (Wang et al., 2017). This mechanism may contribute to alleviate not only Hg induced stress, but also warming-related oxidative stress. We also argue that this antagonistic relation can be partially explained by a $CO_2$-related increase of $H^+$ ion concentrations in the blood and cellular surroundings, counterbalanced by bicarbonate increase (acid-base compensation) to normalize pH levels (Heuer and Grosell, 2014; Michaelidis et al., 2007). By itself, the presence of excessive $H^+$ ions activates free radical neutralizing defenses (Tiedke et al., 2013), which is in line with the present findings when hypercapnia was the sole stressor. However the production of $O_2^-$ and further complementary ROS radicals (e.g. $OH^-$) by other stressors may result in facilitated $H_2O$ and $H_2O_2$ formation, due to chemical reactions balancing equilibrium (e.g. $H^+ + OH^- ⇄ H_2O$), thus eliminating free radicals and decreasing activity of antioxidant enzymes to basal standards." (Page 11/12, Lines 31/1-10) "More so than for oxidative stress, the enhanced removal of damaged proteins and enzymes indirectly promoted by increased $CO_2$ (via up-regulated lysosome-autophagy) may have especially contributed to subside protein chaperone production. Given that Hsp70 production can also be stimulated by high ionic (e.g. $H^+$) concentrations (Feder and Hofmann, 1999), we reason that the same

additional mechanism by which hypercapnia potentially modulates oxidative stress can be applied for heat shock response" (Page 12, Lines 18-22)

In the Conclusions: "In fact, despite negative effects prompted as a sole stressor, acidification consistently elicited antagonistic responses to temperature and contamination effects on oxidative stress (including heat shock response), which may be explained by stimulated removal of damaged proteins and organelles (Wang et al., 2017). Moreover, we also argue that the mechanistic interactions found are coadjuvanted by the coinciding increase of hydrogen (H+) and radical reactive oxygen species (e.g. O2-, OH-), which subsequently nullify each other due to the spontaneous equilibrium of chemical reactions (e.g. H+ + OH- ⇌ H2O)." (Page 12/13, Lines 29/1-6)

References Brauner, C. J. and Baker, D. W.: Patterns of Acid–Base Regulation During Exposure to Hypercarbia in Fishes, in Cardio-Respiratory Control in Vertebrates, edited by M. L. Glass and S. C. Wood, pp. 1–546, Springer Berlin Heidelberg., 2009. Dean, J. B.: Hypercapnia causes cellular oxidation and nitrosation in addition to acidosis: implications for CO2 chemoreceptor function and dysfunction., J. Appl. Physiol., 108(6), 1786–95, doi:10.1152/japplphysiol.01337.2009, 2010. Heuer, R. M. and Grosell, M.: Physiological impacts of elevated carbon dioxide and ocean acidification on fish, AJP Regul. Integr. Comp. Physiol., 307(9), R1061–R1084, doi:10.1152/ajpregu.00064.2014, 2014. Michaelidis, B., Spring, A. and Pörtner, H. O.: Effects of long-term acclimation to environmental hypercapnia on extracellular acid-base status and metabolic capacity in Mediterranean fish Sparus aurata, Mar. Biol., 150(6), 1417–1429, doi:10.1007/s00227-006-0436-8, 2007.

Comment #3: Discussion – The authors suggest the Fulton condition may diminish under mercury contamination. Whilst AIC indicates the best fit model as slightly negative the statistic (p-Value) clearly indicates no significant effect and therefore suggesting this is not the case may mislead readers to interpret a result that is not supported statistically, even if it may support a previous publication.
Response: Following the reviewer's instructions, we have removed any mentioning of negative effects on the Fulton condition. We have changed the introductory paragraph of the Discussion to: "The present study showed that Hg contamination, ocean warming and acidification interactively affected fish physiology at sublethal levels, i.e. zero mortality and also no effects on Fulton condition were registered. The fact that the meagre (A. regius) is a very resilient species and easily adapts to environmental alterations (Monfort, 2010) may explain the absence of deleterious effects at an organism level, after 30 days of exposure." (Page 10, Lines 14-19)

Comment #4: Intro and methods - The justification of mercury, and methylmercury, in fish from coastal regions is insufficient. There is no quantification of levels within the environment from different regions globally, how coastal compares to open ocean and how this then translates into a burden for fish populations. As it stands this is not adequate for a contaminant manuscript, and belies the statement that an environmentally relevant concentration was used, as stated in the methods. What is an environmentally relevant concentration, where does the level chosen fit with measured environmental levels from different regions globally, and even just within the region the study was undertaken in. Finally, if the route of uptake is solely dietary for fish then how do environmental levels correspond to burdens in prey species and thus exposure in the experimental organism? Is the level chosen a typical contaminant level in prey species in an impacted environment or the level in water/sediment? This needs clarifying, and fully justifying in relation to existing literature and levels previously used.

Response: Mercury (originating mainly from industrial residue) accumulates in the sediments of river basins and estuaries (Mason, 2001). Posteriorly, it is transported to the open ocean via particulate and dissolved sediments in water currents and accumulated within animals, but in much less quantity (Guentzel et al., 1996). Thus, it is logical that fish which often make use of estuaries are more vulnerable to mercury accumulation, as we have stated in the Introduction of the manuscript (Page 2, Lines 18-20 and Page 3, Lines 21-22). As the referee correctly inferred, the concentrations of mercury used

for this study were based on levels of contamination found in contaminated coastal areas (specifically the extensively studied, contaminated estuary of Aveiro, Portugal) for species that are natural prey of the meagre(e.g. Cardoso et al., 2014; Nunes et al., 2008). These mercury concentrations can also be found in other areas globally, e.g. Florida, USA (Kannan et al., 1998). We have changed the text in order to provide a more comprehensive picture: "Given our dietary option, ecologically relevant MeHg concentrations were chosen based on levels (low contamination, $\sim$0.12 mg kg-1 wet weight (ww); and high contamination, $\sim$1.6 mg kg-1 ww found in common A. regius prey species from contaminated coastal areas (Cardoso et al., 2014; Kannan et al., 1998; Nunes et al., 2008). The pellets given to fish allocated to non-contaminated and contaminated treatments had approximately 0.60 ± 0.01 mg kg⁻Âź dry weight (dw) and 8.02 ± 0.01 mg kg⁻Âź dw of MeHg, respectively, which were considered to mimic the concentrations found in the field (see Maulvault et al., 2016, 2017). Feed composition, manufacturing and MeHg spiking processes were executed as described by Maulvault et al. (2016)." (Page 4/5, Lines 30-32/1-6)

References Cardoso, P. G., Pereira, E., Duarte, A. C. and Azeiteiro, U. M.: Temporal characterization of mercury accumulation at different trophic levels and implications for metal biomagnification along a coastal food web, Mar. Pollut. Bull., 87(1), 39–47, doi:10.1016/j.marpolbul.2014.08.013, 2014. Guentzel, J. L., Powell, R. T., Landing, W. M. and Mason, R. P.: Mercury associated with colloidal material in an estuarine and an open- ocean environment, Mar. Chem., 55(1–2), 177–188, doi:10.1016/S0304-4203(96)00055-2, 1996. Kannan, K., Smith Jr., R. G., Lee, R. F., Windom, H. L., Heitmuller, P. T., Macauley, J. M. and Summers, J. K.: Distribution of Total Mercury and Methyl Mercury in Water , Sediment , and Fish from South Florida Estuaries, Arch. Environ. Contam. Toxicol., 34, 109–118, doi:10.1007/s002449900294, 1998. Mason, R. P.: The Bioaccumulation of Mercury, Methylmercury and Other Toxic Elements into Pelagic and Benthic Organisms, in Coastal and Estuarine Risk Assessment, edited by N. Newman, M. Roberts jr, and R. Hale, pp. 127–149, Lewis Publishers, Boca Raton, Fl, USA., 2001. Nunes, M., Coelho, J. P., Cardoso, P. G., Pereira, M. E., Duarte, A.

C. and Pardal, M. A.: The macrobenthic community along a mercury contamination in a temperate estuarine system (Ria de Aveiro, Portugal), Sci. Total Environ., 405(1–3), 186–194, doi:10.1016/j.scitotenv.2008.07.009, 2008.

Comment #5: Methods - Following this, the total amount of mercury (mg per kg of food), is higher than the content of methylmercury added as an additive (8.02 MeHg, 8.28 HgT). This is not possible. Also how were these levels measured (or is it nominal)? Response: We assure the reviewer that this is standard for all scientific works where, being the most bioaccumulated form of mercury in the environment, methylmercury (MeHg) is used (Maulvault et al., 2016, 2017; Sampaio et al., 2016; Wang et al., 2013, 2017b). On top of naturally occurring demethylation, higher total mercury concentration is due to the ubiquity of mercury, under several (organic and inorganic) forms, in the natural environment. A standard feed diet is composed of fish meals, oils and other compounds, which already contain a certain quantity of mercury (not only methylmercury, but also in its other chemical forms). Naturally, a control feed, where no spiking is performed, contains trace levels of mercury. Thus, when spiking a diet with MeHg, adding these facts, it is common for the total amount of mercury to be higher than that of methylmercury (e.g. see studies referenced). Lastly, the procedure for the measurement of MeHg is similar to HgT, explicit in section "2.2 Total mercury and Methylmercury accumulation" (Page 6, Lines 5-18). For the sake of clarity, we rephrased: "Afterwards, HgT (all samples) and MeHg (feed samples) were determined (10-15 mg for solids or 100-200 $\mu$l for liquids) by atomic absorption spectrometry (AAS), following EPA (2007) by means of an automatic Hg analyser (AMA 254, LECO, USA) with a detection threshold of 0.005 mg kg-1 ww." (Page 6, Line 5-8) And added: "Feed composition, manufacturing and MeHg spiking processes were executed as described by Maulvault et al. (2016). Fish were fed two to three times a day and total feed quantity provided per day was approximately 1% (standard calculation for aquaculture) of animal weight (at the end of 30 days, each fish was given approximately 0.0106 mg of HgT). Selected feed quantity also minimized food remains, which, in case of existing, were siphoned together with fish faeces after feeding." (Page 5, Line 5-12)

References Maulvault, A. L., Custodio, A., Anacleto, P., Repolho, T., Pousao, P., Nunes, M. L., Diniz, M., Rosa, R. and Marques, A.: Bioaccumulation and elimination of mercury in juvenile seabass (Dicentrarchus labrax) in a warmer environment, Environ. Res., 149, 77–85, doi:10.1016/j.envres.2016.04.035, 2016. Maulvault, A. L., Barbosa, V., Alves, R., Custódio, A., Anacleto, P., Repolho, T., Pousão Ferreira, P., Rosa, R., Marques, A. and Diniz, M.: Ecophysiological responses of juvenile seabass ( Dicentrarchus labrax ) exposed to increased temperature and dietary methylmercury, Sci. Total Environ., 586, 551–558, doi:10.1016/j.scitotenv.2017.02.016, 2017. Sampaio, E., Maulvault, A. L., Lopes, V. M., Paula, J. R., Barbosa, V., Alves, R., Pousão-Ferreira, P., Repolho, T., Marques, A. and Rosa, R.: Habitat selection disruption and lateralization impairment of cryptic flatfish in a warm, acid, and contaminated ocean, Mar. Biol., 163(10), 217, doi:10.1007/s00227-016-2994-8, 2016. Wang, R., Feng, X. Bin and Wang, W. X.: In vivo mercury methylation and demethylation in freshwater tilapia quantified by mercury stable isotopes, Environ. Sci. Technol., 47(14), 7949–7957, doi:10.1021/es3043774, 2013. Wang, X., Wu, F. and Wang, W.-X.: In Vivo Mercury Demethylation in a Marine Fish ( Acanthopagrus schlegeli ), Environ. Sci. Technol., (May), acs.est.7b00923, doi:10.1021/acs.est.7b00923, 2017b.

Comment #6: Methods (page 4 lines 1-7) - The description of the conditions, and particularly their maintenance is not sufficient. It states ammonia, nitrate and nitrite were regularly monitored and kept within recommended levels. How was this tested, what were the accepted levels and what were the levels measured within the experiment? Also how were high levels mitigated against and how often? Furthermore, it mentions salinity was kept at 35.0 $\pm$ 1.0 g/l NaCL? The probe listed is a conductivity probe so does not measure in g/l of NaCl but gives a conductivity measure or salinity as a psu. Also how was salinity maintained? i.e. is this addition of deionised water to compensate for evaporation? Or addition of additional NaCl? The description is confusing, and could be interpreted as additional NaCL addition. Any further addition of NaCl would significantly alter osmolality thus this needs clarifying to explain if input water fluctuated in salinity. A better description of this process is therefore required.

Response: We apologize for not having provided more detail on these matters, but we have been said to be overzealous with these descriptions in recent publications. However, it is our pleasure to fill the gaps the referee points out in this section. Ammonia (NH3/NH4+), nitrite (NO2-) and nitrate (NO3-) concentrations were daily checked (Colorimetric kits, Aquamerk, Germany), and kept below detectable levels (i.e. NH3/NH4+ < 0.25 mg l-1; NO2- < 0.10 mg l-1; NO3-< 0.2 mg l-1). Salinity was not measured through a conductivity probe, we apologize for the omission. We opted instead for a refractometer (V2, TMC Iberia, Portugal) and took daily measurements as with temperature and pH. Salinity was also incorporated in the calculation of seawater carbonate chemistry (Table S1). We acknowledge the mistake (g/l) and have removed salinity units (see below). The addition of deionised water or any kind of water except sea water would modify carbonate chemistry and render our pH manipulation useless (Cornwall and Hurd, 2015). All potential fluctuations in both these parameters were solved by the seawater flux, and in the case of nutrients, by the biological filter described (Page 4, Lines 2-5). As detailed in the Methods section (Page 4, Lines 5-9), each experimental unit (or recirculatory aquatic system, RAS) was a semi-closed system with a constant seawater flux (complete turnover rate in 24h) precisely to maintain parameters such as salinity and nutrients. Thus, the mitigation of potential problems was done a priori and no additional action was needed during the course of the experiment. We have added the pertinent information in the text: "To prevent fluctuations in environmental parameters, each RAS worked as a semi-closed system, with constant low flow external water input (flux > 2 l h-1; 50 l tank turnover rate = 24 h). Consequently, ammonia (NH3/NH4+), nitrite (NO2-) and nitrate (NO3-) concentrations were daily checked (Colourimetric kits, Aquamerk, Germany), and kept below detectable levels (i.e. NH3/NH4+ < 0.25 mg l-1; NO2- < 0.10 mg l-1; NO3- < 0.20 mg l-1), and salinity was kept at 35.0 ± 1.0 (V2 Refractometer, TMC Iberia, Portugal). Temperature and pH (multiparametric probe, Multi3420 SET G, WTW) were measured daily, directly in the holding tanks. Photoperiod was fixed at 12 h light : 12 h dark." (Page 4, Lines 5-14). References Cornwall, C. E. and Hurd, C. L.: Experimental design in ocean acidification

research: problems and solutions, ICES J. Mar. Sci., 73, 572–581, 2015.

Comment #7: Methods - There is no measure (or data presented) of methylmercury or total mercury in experimental water. This is a major omission, and gives no indication as to what proportion of the contaminant leaches from food into water, particularly if any food remains uneaten and in the tank for any time. It also prevents the discussion of amounts of methylmercury that egested immediately into the water by this fish, not being taken up or bioaccumulated.

Response: Our previous study showed that, contrary to inorganic mercury, the quantity of methylmercury leeched from the feed to the water was below detection levels, making water measurements irrelevant (Maulvault et al., 2016). In other words, although measurements in the water are important when working with inorganic mercury, methylmercury is a more strongly lipophilic and hydrophobic molecule. It preferentially adheres to sediment and accumulates in the tissues of animals (i.e. fish) via prey (Mason, 2001). Moreover, the quantity of food administered (1 % fish weight per fish) is standard for aquaculture and has been calculated so that remains are minimum. In the rare occasions food was not ingested, it was immediately siphoned together with fish faeces.

We added this information in the text: "Feed composition, manufacturing and MeHg spiking processes were executed as described by Maulvault et al. (2016). Fish were fed two to three times a day and total feed quantity provided per day was approximately 1% (standard calculation for aquaculture) of animal weight (at the end of 30 days, each fish was given approximately 0.0106 mg of HgT). Selected feed quantity also minimized food remains, which, in case of existing, were siphoned together with fish faeces after feeding." (Page 5, Line 5-12)

References Maulvault, A. L., Custodio, A., Anacleto, P., Repolho, T., Pousao, P., Nunes, M. L., Diniz, M., Rosa, R. and Marques, A.: Bioaccumulation and elimination of mercury in juvenile seabass (Dicentrarchuslabrax) in a warmer environment, Environ. Res.,

149, 77–85, doi:10.1016/j.envres.2016.04.035, 2016. Mason, R. P.: The Bioaccumulation of Mercury, Methylmercury and Other Toxic Elements into Pelagic and Benthic Organisms, in Coastal and Estuarine Risk Assessment, edited by N. Newman, M. Roberts jr, and R. Hale, pp. 127–149, Lewis Publishers, Boca Raton, Fl, USA., 2001.

Comment #8: Throughout - Given the commercial importance of the species, one surprising oversight is the fact that no discussion on different tissue burdens were made with respect to human consumption and climate change impacts. The only place this is alluded to is in the title! This is particularly relevant given the possibility that elevated CO2 reduces Hg accumulation possibly reducing transfer of hg into humans directly via consumption of muscle tissue, which could be an important result. This would provide some wider context in which to place the importance of this study generally, as well as contaminant/climate changes studies more generally. Response: We thank the referee for this thoughtful comment and have introduced considerations on this matter: "From a consumer perspective, our study showed that the counter-acting CO2 effect (hampering warming-stimulated Hg accumulation) was consistent in the muscle, the main tissue ingested by human population. Since this is the most relevant tissue for commercialization, such results constitute an important finding in the area of seafood safety, worthy of further research." (Page 11, Line 16-20) "Further knowledge on climate change and contamination impacts on fish ecophysiology (and biochemical stress-coping mechanisms) will help towards better comprehension of future fish stocks' health condition and tissue-dependent contaminant accumulation, consequently forecasting socio-ecological consequences in the oceans of tomorrow. Another pertinent knowledge gap that has been scarcely addressed is how oxidative stress and lipid peroxidation modify the nutritional value and general palatability of seafood, particularly fish. Thus, further multi-stressor studies on seafood safety and biochemical changes should be performed with the intent of helping stakeholders and regulatory authorities define future consumption recommendations and legislation." (Page 13, Line 8-15)

[Figure]

Technical corrections:

Technical correction #1: Page 1, Line 18-19 – Sentence beginning "Despite the more than likely co-occurrence..." is weak and doesn't read well. Needs stronger justification (see above)to enable stronger conviction in abstract, as well as explicitly highlight that contaminant/climate change stressor interactions are largely overlooked, rather than just "these stressors".

Response: We rephrased: "Future interactive effects between contaminants and climate change stressors are still largely unknown, even though such interactions will play a key role in shaping the ecophysiology of marine organisms." (Page 1, Lines 16-19)

Technical correction #2: Page 1, Line 29 – should read mechanisms not mechanism

Response: Changed.

Technical correction #3: Page 2, Line 2 (and throughout) – should be CO2 sub-scripted, this error occurs in a number of positions throughout manuscript, also sometimes is sub-scripted so inconsistent.

Response: Corrected.

Technical correction #4: Page 2, Line 4-5 –I would argue greenhouse gas effect is increasing global temperatures, and this is resulting in projected further increase (already increased by 0.76 °C from pre-industrial) in surface ocean temperature of . . . by end of the century.

Response: Changed to: "Moreover, conjointly with other "greenhouse" gases, increased CO₂ has triggered a continuous rise in mean ocean temperatures (nowadays increased by 0.76°C from pre-industrial values), and predictions point to a further 0.3-4.8 °C increase by the end of the century (IPCC, 2014)." (Page 2, Lines 2-6)

Technical correction #5: Page 2, Line 22 – Should read ". . .Sampaio et al., 2016) and

ultimately mortality (Coccini et al., 2000)."

Response: Changed.

Technical correction #6: Page 3, Line 7 – protein not proteins

Response: Changed.

Technical correction #7: Page 3, line 11 – responses not response

Response: Changed.

Technical correction #8: Page 3, Line 16 – remove the before estuaries

Response: Removed.

Technical correction #9: Page 4, Line 10 – should be pH controllers not controller

Response: In this case, although multiple pH probes were used, all were connected to a single pH controller, i.e. a Profilux system ($\pm$ 0.1, Profilux 3.1N, GHL). However, we have changed phrasing for the sake of clarity: "We used a Profilux system ($\pm$ 0.1, Profilux 3.1N, GHL) as pH controller, connected to each tank by individual pH probes." (Page 4, Lines 17-18)

Technical correction #10: Page 5, Line 5 – Length3 should be super-scripted

Response:Changed.

Technical correction #11: Page 5, Line 19 – remove with before nitric acid

Response: Removed.

Technical correction #12: Page 5, Line 23 – should be gill not gills

Response: Changed.

Technical correction #13: Page 5, Line 25 – remove posteriorly Response: Removed.

Technical correction #14: Page 5, Line 26 – rewrite as "..response concentrations,

quantified" removing were

Response: Changed.

Technical correction #15: Page 6, Line 18 – assume is potassium periodate not potassium per iodate

Response: Corrected.

Technical correction #16: Page 6, Line 23 (and page 7, line 16) – mg-2 needs super-scripting

Response: Done.

Technical correction #17: Page 6, Line 25 – insert space before Superoxide

Response: Done.

Technical correction #18: Page 7, Line 5 – is the % inhibition of SOD activity calculated as maximum inhibition, average inhibition at each 5 minute time point or from initial and final, just measured every 5 minutes over 25 minutes so potentially have different rates of inhibition and total overall inhibition over this time course

Response: It is the average inhibition from initial to final (25 minutes, 5 minute readings are used to create the slope).We included the information: "..., which allowed the assessment of inhibition percentage per minute (averaged from 25 minutes),..." (Page 7, Line 28-29)

Technical correction #19: Page 7, Line 23 – insert space before and

Response: Done.

Technical correction #20: Page 8, Line 2 – insert space in mg-1total

Response: Done.

Technical correction #21: Page 9, Line 19 (and other places) – A. regius needs italicising

Response: Corrected throughout the manuscript.

Technical correction #22: Page 10, Line 2 – notoriously is an odd choice of words, suggest just removing as reads fine without replacing

Response: Changed according to referee's suggestions.

T echnical correction #23: Page 10, Line 23 (and page 11, line 15) – H20 needs subscripting

Response: Done.

Technical correction #24: Page 18 – Why is the x-axis reversed on figure 1, d, compared to b and c. This confuses comparisons.

Response: Indeed, we apologize for the mistake and have corrected it. See new Figure 1 in the marked manuscript.

Please also note the supplement to this comment:
https://www.biogeosciences-discuss.net/bg-2017-147/bg-2017-147-AC1-supplement.pdf

**Supplement:**

**General comments**

*"This manuscript aims to address a topic of significant importance, namely the interaction between climate change stressors and contamination in coastal regions, and particularly its impact on species of commercial importance. This is certainly a topic of great interest and an area that has been identified as a significant knowledge gap in the field at present. Despite this potential and the undoubted requirement for such a study within the field, regrettably the manuscript presented here does not adequately address this question. As it stands there is insufficient detail presented throughout the methods to adequately appraise what has been done, there appear to be a number of methodological oversights that hamper the interpretation of the results and this has, to a large extent, led to many of the conclusions drawn not being supported by the data. Based on these factors I believe the manuscript at least requires major revisions to include this required detail, as well as restructure the conclusions to match what has actually been undertaken. It would then require re-review to appraise the manuscript in its new form. If it is not possible to include this required detail in full, in its current state the manuscript is not of sufficient quality to be published."*

**Response:** We thank the referee for his suggestions which have served to greatly improve the manuscript. We hope we have now provided sufficient detail on the methodologies employed in this work. We also hope to have clarified some misinterpretations throughout the text. Below, we reply to each comment in a point-by-point manner. Please note that Page and Line numbers now correspond to the marked up version of the manuscript.

**Specific comments**

*Comment #1:Abstract, discussion and conclusions – Throughout the manuscript the authors suggest the reduced accumulation of mercury in tissues under combined exposure is due to metabolic depression, and a subsequent reduced apatite/ingestion of food, initiated by elevated CO2. However, the authors do not measure any parameters in the current study that could confirm or counter this suggestion. There is no indication that these fish ingested less food so the conclusions, certainly as they are presented, are unfounded. Reduced accumulation could in fact be caused by a number of different mechanisms in the organism by which elevated COÂn2 augmented Hg accumulation, either by metabolic depression, reduced appetite (could be caused by alternative mechanism), reduced digestive efficiency, reduced uptake across the gut epithelium, greater egestion of Hg or impacts of Hg transport and complexation in plasma to reduce delivery to measured tissues. All are potentially feasible and at present insufficient information is known about this to surmise it is metabolic depression. It is vital to indicate that whilst altered accumulation is noted, which differs between specific tissues, the mechanism is not known. Following this point, the authors have not cited two key*

*references on ocean acidification and mercury contamination recently published (Li et al Scientific Reports 7;324 2017; and Wang et al ES&T 51:5820 2017). It is possible these were published after initial submission of the current manuscript, but in light of altered mercury accumulation under elevated CO2 these two manuscripts are key as they support the current finding.*

**Response:** The authors acknowledge that no additional parameters were measured to validate the conclusion that lower Hg accumulation under increased $CO_2$ was due to metabolic depression. Nonetheless, based on previous studies, there are reasons to believe this is the case. A wide range of organisms show metabolic decrease in response to increased extracellular acid–base stress(Kroeker et al., 2010), and especially to simultaneous occurrence of warming and acidification (Harley et al., 2006; Harvey et al., 2013; Rosa et al., 2013; Rosa and Seibel, 2008). Concerning $CO_2$, theoretically, the prioritization of acid–base regulation and ion regulatory enzyme machinery for $CO_2$ excretion (e.g. pyruvate kinase) may lead to lower metabolic activity in other enzymes (energy reallocation),as reported in other fish (Perry et al., 1988). As MeHg accumulation rates are positively correlated with metabolic rates (Dijkstra et al., 2013), these results would support the claim that acidification affects toxic compound accumulation rates (Schiedek et al., 2007).

Given our simultaneous exposure to both warming and acidification, which has been shown to undeniably suppress metabolic rates directly (Christensen et al., 2011; Harley et al., 2006; Rosa et al., 2013; Rosa and Seibel, 2008; see also Harvey et al., 2013 and Kroeker et al, 2010) we still hold the conviction that metabolic processes may be at play. However, the authors also acknowledge that they were not aware of the recent research pointed out by the referee, which thoroughly picks apart the causes of behind these mechanisms. We thank the referee for this useful comment and we have altered our interpretation, changing the text:

"However, such effect may be offset by $CO_2$-linked decreases in mercury accumulation (Sampaio et al., 2016; Schiedek et al., 2007; Wang et al., 2017)" (Page 2, Lines 29-31)

"Instead, our results support recent studies demonstrating that hypercapnia dampens Hg accumulation in marine organisms (Li et al., 2017; Sampaio et al., 2016; Wang et al., 2017). There are several possible reasons which may underpin such an interaction, encompassing digestive (reduced digestive efficiency, reduced uptake through the gut membrane, reduced appetite, increased Hg depuration) and molecular (competition between Hg and $H^+$ ions for binding sites, impacts on Hg plasma transport, lower phospholipidic membrane permeability) mechanisms (Li et al., 2017). A recent study has also found that the lysosome-autophagy pathway was up-regulated by combined exposure to Hg and increased $CO_2$, enabling better animal fitness which may potentially reduce Hg accumulation and toxicity (Wang et al., 2017). In addition, taking into account that the occurrence of both warming and acidification changes physiological thresholds (Christensen et al., 2011; Harley et al., 2006; Rosa et al., 2013; Rosa and Seibel, 2008), a degree of metabolic depression may also play a role on decreasing HgT accumulation (Dijkstra et al., 2013; Sampaio et al., 2016)." (Page 11, Lines 4-16)

"In general, warming conditions enhanced MeHg accumulation but $CO_2$-linked impacts countered this effect." (Page 12, Lines 28-29)

***Comment #2:*** *Abstract, discussion and conclusions – Similarly to the point above, the authors repeatedly suggest elevated H+ impacts mercury accumulation/toxicity at a molecular level, but no acid base measures*

*were made. Also it is a common misunderstanding that elevated CO2 results in chronic acidosis in fish plasma, this is not the case. Elevated CO2 results in acute acidosis which is rapidly compensated for by an elevation in bicarbonate, returning the plasma H+ to normal levels. Therefore the suggestion that elevated H+ impacts on mercury toxicity/accumulation is not supported, especially as acid-base parameters are not presented that counter this common response noted in acid-base compensating species such as fish. The authors need to again re-interpret data and re-write conclusions to better reflect the demonstrated results and not make broad unsupported conclusions, pinned loosely on previously published literature that has been misinterpreted/misunderstood.*

**Response:** We would like to point out that we never said that acidosis was present in a long-term perspective, nor did we assume that fish are not able to acid-base compensate, a mechanism that is already extensively described (Brauner and Baker, 2009; Heuer and Grosell, 2014; Michaelidis et al., 2007; among many more). In fact, besides some logistical and time constrains, that was the main reason why no acid-base measurements were performed. Having said that, acid-base compensation occurs mainly by increasing bicarbonate ($HCO_3^-$) levels in both blood and cellular, which in turn leads to a normalization of intracellular and extracellular pH (Heuer and Grosell, 2014; Michaelidis et al., 2007). The chemical equation that underpins this reaction is as follows:

$$CO_2 + H_2O \rightleftharpoons H_2CO_3 \rightleftharpoons H^+ + HCO_3^-$$

Thus, despite pH being normalized by balancing the ratio between $H^+ + HCO_3^-$ and $H_2CO_3$ (it generally stabilizes at ~0.05/0.1 units lower than in normocapic conditions), it is important to note that $H^+$ levels in the organism are still increased relatively to basal levels, especially in long-term acclimations to hypercapnia- **where there is a constant influx of $H^+$ ions** (Heuer and Grosell, 2014; Michaelidis et al., 2007). Moreover, due to cell prioritization, intracellular and extracellular pH often display significantly different values: the former is up-regulated to normocapnic levels or higher, while the latter generally stabilizes at lower pH (ΔpHcan reach~0.3-0.7) (Brauner and Baker, 2009; Heuer and Grosell, 2014). This also partially contributes to increased $H^+$ levels. As our reasoning is grounded on molecular interactions (both oxidative stress-inducing and ROS-mitigating) of increased $H^+$ chemical reactions (see also Dean, 2010), it does not imply for fish acid-base compensation to fail.

In light of the new recent studies mentioned by the referee, we have introduced some new considerations to our Abstract/Discussion/Conclusion. The reason we do not believe that the lysozyme-autophagy pathway (Wang et al., 2017) is solely responsible for the antagonistic relationship between stressors is that it does not account for hypercapnia-induced oxidative stress and chaperone activation. Within this context, we have rephrased our interpretations and changed the text accordingly:

In the Abstract:

"Together with $CO_2$-promoted removal of damaged proteins and enzymes, we argue that simultaneous increase in hydrogen ($H^+$) and reactive oxygen species (e.g. $O_2^-$) radicals is partially compensated through chemical reaction equilibrium balancing." (Page 1, Lines 26-29)

In the Discussion:

"Increased $CO_2$ (co-occuring with Hg contamination) may elicit the up-regulation of the lysosome-autophagy pathway, which is responsible for removing damaged proteins and organelles, effectively reducing oxidative stress (Wang et al., 2017). This mechanism may contribute to alleviate not only Hg induced stress, but also warming-related oxidative stress. We also argue that this antagonistic relation can be partially explained by a $CO_2$-related increase of $H^+$ ion concentrations in the blood and cellular surroundings, counterbalanced by bicarbonate increase (acid-base compensation) to normalize pH levels (Heuer and Grosell, 2014; Michaelidis et al., 2007). By itself, the presence of excessive $H^+$ ions activates free radical neutralizing defenses (Tiedke et al., 2013), which is in line with the present findings when hypercapnia was the sole stressor. However the production of $O_2^-$ and further complementary ROS radicals (e.g. $OH^-$) by other stressors may result in facilitated $H_2O$ and $H_2O_2$ formation, due to chemical reactions balancing equilibrium (e.g. $H^+ + OH^- \rightleftharpoons H_2O$), thus eliminating free radicals and decreasing activity of antioxidant enzymes to basal standards." (Page 11/12, Lines 31/1-10)

"More so than for oxidative stress, the enhanced removal of damaged proteins and enzymes indirectly promoted by increased $CO_2$ (via up-regulated lysosome-autophagy) may have especially contributed to subside protein chaperone production. Given that Hsp70 production can also be stimulated by high ionic (e.g. $H^+$) concentrations (Feder and Hofmann, 1999), we reason that the same additional mechanism by which hypercapnia potentially modulates oxidative stress can be applied for heat shock response" (Page 12, Lines 18-22)

In the Conclusions:

"In fact, despite negative effects prompted as a sole stressor, acidification consistently elicited antagonistic responses to temperature and contamination effects on oxidative stress (including heat shock response), which may be explained by stimulated removal of damaged proteins and organelles (Wang et al., 2017). Moreover, we also argue that the mechanistic interactions found are coadjuvanted by the coinciding increase of hydrogen ($H^+$) and radical reactive oxygen species (e.g. $O_2^-$, $OH^-$), which subsequently nullify each other due to the spontaneous equilibrium of chemical reactions (e.g. $H^+ + OH^- \rightleftharpoons H_2O$)." (Page 12/13, Lines 29/1-6)

***Comment #6:*** *Methods (page 4 lines 1-7) - The description of the conditions, and particularly their maintenance is not sufficient. It states ammonia, nitrate and nitrite were regularly monitored and kept within recommended levels. How was this tested, what were the accepted levels and what were the levels measured within the experiment? Also how were high levels mitigated against and how often? Furthermore, it mentions salinity was kept at 35.0 ± 1.0 g/l NaCl? The probe listed is a conductivity probe so does not measure in g/l of NaCl but gives a conductivity measure or salinity as a psu. Also how was salinity maintained? i.e. is this addition of deionised water to compensate for evaporation? Or addition of additional NaCl? The description is confusing, and could be interpreted as additional NaCL addition. Any further addition of NaCl would significantly alter osmolality thus this needs clarifying to explain if input water fluctuated in salinity. A better description of this process is therefore required.*

**Response:** We apologize for not having provided more detail on these matters, but we have been said to be overzealous with these descriptions in recent publications. However, it is our pleasure to fill the gaps the referee points out in this section.

Ammonia ($NH_3/NH_4^+$), nitrite ($NO_2^-$) and nitrate ($NO_3^-$) concentrations were daily checked (Colorimetric kits, Aquamerk, Germany), and kept below detectable levels (i.e. $NH_3/NH_4^+ < 0.25$ mg $l^{-1}$; $NO_2^- < 0.10$ mg $l^{-1}$; $NO_3^- < 0.2$ mg $l^{-1}$).

Salinity was not measured through a conductivity probe, we apologize for the omission. We opted instead for a refractometer (V2, TMC Iberia, Portugal) and took daily measurements as with temperature and pH. Salinity was also incorporated in the calculation of seawater carbonate chemistry (Table S1). We acknowledge the mistake (g/l) and have removed salinity units (see below).

The addition of deionised water or any kind of water except sea water would modify carbonate chemistry and render our pH manipulation useless (Cornwall and Hurd, 2015). All potential fluctuations in both these parameters were solved by the seawater flux, and in the case of nutrients, by the biological filter described (Page 4, Lines 2-5). As detailed in the Methods section (Page 4, Lines 5-9), each experimental unit (or recirculatory aquatic system, RAS) was a semi-closed system with a constant seawater flux (complete turnover rate in 24h) precisely to maintain parameters such as salinity and nutrients. Thus, the mitigation of

potential problems was done *a priori* and no additional action was needed during the course of the experiment.

We have added the pertinent information in the text:

"To prevent fluctuations in environmental parameters, each RAS worked as a semi-closed system, with constant low flow external water input (flux > 2 l h$^{-1}$; 50 l tank turnover rate = 24 h). Consequently, ammonia ($NH_3/NH_4^+$), nitrite ($NO_2^-$) and nitrate ($NO_3^-$) concentrations were daily checked (Colourimetric kits, Aquamerk, Germany), and kept below detectable levels (i.e. $NH_3/NH_4^+ < 0.25$ mg l$^{-1}$; $NO_2^- < 0.10$ mg l$^{-1}$; $NO_3^- < 0.20$ mg l$^{-1}$), and salinity was kept at $35.0 \pm 1.0$ (V2 Refractometer, TMC Iberia, Portugal). Temperature and pH (multiparametric probe, Multi3420 SET G, WTW) were measured daily, directly in the holding tanks. Photoperiod was fixed at 12 h light : 12 h dark." (Page 4, Lines 5-14).

*Comment #8: Throughout - Given the commercial importance of the species, one surprising oversight is the fact that no discussion on different tissue burdens were made with respect to human consumption and climate change impacts. The only place this is alluded to is in the title! This is particularly relevant given the possibility that elevated CO2 reduces Hg accumulation possibly reducing transfer of hg into humans directly via consumption of muscle tissue, which could be an important result. This would provide some wider context in which to place the importance of this study generally, as well as contaminant/climate changes studies more generally.*

**Response:** We thank the referee for this thoughtful comment and have introduced considerations on this matter:

"From a consumer perspective, our study showed that the counter-acting $CO_2$ effect (hampering warming-stimulated Hg accumulation) was consistent in the muscle, the main tissue ingested by human population. Since this is the most relevant tissue for commercialization, such results constitute an important finding in the area of seafood safety, worthy of further research." (Page 11, Line 16-20)

"Further knowledge on climate change and contamination impacts on fish ecophysiology (and biochemical stress-coping mechanisms) will help towards better comprehension of future fish stocks' health condition and tissue-dependent contaminant accumulation, consequently forecasting socio-ecological consequences in the oceans of tomorrow. Another pertinent knowledge gap that has been scarcely addressed is how oxidative stress and lipid peroxidation modify the nutritional value and general palatability of seafood, particularly fish. Thus, further multi-stressor studies on seafood safety and biochemical changes should be performed with the intent of helping stakeholders and regulatory authorities define future consumption recommendations and legislation." (Page 13, Line 8-15)

**Technical corrections:**

***Technical correction #1:*** *Page 1, Line 18-19 – Sentence beginning "Despite the more than likely co-occurrence..." is weak and doesn't read well. Needs stronger justification (see above)to enable stronger conviction in abstract, as well as explicitly highlight that contaminant/climate change stressor interactions are largely overlooked, rather than just "these stressors".*

**Response:** We rephrased:

"Future interactive effects between contaminants and climate change stressors are still largely unknown, even though such interactions will play a key role in shaping the ecophysiology of marine organisms." (Page 1, Lines 16-19)

***Technical correction #2:*** *Page 1, Line 29 – should read mechanisms not mechanism*

**Response:** Changed.

***Technical correction #3:*** *Page 2, Line 2 (and throughout) – should be CO2 sub-scripted, this error occurs in a number of positions throughout manuscript, also sometimes is sub-scripted so inconsistent.*

**Response:** Corrected.

***Technical correction #4:*** *Page 2, Line 4-5 –I would argue greenhouse gas effect is increasing global temperatures, and this is resulting in projected further increase (already increased by 0.76 ºC from pre-industrial) in surface ocean temperature of . . . by end of the century.*

**Response:** Changed to:

"Moreover, conjointly with other "greenhouse" gases, increased $CO_2$ has triggered a continuous rise in mean ocean temperatures (nowadays increased by 0.76ºC from pre-industrial values), and predictions point to a further 0.3-4.8 °C increase by the end of the century (IPCC, 2014)." (Page 2, Lines 2-6)

***Technical correction #5:*** *Page 2, Line 22 – Should read ". . .Sampaio et al., 2016) and ultimately mortality (Coccini et al., 2000)."*

**Response:** Changed.

***Technical correction #6:*** *Page 3, Line 7 – protein not proteins*

**Response:** Changed.

***Technical correction #7:*** *Page 3, line 11 – responses not response*

**Response:** Changed.

***Technical correction #8:*** *Page 3, Line 16 – remove the before estuaries*

**Response:** Removed.

***Technical correction #9:*** *Page 4, Line 10 – should be pH controllers not controller*

**Response:** In this case, although multiple pH probes were used, all were connected to a single pH controller, i.e. a Profilux system (± 0.1, Profilux 3.1N, GHL). However, we have changed phrasing for the sake of clarity:

"We used a Profilux system (± 0.1, Profilux 3.1N, GHL) as pH controller, connected to each tank by individual pH probes." (Page 4, Lines 17-18)

***Technical correction #10:*** *Page 5, Line 5 – Length3 should be super-scripted*

**Response:** Changed.

***Technical correction #11:*** *Page 5, Line 19 – remove with before nitric acid*

**Response:** Removed.

***Technical correction #12:*** *Page 5, Line 23 – should be gill not gills*

**Response:** Changed.

***Technical correction #13:*** *Page 5, Line 25 – remove posteriorly*

**Response:** Removed.

*Technical correction #14: Page 5, Line 26 – rewrite as "..response concentrations, quantified" removing were*

**Response:** Changed.

*Technical correction #15: Page 6, Line 18 – assume is potassium periodate not potassium per iodate*

**Response:** Corrected.

*Technical correction #16: Page 6, Line 23 (and page 7, line 16) – mg-2 needs super-scripting*

**Response:** Done.

*Technical correction #17: Page 6, Line 25 – insert space before Superoxide*

**Response:** Done.

*Technical correction #18: Page 7, Line 5 – is the % inhibition of SOD activity calculated as maximum inhibition, average inhibition at each 5 minute time point or from initial and final, just measured every 5 minutes over 25 minutes so potentially have different rates of inhibition and total overall inhibition over this time course*

**Response:** It is the average inhibition from initial to final (25 minutes, 5 minute readings are used to create the slope).We included the information:

"…, which allowed the assessment of inhibition percentage per minute (averaged from 25 minutes),…" (Page 7, Line 28-29)

*Technical correction #19: Page 7, Line 23 – insert space before and*

**Response:** Done.

*Technical correction #20: Page 8, Line 2 – insert space in mg-1total*

**Response:** Done.

***Technical correction #21:*** *Page 9, Line 19 (and other places) – A. regius needs italicising*

**Response:** Corrected throughout the manuscript.

***Technical correction #22:*** *Page 10, Line 2 – notoriously is an odd choice of words, suggest just removing as reads fine without replacing*

**Response:** Changed according to referee's suggestions.

***Technical correction #23:*** *Page 10, Line 23 (and page 11, line 15) – H20 needs subscripting*

**Response:** Done.

***Technical correction #24:*** *Page 18 – Why is the x-axis reversed on figure 1, d, compared to b and c. This confuses comparisons.*

**Response:** Indeed, we apologize for the mistake and have corrected it. See new Figure 1 in the marked manuscript.

**Ocean acidification dampens warming and contamination effects on the physiological stress response of a commercially important fish**

[revised manuscript text omitted]

increases metal availability (Wiener et al., 1990) and toxicity (Han et al., 2014). However,  such effect may be offset by $CO_2$-linked  decrease in mercury accumulation (Sampaio et al., 2016; Schiedek et al., 2007; Wang et al., 2017). Under environmental stressor exposure, a general deleterious biochemical pathway triggered is the formation of oxygen reactive species (ROS) in the organism's cells.

Although there is some proof linking ROS production to hypercapnic scenarios (Pimentel et al., 2015), such is particularly true for increased temperature and mercury contamination (Berntssen et al., 2003; Portner, 2002). Increasing ROS concentrations cause protein damage and lipid peroxidation, i.e. oxidative stress, cascading in augmented malondialdehyde content (MDA), one of the final products of lipid peroxidation (Lesser, 2006). As a physiological defense response, ROS production elicits antioxidant activity in the organism. Specifically, a battery of enzymes are activated to eliminate ROS and prevent MDA build-up: superoxide dismutase (SOD), which converts superoxide ($O_2^-$) into hydrogen peroxide ($H_2O_2$); catalase (CAT) which converts $H_2O_2$ into water ($H_2O$) and oxygen ($O_2$); and glutathione S-transferase (GST), which is involved in the protection against xenobiotics and linked to antioxidant defense (Lesser, 2006; Wang et al., 2000). Moreover, tissue-specific heat shock protein (Hsp70) production are also correlated with thermal stress, i.e. high temperatures (Repolho et al., 2014; Rosa et al., 2012, 2014a) and metal contamination (Rajeshkumar and Munuswamy, 2011; Williams et al., 1996). Heat shock proteins help repair, refold and eliminate damaged or denatured proteins, as well as protect and control ROS formation (Sokolova et al., 2011). Given their wide scope, these constituents of the antioxidant enzymatic and protein chaperone machineries are widely used as biomarkers in ecotoxicology to assess fish physiological stress response (e.g. Anacleto et al., 2014; Fonseca et al., 2011; Rosa et al., 2014b).

Despite the inevitability of marine organisms having to cope with simultaneous effects of ocean warming, acidification and persistent contamination (MeHg), no studies have focused on how the interactive effects between these three stressors will challenge  fish ecophysiology. Due to its coastal distribution, the meagre *(Argyrosomus regius)* is particularly susceptible to MeHg accumulation, especially when they migrate towards  estuaries to spawn (Durrieu et al., 2005). Understanding how this commercially important species will deal with the predicted climate change scenarios may provide valuable information on future stock population conditions and potential impacts on coastal food-webs. Within this context, here we performed a 30-day acclimation experiment to investigate organ-dependent Hg accumulation (gills, liver and muscle) under a warming ($\Delta T = 4$ ºC) and acidification ($\Delta CO_2 = 1100$ µatm) context, as well as the respective phenotypic responses of molecular chaperone (Hsp70) and antioxidant enzymatic (SOD, CAT and GST) machineries, in commercially important fish (*A. regius*). The direct consequences at organism (survival rates and condition index) and cellular (lipid peroxidation, MDA) levels were also evaluated.

**2 Material and Methods**

**2.1 Experimental setup and incubation**

Juvenile *Argyrosomus regius* (n $\simeq$ 100; Fig. 5) (mean $\pm$ SD; total weight: 4.26 $\pm$ 2.8 g; total length: 6.30 $\pm$ 1.2 cm) from EPPO - IPMA (Estação Piloto de Piscicultura de Olhão – Instituto Português do Mar e da Atmosfera, Portugal) where fish were maintained under standard summer season environmental parameters (pH = 8.0 and 19 ºC). In August 2014, fish were transported to the facilities of Laboratório Marítimo da Guia (LMG, MARE, Faculdade de Ciências, Universidade de Lisboa) in August 2014. Fish were randomly placed in twenty-four 50l tanks (n = 3-4 per tank) with individual recirculating aquaculture systems (RAS) equipped with glass wool (physical filtration), bio-balls (Fernando Ribeiro Lda) and protein skimmers (biological filtration, ReefSkimPro 850, TMC Iberia), as well as additional UV disinfection (Vecton 120, TMC Iberia) to maintain superior water quality. Natural seawater was pumped directly from the ocean into an 8 m$^3$ storage tank, and subsequently filtered (0.35 µm filters, Fernando Ribeiro Lda) and UV-sterilized (Vecton600, TMC Iberia), before pumping into mixing (n = 24) and respective experimental (n = 24, 50 l) tanks/RAS. To prevent fluctuations in environmental parameters, eEach RAS worked as a semi-closed system, with constant low flow external water input (flux $\geq$ 2 l h$^{-1}$; 50 l tank turnover rate = 24 h). Consequently, ammonia (NH$_3$/NH$_4^+$), nitrite (NO$_2^-$) and nitrate (NO$_3^-$) concentrations were daily checked (Colourimetric kits, Aquamerk, Germany), and kept below detectable levels (i.e. NH$_3$/NH$_4^+$ < 0.25 mg l$^{-1}$; NO$_2^-$ < 0.10 mg l$^{-1}$; NO$_3^-$ < 0.20 mg l$^{-1}$), and Ammonia, nitrate and nitrite were regularly monitored and kept within recommended levels (Aquamerk). sSalinity was kept at 35.0 $\pm$ 1.0 (V2 Refractometer, TMC Iberia, Portugal). g l$^{-1}$ and thephotoperiod was fixed to 12 h light: 12 h dark. Temperature , salinity and pH (multiparametric probe, Multi3420 SET G, WTW) were daily measured daily, directly in the holding tanks. Photoperiod was fixed at 12 h light : 12 h dark.

As per experimental conditions, temperature in the tanks was down-regulated using chillers ($\pm$ 0.1 ºC, Frimar, Fernando Ribeiro Lda), and up-regulated by submerged 200 W heaters (V2Therm, TMC Iberia). Seawater carbonate chemistry was altered through CO$_2$-enriched air input, with pH (8.0 and 7.5) used as proxy measurement. As pH controller, Wwe used a Profilux system ($\pm$ 0.1, Profilux 3.1N, GHL) as pH controller, connected to each tank by individual pH probes. Within each RAS, pH was down-regulated by injection of the certified CO$_2$-enriched air (Air Liquide), and up-regulated by injection of atmospheric air. Seawater carbonate system speciation (Table S1) was calculated once every week from pH$_{total\ scale}$ (pH$_T$ ) and total alkalinitythe latter (wavelength = 595 nm,)using a base neutralization by formic acid and a pH sensitive dye (bromophenol blue), following Sarazin et al. (1999). pH$_T$ was quantified via a Metrohm pH meter (826 pH mobile, Metrohm, Filderstadt, Germany) connected to a glass electrode (Schott IoLine, SI analytics, $\pm$ 0.001) and calibrated against TRIS–HCl (TRIS) and 2-aminopyridine-HCl (AMP; Mare, Liège, Belgium) seawater buffers (Dickson et al., 2007). Total alkalinity was measured spectrophotometrically (wavelength = 595 nm; UV-1800 Shimadzu, Japan) through base neutralization by formic acid and a pH sensitive dye (bromophenol blue), following Sarazin et al. (1999). Total dissolved

inorganic carbon ($C_T$), $pCO_2$ and aragonite saturation were calculated using CO2SYS software (Lewis and Wallace, 1998), with dissociation constants from Mehrbach et al. (1973) as refitted by Dickson and Millero (1987). The non-contaminated and contaminated fish were fed similar diets, differing only on MeHg content. Contaminated diet was fortified with MeHg (inserted in the form of MeHg(II) chloride, CH$_3$ClHg, 99.8 %, Sigma-Aldrich, solubilized previously in ethanol). Given our dietary option, ecologically relevant MeHg concentrations ere chosen based on levels (low contamination, ~0.12 mg kg$^{-1}$ wet weight (ww); and high contamination, ~1.6 mg kg$^{-1}$ ww found in common *A. regius* prey species from contaminated coastal areas (Cardoso et al., 2014; Kannan et al., 1998; Nunes et al., 2008). The pellets given to fish allocated to non-contaminated and contaminated treatments had approximately 0.60 ± 0.01 mg kg$^{-1}$ dry weight (dw) and 8.02 ± 0.01 mg kg$^{-1}$  of MeHg, respectively, in the contaminated treatment which were considered to mimick the concentrations found in the field (see Maulvault et al., 2016, 2017). Feed composition, manufacturing and MeHg spiking processes were executed as described by (Maulvault et al.,(2016).  Fish were fed two to three times a day and total food quantity provided per day was approximately 1% (standard calculation for aquaculture) of animal weight Selected feed quantity also minimized food remains, which, in case of existing, were siphoned together with fish faeces  after feeding.

[revised manuscript text omitted]
 physiology  at sublethal levels, i.e. zero mortality and also no effects on Fulton condition wer registered.

25  The fact that the meagre (*A. regius*) is a very resilient species and easily adapts to environmental alterations (Monfort, 2010) may explain the absence of deleterious effects at an organism level,after 30 days of exposure .

Affinity for metal accumulation varied between fish tissues with increasing Hg accumulation as follows: muscle < gills < liver. These results are supported by previous reports on mercury tissue preferential accumulation. The muscle is an organ

30 tissue generally characterized for its low metal affinity (Jezierska and Witeska, 2006) compared to the liver, where

metals accumulate at higher levels, due to its key role in metal accumulation and detoxification (Gbem et al., 2001; Wagner and Boman, 2003). Furthermore, as a result of increased blood supply, gills are organs likewise known to possess higher Hg affinity than the muscle (Jezierska and Witeska, 2006; Vergilio et al., 2012).

**4.2 Environmental influence on mercury accumulation**

Mercury accumulation in fish is known to depend on the water physicochemical properties (e.g. temperature, pH, alkalinity) (Harris and Bodaly, 1998; Ponce and Bloom, 1991; Wren et al., 1991). Indeed, we also showed a consistent increase in Hg accumulation under the warming scenario. However, when both temperature and $CO_2$ stressors were present, Hg accumulation was  decreased. Temperature increases Hg bioaccumulation in fish due to enhanced metabolism and consequent higher intake of MeHg-contaminated prey (Dijkstra et al., 2013; MacLeod and Pessah, 1973). Despite previous evidence that lowered pH (< 7.0 units) increases Hg accumulation in freshwater fish (Haines et al., 1992; Ponce and Bloom, 1991), the current findings do not reflect this pattern, arguably due to the magnitude of pH decrease (here we used pH 7.5). Instead, our results support recent studies  demonstrating that  hypercapnia dampens Hg accumulation in marine organisms (Li et al., 2017; Sampaio et al., 2016; Wang et al., 2017). There are several possible reasons which may underpin such an interaction, encompassing digestive (reduced digestive efficiency, reduced uptake through the gut membrane, reduced appetite, increased Hg depuration) and molecular (competition between Hg and $H^+$ ions for binding sites, impacts on Hg plasma transport, lower phospholipidic membrane permeability) mechanisms (Li et al., 2017). A recent study has also found that the lysosome-autophagy pathway was up-regulated by combined exposure to Hg and increased $CO_2$, enabling better animal fitness which may potentially reduceHg accumulation and toxicity (Wang et al., 2017). In addition, . Thus, taking  into account that the occurrence of both  warming and acidification  changes physiological  thresholds (Christensen et al., 2011; Harley et al., 2006; Rosa et al., 2013; Rosa and Seibel, 2008),  a  degree of metabolic depression  may also played  role on decreasing HgT  accumulation (Dijkstra et al., 2013; Sampaio et al., 2016). From a consumer perspective, our study showed that the counteracting $CO_2$ effect (hampering warming-stimulated Hg accumulation) was consistent in the muscle, the main tissue ingested by human population. Since thisis the most relevant tissue for commercialization, such results constitute an important finding in the area of seafood safety, worthy of further research.

**4.3 Oxidative stress under a multi-stressor environment**

Exposure to MeHg contamination, ocean warming and acidification potentiated significant changes in meagre physiology. As expected, lipid peroxidation and consequent MDA build-up was higher under MeHg contamination (Berntssen et al., 2003; Vieira et al., 2009). The fact that contamination and warming per se elicited only small MDA build-up, is likely due to the fact that *A. regius* is a highly resilient estuarine species, i.e. great tolerance to environmental stressors (Monfort, 2010). Moreover, to cope with oxidative stress, *A. regius* displayed enhanced CAT, SOD and GST activities under contaminated and warming scenarios, which is in line with previous studies reporting an enhanced anti oxidative stress response in fish (Maulvault et al., 2017; Pimentel et al., 2015; Vieira et al., 2009). While it is worth mentioning that increased $CO_2$ played a minor role in CAT activity (non-significant, p = 0.116), regarding the other enzymes, hypercapnia as a sole stressor significantly augmented antioxidant activity. However when combined with other stressors, elevated $CO_2$ antagonized the co-ocurring stressor's effect (i.e. contamination and/or warming). Increased $CO_2$ (co-occuring with Hg contamination) may elicit the up-regulation of the lysosome-autophagy pathway, which is responsible for removing damaged proteins and organelles, effectively reducing oxidative stress (Wang et al., 2017). This potential mechanism may contribute to alleviate not only Hg induced stress, but also warming-related oxidative stress. We also argue that  this antagonistic relation can be partially explained by a $CO_2$-related increase of $H^+$ ion concentrations in the blood and cellular surroundings, counterbalanced by bicarbonate increase (acid-base compensation) to normalize pH levels (Heuer and Grosell, 2014; Michaelidis et al., 2007). By itself, the presence of excessive $H^+$ ions activates free radical neutralizing defenses (Tiedke et al., 2013), which is in line with the present findings when hypercapnia was the sole stressor. However, the production of $O_2^-$ and further complementary ROS free radicals (e.g. $OH^-$) by other stressors may result in facilitated $H_2O$ and $H_2O_2$ formation, due to chemical reactions balancing equilibrium (e.g. $H^+ + OH^- \rightleftharpoons H_2O$), thus eliminating free radicals and decreasing activity of antioxidant enzymes to basal standards.

**4.4 Protein chaperone functioning under a multi-stressor environment**

Hsp70 response was tissue-dependent, showing a pattern similar to HgT tissue preferential accumulation (see first section). Higher liver expression is not unexpected given the fact that this organ plays a key role in metal accumulation and detoxification (Gbem et al., 2001; Wagner and Boman, 2003). More importantly, as observed in antioxidant stress enzymatic machinery, hypercapnia revealed the same antagonistic relationship with other stressor's effects: increased $CO_2$ down-regulated heat shock response in the livers of contaminated fish and in the muscle of fish under warming. As such, this study confirms that Hsp70 expression is closely correlated with other forms of antioxidant response, such as CAT, SOD and GST (Iwama et al., 1998; Rosa et al., 2012, 2014a). More so than for oxidative stress, the enhanced removal of damaged proteins and enzymes indirectly promoted by increased $CO_2$ (via up-regulated lysosome-autophagy) may have especially contributed to subside protein chaperone production. Given that Hsp70 production can also be stimulated by  high

ionic (e.g. $H^+$) concentrations (Feder and Hofmann, 1999), we reason that the same additional mechanism by which hypercapnia potentially modulates oxidative stress can be applied for heat shock response . Enhanced $CO_2$ leads to increased $H^+$ concentration triggering physiological stress responses, while the facilitated conversion of free ions and radicals ($H^+$ and O-associated molecules) into $H_2O$ and $H_2O_2$ leads to reduced stress input by warming, contamination (and hypercapnia itself).

**5 Conclusions**

In this study, we observed that sublethal MeHg contamination is organ selective (accumulating to higher levels in the liver) and found that future abiotic conditions modulate its accumulation throughout the organism. In general, warming conditions enhanced MeHg accumulation but $CO_2$-linked impacts countered this effect. In fact, despite negative effects prompted as a sole stressor, acidification consistently elicited antagonistic responses to temperature and contamination effects on oxidative stress (including  heat shock response), which may be explained by stimulated removal of damaged proteins and organelles (Wang et al., 2017). Moreover, we also argue that the mechanistic interactions found are underpinned by the coinciding increase of  hydrogen ($H^+$) and radical reactive oxygen species (e.g. $O_2^-$, $OH^-$), which subsequently nullify each other due to the spontaneous equilibrium of chemical reactions (e.g. $H^+ + OH^- \rightleftharpoons H_2O$).

In the future, it is important to deepen our understanding on this mechanism and evaluate if this antagonistic relationship is conservative throughout other less-resilient species (e.g. non-estuarine ones). Further knowledge on climate change and contamination impacts on fish ecophysiology (and biochemical stress-coping mechanisms) will help towards better comprehension of future fish stocks' health condition and tissue-dependent contaminant accumulation, consequently forecast socio-ecological consequences in the oceans of tomorrow. Another pertinent knowledge gap that has been scarcely addressed is how oxidative stress and lipid peroxidation modify the nutritional value and general palatability of seafood, particularly fish. Thus, further multi-stressor studies on seafood safety and biochemical changes should be performed with the intent of helping stakeholders and regulatory authorities define future consumption recommendations and legislation.

**6 Code availability**

R code used in the analysis is available as Supplemental material.

**7 Data availability**

The full dataset is made available as Supplemental material.

**8 Competing interests**

The authors declare no conflict of interest.

**9 Author contributions**

ES, ARL, SF, AM, PP and RR designed the study. JRP, MP, TR and TFG assisted during the experiment and sampling. AL and SF quantified HgT accumulation. ARL, SF, MP and JRP quantified the enzymes. ES, TR, TFG and RR performed the statistical analysis. ES and ARL wrote the paper, for which all authors contributed with discussion and earlier drafts.

**Acknowledgments**

We thank Kenneth Storey for the helpful discussion and IPMA-Olhão for providing juvenile meagre specimens for the trials. This work was supported by Fundação para a Ciência e Tecnologia (FCT): PhD (ARL, SFRH/BD/97070/2013;JRP, SFRH/BD/111153/2015; ALM, SFRH/BD/103569/2014) and post-doctoral (TFG, SFRH/BPD/98590/2013; TR, SFRH/BPD/98590/2013) scholarships, as well as RR and AM in the framework of the IF 2013 and IF2014 programs.

[revised manuscript text omitted]

---

## Author Comment (AC2) · 4 Jul 2017

Referee #2 (Manoela Orte, PhD)

General comments

"The interactive effects of acidification, warming and the presence of the metal Hg was assessed in the Fish Argyrosomus regius. Bioaccumulation of Hg was measured in different organs of the fish and sublethal toxic responses were also analyzed by the use of biomarkers. The topic is highly relevant since research regarding global change issues

should preferably focus on a multi-stressors approach. Furthermore, mercury is an important persistent contaminant found in coastal environments around the world and information regarding its interactive toxicological effects with other parameters such as acidification and warming are of great value. In general, the writing is clear and the data obtained is interesting. However, some issues regarding the methodological approach used are not well explained and there are some information at the results and discussion section that should be included. Therefore I recommend that the authors perform the suggested corrections before the article is published."

Response: We thank the referee for her comments and suggested terminology which helped to contextualize our manuscript better and improve the overall scientific outcomes found. We have addressed the lack of methodological procedures and hope that we have reached the publication standards upheld by the referee. We have accepted most of the referee's suggestions and, below, discuss each comment in a point-by-point manner. Please note that Page and Line numbers now correspond to the marked up version of the manuscript.

Specific comments

Comment #1: "The focus of the study is the evaluation of toxic responses of the metal Hg in a global change scenario. It is mentioned that concentration of Hg was chosen according to environmental measurements, however data on the range of toxic concentrations of this metal to this species or other fish species is not included. Considering that the article uses an ecotoxicological approach and therefore it is based on dose-response concentration it is crucial that more details on this subject is included, such as values of toxicity for fishes and environmental values within contaminated and non contaminated areas, especially in the area where the study was conducted."

Response: Mercury concentrations chosen for this study were based on levels of contamination found in contaminated coastal areas (specifically the extensively studied contaminated estuary of Aveiro, Portugal) for species that are natural prey ofthe meagre (e.g. Cardoso et al., 2014; Nunes et al., 2008). Not exclusive to the Eastern Atlantic coast, these mercury concentrations can also be found in other areas globally (e.g. Kannan et al., 1998). We thank the reviewer for pointing out the need for contextualization and have added: "Given our dietary option, ecologically relevant MeHg concentrations were chosen based on levels (low contamination, ~0.12 mg kg-1 wet weight (ww); and high contamination, ~1.6 mg kg-1ww found in common A. regius prey species from contaminated coastal areas (Cardoso et al., 2014; Kannan et al., 1998; Nunes et al., 2008). The pellets given to fish allocated to non-contaminated and contaminated treatments had approximately 0.60 ± 0.01 mg kg⁻Âź dry weight (dw) and 8.02 ± 0.01 mg kg⁻Âź dw of MeHg, respectively, which were considered to mimic the concentrations found in the field (see Maulvault et al., 2016, 2017). Feed composition, manufacturing and MeHg spiking processes were executed as described by Maulvault et al. (2016)." (Page 4/5, Lines 30-32/1-6)

References Cardoso, P. G., Pereira, E., Duarte, A. C. and Azeiteiro, U. M.: Temporal characterization of mercury accumulation at different trophic levels and implications for metal biomagnification along a coastal food web, Mar. Pollut. Bull., 87(1), 39–47, doi:10.1016/j.marpolbul.2014.08.013, 2014. Kannan, K., Smith Jr., R. G., Lee, R. F., Windom, H. L., Heitmuller, P. T., Macauley, J. M. and Summers, J. K.: Distribution of Total Mercury and Methyl Mercury in Water , Sediment , and Fish from South Florida Estuaries, Arch. Environ. Contam. Toxicol., 34, 109–118, doi:10.1007/s002449900294, 1998. Nunes, M., Coelho, J. P., Cardoso, P. G., Pereira, M. E., Duarte, A. C. and Pardal, M. A.: The macrobenthic community along a mercury contamination in a temperate estuarine system (Ria de Aveiro, Portugal), Sci. Total Environ., 405(1–3), 186–194, doi:10.1016/j.scitotenv.2008.07.009, 2008.

Comment #2: "In the discussion section, comparative results of mercury accumulation and biomarker response are missing. The study of Biomarkers is quite complex as responses can be influenced by many parameters. In this sense, there are several studies on biomarker response to mercury in the literature. Such studies should also

be mentioned to provide information on the sensitivity of this species comparing to others, as well as to know the relevance of the used Hg concentration."

Response: The authors would like to point out that we have already synthetized some of the literature available on how these stressors prompt oxidative stress response system in the Introduction when we present the reasoning underpinning our approach (Page 2/3, Lines 26-30/1-15). We would also like to highlight that the reason we did not use a comparative Hg toxicity approach was that it is not the main aim of our work. Using the same MeHg contaminated feed, our group has recently published (inclusively this year) other experimental works where we compare the accumulation and toxicological effects (namely on oxidative stress and other enzymes) of mercury with what is described in the general literature (mainly Maulvault et al., 2017; but see also Maulvault et al., 2016 and Sampaio et al., 2016). Our main goal was to disentangle how the triple interaction of warming, acidification and mercury can modulate organism physiology (mainly through oxidative stress), and help predict fish physiological status in future ocean conditions. It was not our intention to give emphasis on mercury effects per se. Furthermore, from our perspective, the most important finding in the manuscript is that acidification counteracted the effects of both mercury contamination and warming. Thus, if we had to set a hierarchy of stressor "importance" to be explained, acidification would be on the first place, not mercury contamination. Moreover, what is important and novel in the present work is not the isolated stressors, but the interactions between them. However, taking the referee's comment into account, we do agree that it would be useful to better contextualize our study. Thus, following this suggestion, we compared these results with other studies where interactions between Hg and climate stressors were assessed: "Moreover, to cope with oxidative stress, A. regius displayed enhanced CAT, SOD and GST activities under contaminated and warming scenarios, which is in line with previous studies reporting an enhanced anti oxidative stress response in fish (Maulvault et al., 2017; Pimentel et al., 2015; Vieira et al., 2009)." (Page 11, Lines 25-29) "Increased $CO_2$ (co-occuring with Hg contamination) is linked to upregulation of the lysosome-autophagy pathway, which is responsible for removing damaged pro-

teins and organelles, effectively reducing oxidative stress (Wang et al., 2017). This mechanism may contribute to alleviate not only Hg induced stress, but also warming-related oxidative stress." (Pages11/12, Lines 31/1-3)

References Maulvault, A. L., Custodio, A., Anacleto, P., Repolho, T., Pousao, P., Nunes, M. L., Diniz, M., Rosa, R. and Marques, A.: Bioaccumulation and elimination of mercury in juvenile seabass (Dicentrarchus labrax) in a warmer environment, Environ. Res., 149, 77–85, doi:10.1016/j.envres.2016.04.035, 2016. Maulvault, A. L., Barbosa, V., Alves, R., Custódio, A., Anacleto, P., Repolho, T., Pousão Ferreira, P., Rosa, R., Marques, A. and Diniz, M.: Ecophysiological responses of juvenile seabass ( Dicentrarchus labrax ) exposed to increased temperature and dietary methylmercury, Sci. Total Environ., 586, 551–558, doi:10.1016/j.scitotenv.2017.02.016, 2017. Sampaio, E., Maulvault, A. L., Lopes, V. M., Paula, J. R., Barbosa, V., Alves, R., Pousão-Ferreira, P., Repolho, T., Marques, A. and Rosa, R.: Habitat selection disruption and lateralization impairment of cryptic flatfish in a warm, acid, and contaminated ocean, Mar. Biol., 163(10), 217, doi:10.1007/s00227-016-2994-8, 2016.

Comment #3: "In the abstract, (page 1 line 20), introduction (page 3 line 20) and methodology (page 4 line 23) pCO2 concentration is given as 1100 $\mu$atm, while the actual value used was 1500 $\mu$atm. Please correct."

Response: The authors would like to clarify that 1100 $\mu$atm was the difference between both CO2 levels used (400 and 1500 $\mu$atm), i.e. delta ($\Delta$) CO2 = 1100 $\mu$atm. The presentation rationale follows that used for presenting temperature effects: we used 19 and 23 °C, i.e. delta ($\Delta$) T = 4 °C.

Comment #4: "The fishes were taken from an aquaculture station. Were the physico-chemical parameters measured at the station? This is relevant to known the levels of pH and temperature that organisms were acclimated at the long-term."

Response: Physico-chemical parameters at the aquaculture station were maintained under normal levels of ambient pH ($\sim$8.00) and seawater temperatures registered at

that time of the year (19 °C), which we used to serve as our control parameters. We have added this information in the text: "Juvenile *Argyrosomus regius* (n ≃ 100; Fig. 5) (mean ± SD; total weight: 4.26 ± 2.8 g; total length: 6.30 ± 1.2 cm) from EPPO - IPMA (Estação Piloto de Piscicultura de Olhão – Instituto Português do Mar e da Atmosfera, Portugal) where fish were maintained under standard summer season environmental parameters (pH = 8.0 and 19 °C). In August 2014, fish were transported to the facilities of Laboratório Marítimo da Guia (LMG, MARE, Faculdade de Ciências, Universidade de Lisboa)." (Page 3/4, Lines 28-30/1-2)

Comment #5:"Page 4 Line 5- Ammonia levels is an important issue at toxicity tests, especially with fishes, as it can interfere on the toxic responses. Authors mention that ammonia (along with nitrate and nitrite) levels were kept within recommended levels. How was this performed? What are the recommended levels? Please give more details."

Response: We apologize for not having provided more detail on these matters. Specifically: Ammonia ($NH_3/NH_4+$), nitrite ($NO_2-$) and nitrate ($NO_3-$) concentrations were daily checked (Colorimetric kits, Aquamerk, Germany), and kept below detectable levels (i.e. $NH_3/NH_4+ < 0.25$ mg l-1; $NO_2- < 0.10$ mg l-1; $NO_3- < 0.2$ mg l-1). They were kept such low levels by a continuous seawater flux, and by the biological filter described (Page 4, Lines 2-5). As detailed in the Methods section (Page 4, Lines 5-9), each experimental unit (or recirculatory aquatic system, RAS) was a semi-closed system with a constant seawater flux (complete turnover rate in 24h) precisely to maintain environmental parameters such as salinity and nutrients. We have added the pertinent information in the text: "To prevent fluctuations in environmental parameters, each RAS worked as a semi-closed system, with constant low flow external water input (flux > 2 l h-1; 50 l tank turnover rate = 24 h). Consequently, ammonia ($NH_3/NH_4+$), nitrite ($NO_2-$) and nitrate ($NO_3-$) concentrations were daily checked (Colourimetric kits, Aquamerk, Germany), and kept below detectable levels (i.e. $NH_3/NH_4+ < 0.25$ mg l-1; $NO_2- < 0.10$ mg l-1; $NO_3- < 0.20$ mg l-1), and salinity was kept at 35.0 ± 1.0 (V2 Refractometer,

TMC Iberia, Portugal)." (Page 4, Lines 5-12).

Comment #6:"Salinity should be given as psu or without unit."

Response: We have removed units from salinity measurements.

Comment #7:"Page 4 line 13- Please give more details on alkalinity measurements, such as the equipment used, storage of samples, the use of certified materials..."

Response: We have added the requested information for alkalinity and pHT: "Seawater carbonate system speciation (Table S1) was calculated once every week from pHtotal scale (pHT ) and total alkalinity. pHT was quantified via a Metrohm pH meter (826 pH mobile, Metrohm, Filderstadt, Germany) connected to a glass electrode (Schott IoLine, SI analytics, $\pm$ 0.001) and calibrated against TRIS–HCl (TRIS) and 2-aminopyridine-HCl (AMP; Mare, Liège, Belgium) seawater buffers (Dickson et al., 2007). Total alkalinity was measured spectrophotometrically (wavelength = 595 nm; UV-1800 Shimadzu, Japan) through base neutralization by formic acid and a pH sensitive dye (bromophenol blue), following Sarazin et al. (1999). Total dissolved inorganic carbon (CT), pCO₂ and aragonite saturation were calculated using CO2SYS software (Lewis and Wallace, 1998), with dissociation constants from Mehrbach et al. ( 1973) as refitted by Dickson and Millero (1987)." (Page 4, Lines 20-28).

Comment #8:"Page 4 Line 20- The method for mercury contamination is confusing. MeHg exposure was performed by food intake and fished were fed two to three times a day. How was the difference between food intakes measured? Authors states that ingestion decreased due to changes in metabolism, but how was this measured? Where is this result? How much mercury was given as total in the experiment? How much of this metal remain dissolved in the water column?"

Response: We address each question below: Differences in food intake were not measured, as rare uneaten pellets were removed together with faeces (Page 5, Lines 8-9) and were not weighted. Thus, we have removed changes in food intake as the main underlying mechanism for differences Hg accumulation and, following further comments from Referee 1, have changed our rationale to a more broader perspective: "Instead, our results support recent studies demonstrating that hypercapnia dampens Hg accumulation in marine organisms (Li et al., 2017; Sampaio et al., 2016; Wang et al., 2017). There are several possible reasons which may underpin such an interaction, encompassing digestive (reduced digestive efficiency, reduced uptake through the gut membrane, reduced appetite, increased Hg depuration) and molecular (competition between Hg and H+ ions for binding sites, impacts on Hg plasma transport, lower phospholipidic membrane permeability) mechanisms (Li et al., 2017). A recent study has also found that the lysosome-autophagy pathway was up-regulated by combined exposure to Hg and increased CO2, enabling better animal fitness which may potentially reduce Hg accumulation and toxicity (Wang et al., 2017). In addition, taking into account that the occurrence of both warming and acidification changes physiological thresholds (Christensen et al., 2011; Harley et al., 2006; Rosa et al., 2013; Rosa and Seibel, 2008), a degree of metabolic depression may also play a role on decreasing HgT accumulation (Dijkstra et al., 2013; Sampaio et al., 2016)." (Page 11, Lines 4-16) Fish were fed 2-3 times a day, but the amount of food per day was fixed at 1% mean fish weight: 4.26 g (as specified in Page 5, line 8/9) * 0.01 = 42.6 mg. Since there were 30 experimental days, then: 42.6 mg * 30 d =1278 mg or 0.001278 kg feed per fish In the pellet we have approximately 8.28 mg of HgT per Kg of food (dry weight), thus: 8.28 * 0.001278 = 0.0106 mg of HgT were given per fish, at the end of the 30-day trial. We have added the following information in the text: "..at the end of 30 days, each fish was given approximately 0.0106 mg of HgT." (Page 5, Line 10/11) Previous studies using the same food pellet manufacturing and MeHg spiking process have found that no mercury was leeched into the water column with this feed (below detectable levels; Maulvault et al., 2016).

References Maulvault, A. L., Custodio, A., Anacleto, P., Repolho, T., Pousao, P., Nunes, M. L., Diniz, M., Rosa, R. and Marques, A.: Bioaccumulation and elimination of mercury in juvenile seabass (Dicentrarchus labrax) in a warmer environment, Environ. Res.,

149, 77–85, doi:10.1016/j.envres.2016.04.035, 2016.

Comment #9: "In the experimental set-up, the setup "IV" is the same as the setup "II", 19 ◦C, 400 pCO2 $\mu$atm and contaminated feed (MeHg: 8.02 mg kgÂ'z; HgT: 8.28 mg kgÂ'z). Setup IV should be 19 ◦C, 1500$\mu$atm and contaminated feed."

Response: We have corrected the characteristics of setup iv): "19 °C, 1500 pCO2 $\mu$atm and contaminated feed"

Comment #10: "In the methodology section, it is mentioned that Reference material was also used to validate measurements of metal content. However, results of recovery percentage in not given. Please include this data as it validates the measurements."

Response: We have included a new table (Table S1), where we include this information:

Standard reference material Total Hg Present work DORM-4* 0.390 ± 0.025 Certified value 0.410 ± 0.055

Comment #11: "Page 8 line 20-25 concentration of Hg was lower in muscle but concentration in liver and gills was actually the same considering error between replicates."

Response: Indeed our p-value comparing levels in Liver & Gill was 0.181 and we have corrected the sentence, removing the implicated difference between HgT accumulation in the liver and the gills: "Hg concentration was lower in the muscle compared to the other two organs analyzed (Muscle & Liver / Muscle & Gills, p < 0.001, GLM Analysis in Table 1, Figure 1a)." (Page 9, Lines 18-20)

Comment #12: "Figure 1d the 400 and 1500 $\mu$atm are inverted"

Response: Corrected.

Comment #13: "Page 8 line27- As expected, catalase activity was affected by mercury contamination, but was this biomarker affected by pCO2 also? What about warming? This is briefly mentioned in the discussion section, but the results are not given."

[Figure]

Response: As we have detailed in the Methods section: "Best model selection fit for our data was found using the Akaike Information Criterion (AIC), a widespread indicator that balances model complexity with model quality of fitness (Quinn and Keough, 2002). Thus, models were simplified and factors that did not influence data variation were removed." (Page 8, Lines 5-8) In other words, using the AIC we can remove factors and interactions that do not help in explaining the data, but only add noise to the analysis. Thus, we can safely say that warming did not have an effect on CAT activity since the AIC excluded this factor from the analysis completely. As for increased CO2, the AIC did include it in the model, which means that it has influence over our data, but that influence is not significant (as we usually set an $\alpha$ = 0.05 in biological statistics and our analysis yielded a p = 0.116 for CO2 * MeHg). It is important to state that there is a continuous argument between statisticians over what is relevant to include or not in the discussion of this type of analysis. In our opinion, given the consistent effects on the rest of the antioxidant and physiological defense response machinery, we felt it was important to mention that an effect of CO2 in shaping CAT activity is a possibility, maybe just not detected on this study. However, we acknowledge that it was a non-significant effect. "While it is worth mentioning that increased CO2 played a minor role in CAT activity (non-significant, p = 0.116), regarding the other enzymes, hypercapnia as a sole stressor significantly augmented antioxidant activity." (Page 11, Lines 28-30)

Comment #14: "While the values for Hsp70 are given in each organ analyzed, the results for the other biomarkers are not specified. Were they measured only in the liver or other parts? Please include this information in the results and also in the methodology."

Response: Unfortunately we did not have enough tissue to perform enzymatic assays for oxidative stress in the liver and gills. Mercury concentration determination required almost the totally of these organs, which left us only enough sample for heat shock protein response (requires only a small tissue). Thus, the rest of the enzymatic assays were all performed in the muscle. As requested, we have added this information throughout the text, including figure captions: "As an end-product of oxidative stress,

malondialdehyde (MDA) concentration was used as a proxy to assess extent of lipid peroxidation in the muscle." (Page 6, Lines 25-26) "Catalase activity in the muscle was assessed through an adaptation of the method described by Johansson and Borg (1988)." (Page 7, Line 8) "SOD activity in the muscle was determined following the nitro blue tetrazolium (NBT) method adapted from Sun et al. (1988)."(Page 6, Line 19-20) "GSTactivity in the muscle was determined according to the procedure described by Habig et al. (1974) and optimized for 96-well microplate (Sigma Technical Bulletin, GST Assay Kit CS0410)." (Page 7, Lines 2-4) "Heat shock protein (Hsp70/Hsc70) content in the muscle, liver and gills was assessed by Enzyme-Linked Immunoabsorbent Assay (ELISA) protocol adapted from Njemini et al. (2005)." (Page 8, Lines 13-14) "Subsequently, lipid peroxidation and oxidative stress were measured in the muscle tissue. A significant antagonistic effect. . ." (Page 9, Lines 23) "Figure 2. Malondialdehyde (MDA) build-up concentrations (mean ± SE) in A. regius muscle driven by an interaction" (Page 2, Line 5) "Figure 3. a) Catalase (CAT) enzyme activities (mean ± SE) driven by MeHg contamination (Non-contaminated and Contaminated). b) Superoxide dismutase (SOD) activities (mean ± SE) in A. regius muscle. . ." (Page 24, Line 5-6) "Figure 4. Glutathione S-Transferase (GST) activities (mean ± SE) in A. regius muscle driven by:" (Page 25, Line 3)

Comment #15: "Page 9 lines 15-20 the information "However, our AIC-chosen best model indicated that mercury may diminish organism Fulton condition" is contradictory to what is mentioned on the results: "Fulton condition (K) did not show any significant differences between treatments (MeHg, p > 0.05, GLM analysis in Table 1).""

Response: We have removed this statement. "The present study showed that Hg contamination, ocean warming and acidification interactively affected fish physiology at sublethal levels, i.e. zero mortality and also no effects on Fulton condition were registered." (Page 10, Lines 14-19)

Technical corrections:

Technical correction #1: "Page 2 Lines 1-2: CO2 should be subscript"

Response: Corrected.

Technical correction #2: "Page 4 Line 2: m3 should be superscript"

Response: Corrected.

Technical correction #3: "Page 4 Line 10: CO2 should be subscript"

Response: Corrected.

Technical correction #4: "Page 5 Line 5: lenght3 check type error"

Response: Corrected.

Technical correction #5: "Page 6 Line 12: mg-1 should be superscript"

Response: Corrected.

Technical correction #6: "Page 6 Line 23: mg-2 should be superscript"

Response: Corrected.

Technical correction #7: "Pag 10 Line 20: H+ should be superscript"

Response: Corrected.

Technical correction #8: "Page 9 line 17: the word non-lethal could be replaced by sublethal, which is more often used in toxicity studies"

Response: We have changed the terms.

Technical correction #9: "Page 9 line 19: A. regius should be written in italic"

Response: Changed.

Please also note the supplement to this comment:
https://www.biogeosciences-discuss.net/bg-2017-147/bg-2017-147-AC2-

supplement.pdf

**Supplement:**

**Referee #2 (Manoela Orte, PhD)**

**General comments**

*"The interactive effects of acidification, warming and the presence of the metal Hg was assessed in the Fish Argyrosomus regius. Bioaccumulation of Hg was measured in different organs of the fish and sublethal toxic responses were also analyzed by the use of biomarkers. The topic is highly relevant since research regarding global change issues should preferably focus on a multi-stressors approach. Furthermore, mercury is an important persistent contaminant found in coastal environments around the world and information regarding its interactive toxicological effects with other parameters such as acidification and warming are of great value. In general, the writing is clear and the data obtained is interesting. However, some issues regarding the methodological approach used are not well explained and there are some information at the results and discussion section that should be included. Therefore I recommend that the authors perform the suggested corrections before the article is published."*

**Response:** We thank the referee for her comments and suggested terminology which helped to contextualize our manuscript better and improve the overall scientific outcomes found. We have addressed the lack of methodological procedures and hope that we have reached the publication standards upheld by the referee. We have accepted most of the referee's suggestions and, below, discuss each comment in a point-by-point manner. Please note that Page and Line numbers now correspond to the marked up version of the manuscript.

**Specific comments**

*Comment #1:* *"The focus of the study is the evaluation of toxic responses of the metal Hg in a global change scenario. It is mentioned that concentration of Hg was chosen according to environmental measurements, however data on the range of toxic concentrations of this metal to this species or other fish species is not included. Considering that the article uses an ecotoxicological approach and therefore it is based on dose-response concentration it is crucial that more details on this subject is included, such as values of toxicity for fishes and environmental values within contaminated and non contaminated areas, especially in the area where the study was conducted."*

**Response:** Mercury concentrations chosen for this study were based on levels of contamination found in contaminated coastal areas (specifically the extensively studied contaminated estuary of Aveiro, Portugal) for species that are natural prey ofthe meagre (e.g. Cardoso et al., 2014; Nunes et al., 2008). Not exclusive to the Eastern Atlantic coast, these mercury concentrations can also be found in other areas globally (e.g. Kannan et al., 1998).

We thank the reviewer for pointing out the need for contextualization and have added:

"Given our dietary option, ecologically relevant MeHg concentrations were chosen based on levels (low contamination, ~0.12 mg kg$^{-1}$ wet weight (ww); and high contamination, ~1.6 mg kg$^{-1}$ww found in common *A. regius* prey species from contaminated coastal areas (Cardoso et al., 2014; Kannan et al., 1998; Nunes et al., 2008). The pellets given to fish allocated to non-contaminated and contaminated treatments had approximately $0.60 \pm 0.01$ mg kg$^{-1}$ dry weight (dw) and $8.02 \pm 0.01$ mg kg$^{-1}$ dw of MeHg, respectively, which were considered to mimic the concentrations found in the field (see Maulvault et al., 2016, 2017). Feed composition, manufacturing and MeHg spiking processes were executed as described by Maulvault et al. (2016)." (Page 4/5, Lines 30-32/1-6)

References

Cardoso, P. G., Pereira, E., Duarte, A. C. and Azeiteiro, U. M.: Temporal characterization of mercury accumulation at different trophic levels and implications for metal biomagnification along a coastal food web, Mar. Pollut. Bull., 87(1), 39–47, doi:10.1016/j.marpolbul.2014.08.013, 2014.

Kannan, K., Smith Jr., R. G., Lee, R. F., Windom, H. L., Heitmuller, P. T., Macauley, J. M. and Summers, J. K.: Distribution of Total Mercury and Methyl Mercury in Water , Sediment , and Fish from South Florida Estuaries, Arch. Environ. Contam. Toxicol., 34, 109–118, doi:10.1007/s002449900294, 1998.

Nunes, M., Coelho, J. P., Cardoso, P. G., Pereira, M. E., Duarte, A. C. and Pardal, M. A.: The macrobenthic community along a mercury contamination in a temperate estuarine system (Ria de Aveiro, Portugal), Sci. Total Environ., 405(1–3), 186–194, doi:10.1016/j.scitotenv.2008.07.009, 2008.

***Comment #2:*** *"In the discussion section, comparative results of mercury accumulation and biomarker response are missing. The study of Biomarkers is quite complex as responses can be influenced by many parameters. In this sense, there are several studies on biomarker response to mercury in the literature. Such studies should also be mentioned to provide information on the sensitivity of this species comparing to others, as well as to know the relevance of the used Hg concentration."*

**Response:** The authors would like to point out that we have already synthetized some of the literature available on how these stressors prompt oxidative stress response system in the Introduction when we present the reasoning underpinning our approach (Page 2/3, Lines 26-30/1-15).

We would also like to highlight that the reason we did not use a comparative Hg toxicity approach was that it is not the main aim of our work. Using the same MeHg contaminated feed, our group has recently published (inclusively this year) other experimental works where we compare the accumulation and toxicological effects (namely on oxidative stress and other enzymes) of mercury with what is described in the general literature (mainly Maulvault et al., 2017; but see also Maulvault et al., 2016 and Sampaio et al., 2016).

Our main goal was to disentangle how the triple interaction of warming, acidification and mercury can modulate organism physiology (mainly through oxidative stress), and help predict fish physiological status in future ocean conditions. It was not our intention to give emphasis on mercury effects per se. Furthermore, from our perspective, the most important finding in the manuscript is that acidification counteracted the effects of both mercury contamination and warming. Thus, if we had to set a hierarchy of stressor "importance" to be explained, acidification would be on the first place, not mercury contamination. Moreover, what is important and novel in the present work is not the isolated stressors, but the interactions between them.

However, taking the referee's comment into account, we do agree that it would be useful to better contextualize our study. Thus, following this suggestion, we compared these results with other studies where interactions between Hg and climate stressors were assessed:

"Moreover, to cope with oxidative stress, *A. regius* displayed enhanced CAT, SOD and GST activities under contaminated and warming scenarios, which is in line with previous studies reporting an enhanced anti oxidative stress response in fish (Maulvault et al., 2017; Pimentel et al., 2015; Vieira et al., 2009)." (Page 11, Lines 25-29)

"Increased $CO_2$ (co-occuring with Hg contamination) is linked to upregulation of the lysosome-autophagy pathway, which is responsible for removing damaged proteins and organelles, effectively reducing oxidative stress (Wang et al., 2017). This mechanism may contribute to alleviate not only Hg induced stress, but also warming-related oxidative stress." (Pages11/12, Lines 31/1-3)

**Comment #3:** *"In the abstract, (page 1 line 20), introduction (page 3 line 20) and methodology (page 4 line 23) pCO2 concentration is given as 1100 µatm, while the actual value used was 1500 µatm. Please correct."*

**Response:** The authors would like to clarify that 1100 µatm was the difference between both $CO_2$ levels used (400 and 1500 µatm), i.e. delta ($\Delta$) $CO_2$ = 1100 µatm. The presentation rationale follows that used for presenting temperature effects: we used 19 and 23 ºC, i.e. delta ($\Delta$) T = 4 ºC.

*Comment #4:* *"The fishes were taken from an aquaculture station. Were the physico-chemical parameters measured at the station? This is relevant to known the levels of pH and temperature that organisms were acclimated at the long-term."*

**Response:** Physico-chemical parameters at the aquaculture station were maintained under normal levels of ambient pH (~8.00) and seawater temperatures registered at that time of the year (19 ºC), which we used to serve as our control parameters. We have added this information in the text:

"Juvenile *Argyrosomus regius* (n $\simeq$ 100; Fig. 5) (mean $\pm$ SD; total weight: 4.26 $\pm$ 2.8 g; total length: 6.30 $\pm$ 1.2 cm) from EPPO - IPMA (Estação Piloto de Piscicultura de Olhão – Instituto Português do Mar e da Atmosfera, Portugal) where fish were maintained under standard summer season environmental parameters (pH = 8.0 and 19 ºC). In August 2014, fish were transported to the facilities of Laboratório Marítimo da Guia (LMG, MARE, Faculdade de Ciências, Universidade de Lisboa)." (Page 3/4, Lines 28-30/1-2)

*Comment #5:* *"Page 4 Line 5- Ammonia levels is an important issue at toxicity tests, especially with fishes, as it can interfere on the toxic responses. Authors mention that ammonia (along with nitrate and nitrite) levels were kept within recommended levels. How was this performed? What are the recommended levels? Please give more details."*

**Response:** We apologize for not having provided more detail on these matters. Specifically:

Ammonia ($NH_3/NH_4^+$), nitrite ($NO_2^-$) and nitrate ($NO_3^-$) concentrations were daily checked (Colorimetric kits, Aquamerk, Germany), and kept below detectable levels (i.e. $NH_3/NH_4^+ < 0.25$ mg $l^{-1}$; $NO_2^- < 0.10$ mg $l^{-1}$; $NO_3^- < 0.2$ mg $l^{-1}$).

They were kept such low levels by a continuous seawater flux, and by the biological filter described (Page 4, Lines 2-5). As detailed in the Methods section (Page 4, Lines 5-9), each experimental unit (or recirculatory aquatic system, RAS) was a semi-closed system with a constant seawater flux (complete turnover rate in 24h) precisely to maintain environmental parameters such as salinity and nutrients.

We have added the pertinent information in the text:

"To prevent fluctuations in environmental parameters, each RAS worked as a semi-closed system, with constant low flow external water input (flux > 2 l h$^{-1}$; 50 l tank turnover rate = 24 h). Consequently, ammonia ($NH_3/NH_4^+$), nitrite ($NO_2^-$) and nitrate ($NO_3^-$) concentrations were daily checked (Colourimetric kits,

Aquamerk, Germany), and kept below detectable levels (i.e. $NH_3/NH_4^+ < 0.25$ mg l$^{-1}$; $NO_2^- < 0.10$ mg l$^{-1}$; $NO_3^-$ $< 0.20$ mg l$^{-1}$), and salinity was kept at $35.0 \pm 1.0$ (V2 Refractometer, TMC Iberia, Portugal)." (Page 4, Lines 5-12).

*Comment #6:* "*Salinity should be given as psu or without unit.*"

**Response:** We have removed units from salinity measurements.

*Comment #7:* "*Page 4 line 13- Please give more details on alkalinity measurements, such as the equipment used, storage of samples, the use of certified materials…*"

**Response:** We have added the requested information for alkalinity and pH$_T$:

"Seawater carbonate system speciation (Table S1) was calculated once every week from pH$_{total\ scale}$ (pH$_T$ ) and total alkalinity. pH$_T$ was quantified via a Metrohm pH meter (826 pH mobile, Metrohm, Filderstadt, Germany) connected to a glass electrode (Schott IoLine, SI analytics, $\pm$ 0.001) and calibrated against TRIS–HCl (TRIS) and 2-aminopyridine-HCl (AMP; Mare, Liège, Belgium) seawater buffers (Dickson et al., 2007). Total alkalinity was measured spectrophotometrically (wavelength = 595 nm; UV-1800 Shimadzu, Japan) through base neutralization by formic acid and a pH sensitive dye (bromophenol blue), following Sarazin et al. (1999). Total dissolved inorganic carbon (C$_T$), pCO$_2$ and aragonite saturation were calculated using CO2SYS software (Lewis and Wallace, 1998), with dissociation constants from Mehrbach et al. ( 1973) as refitted by Dickson and Millero (1987)." (Page 4, Lines 20-28).

*Comment #8:* "*Page 4 Line 20- The method for mercury contamination is confusing. MeHg exposure was performed by food intake and fished were fed two to three times a day. How was the difference between food intakes measured? Authors states that ingestion decreased due to changes in metabolism, but how was this measured? Where is this result? How much mercury was given as total in the experiment? How much of this metal remain dissolved in the water column?*"

**Response:** We address each question below:

Differences in food intake were not measured, as rare uneaten pellets were removed together with faeces (Page 5, Lines 8-9) and were not weighted. Thus, we have removed changes in food intake as the main underlying mechanism for differences Hg accumulation and, following further comments from Referee 1, have changed our rationale to a more broader perspective:

"Instead, our results support recent studies demonstrating that hypercapnia dampens Hg accumulation in marine organisms (Li et al., 2017; Sampaio et al., 2016; Wang et al., 2017). There are several possible reasons

which may underpin such an interaction, encompassing digestive (reduced digestive efficiency, reduced uptake through the gut membrane, reduced appetite, increased Hg depuration) and molecular (competition between Hg and $H^+$ ions for binding sites, impacts on Hg plasma transport, lower phospholipidic membrane permeability) mechanisms (Li et al., 2017). A recent study has also found that the lysosome-autophagy pathway was up-regulated by combined exposure to Hg and increased $CO_2$, enabling better animal fitness which may potentially reduce Hg accumulation and toxicity (Wang et al., 2017). In addition, taking into account that the occurrence of both warming and acidification changes physiological thresholds (Christensen et al., 2011; Harley et al., 2006; Rosa et al., 2013; Rosa and Seibel, 2008), a degree of metabolic depression may also play a role on decreasing HgT accumulation (Dijkstra et al., 2013; Sampaio et al., 2016)." (Page 11, Lines 4-16)

Fish were fed2-3 times a day, but the amount of food per day was fixed at 1% mean fish weight: 4.26 g (as specified in Page 5, line 8/9) * 0.01 = 42.6 mg.

Since there were 30 experimental days, then: 42.6 mg * 30 d =1278 mg or 0.001278 kg feed per fish

In the pellet we have approximately 8.28 mg of HgT per Kg of food (dry weight), thus: 8.28 * 0.001278 = **0.0106 mg of HgT** were given **per fish**, at the end of the **30-day trial**. We have added the following information in the text:

"..at the end of 30 days, each fish was given approximately 0.0106 mg of HgT." (Page 5, Line 10/11)

Previous studies using the same food pellet manufacturing and MeHg spiking process have found that no mercury was leeched into the water column with this feed (below detectable levels; Maulvault et al., 2016).

References

Maulvault, A. L., Custodio, A., Anacleto, P., Repolho, T., Pousao, P., Nunes, M. L., Diniz, M., Rosa, R. and Marques, A.: Bioaccumulation and elimination of mercury in juvenile seabass (Dicentrarchus labrax) in a warmer environment, Environ. Res., 149, 77–85, doi:10.1016/j.envres.2016.04.035, 2016.

**Comment #9:** *"In the experimental set-up, the setup "IV" is the same as the setup "II", 19 ∘C, 400 pCO2 μatm and contaminated feed (MeHg: 8.02 mg kg´z; HgT: 8.28 mg kg´z). Setup IV should be 19 ∘C, 1500μatm and contaminated feed."*

**Response**: We have corrected the characteristics of setup iv): "19 °C, 1500 pCO$_2$ μatm and contaminated feed"

*Comment #10:* *"In the methodology section, it is mentioned that Reference material was also used to validate measurements of metal content. However, results of recovery percentage in not given. Please include this data as it validates the measurements."*

**Response:** We have included a new table (Table S1), where we include this information:

| | Standard reference material | Total Hg |
|---|---|---|
| Present work | DORM-4* | $0.390 \pm 0.025$ |
| Certified value | | $0.410 \pm 0.055$ |

*Comment #11:* *"Page 8 line 20-25 concentration of Hg was lower in muscle but concentration in liver and gills was actually the same considering error between replicates."*

**Response:** Indeed our p-value comparing levels in Liver & Gill was 0.181 and we have corrected the sentence, removing the implicated difference between HgT accumulation in the liver and the gills:

"Hg concentration was lower in the muscle compared to the other two organs analyzed (Muscle & Liver / Muscle & Gills, $p < 0.001$, GLM Analysis in Table 1, Figure 1a)." (Page 9, Lines 18-20)

*Comment #12:* *"Figure 1d the 400 and 1500 μatm are inverted"*

**Response:** Corrected.

*Comment #13:* *"Page 8 line27- As expected, catalase activity was affected by mercury contamination, but was this biomarker affected by pCO2 also? What about warming? This is briefly mentioned in the discussion section, but the results are not given."*

**Response:** As we have detailed in the Methods section:

"Best model selection fit for our data was found using the Akaike Information Criterion (AIC), a widespread indicator that balances model complexity with model quality of fitness (Quinn and Keough, 2002). Thus, models were simplified and factors that did not influence data variation were removed." (Page 8, Lines 5-8)

In other words, using the AIC we can remove factors and interactions that do not help in explaining the data, but only add noise to the analysis. Thus, we can safely say that warming did not have an effect on CAT activity since the AIC excluded this factor from the analysis completely.

As for increased $CO_2$, the AIC did include it in the model, which means that it has influence over our data, but that influence is not significant (as we usually set an $\alpha = 0.05$ in biological statistics and our analysis yielded a $p = 0.116$ for $CO_2$ * MeHg). It is important to state that there is a continuous argument between statisticians over what is relevant to include or not in the discussion of this type of analysis. In our opinion, given the consistent effects on the rest of the antioxidant and physiological defense response machinery, we felt it was important to mention that an effect of $CO_2$ in shaping CAT activity is a possibility, maybe just not detected on this study. However, we acknowledge that it was a non-significant effect.

"While it is worth mentioning that increased $CO_2$ played a minor role in CAT activity (non-significant, $p = 0.116$), regarding the other enzymes, hypercapnia as a sole stressor significantly augmented antioxidant activity." (Page 11, Lines 28-30)

**Comment #14:** *"While the values for Hsp70 are given in each organ analyzed, the results for the other biomarkers are not specified. Were they measured only in the liver or other parts? Please include this information in the results and also in the methodology."*

**Response:** Unfortunately we did not have enough tissue to perform enzymatic assays for oxidative stress in the liver and gills. Mercury concentration determination required almost the totally of these organs, which left us only enough sample for heat shock protein response (requires only a small tissue). Thus, the rest of the enzymatic assays were all performed in the muscle. As requested, we have added this information throughout the text, including figure captions:

"As an end-product of oxidative stress, malondialdehyde (MDA) concentration was used as a proxy to assess extent of lipid peroxidation in the muscle." (Page 6, Lines 25-26)

"Catalase activity in the muscle was assessed through an adaptation of the method described by Johansson and Borg (1988)." (Page 7, Line 8)

"SOD activity in the muscle was determined following the nitro blue tetrazolium (NBT) method adapted from Sun et al. (1988)."(Page 6, Line 19-20)

"GSTactivity in the muscle was determined according to the procedure described by Habig et al. (1974) and optimized for 96-well microplate (Sigma Technical Bulletin, GST Assay Kit CS0410)." (Page 7, Lines 2-4)

"Heat shock protein (Hsp70/Hsc70) content in the muscle, liver and gills was assessed by Enzyme-Linked Immunoabsorbent Assay (ELISA) protocol adapted from Njemini et al. (2005)." (Page 8, Lines 13-14)

"Subsequently, lipid peroxidation and oxidative stress were measured in the muscle tissue. A significant antagonistic effect…" (Page 9, Lines 23)

"Figure 2. Malondialdehyde (MDA) build-up concentrations (mean ± SE) in *A. regius* muscle driven by an interaction" (Page 2, Line 5)

"Figure 3. a) Catalase (CAT) enzyme activities (mean ± SE) driven by MeHg contamination (Non-contaminated and Contaminated). b) Superoxide dismutase (SOD) activities (mean ± SE) in *A. regius* muscle…" (Page 24, Line 5-6)

"Figure 4. Glutathione S-Transferase (GST) activities (mean ± SE) in *A. regius* muscle driven by:" (Page 25, Line 3)

**Comment #15:** *"Page 9 lines 15-20 the information "However, our AIC-chosen best model indicated that mercury may diminish organism Fulton condition" is contradictory to what is mentioned on the results: "Fulton condition (K) did not show any significant differences between treatments (MeHg, p > 0.05, GLM analysis in Table 1)."* "

**Response:** We have removed this statement.

"The present study showed that Hg contamination, ocean warming and acidification interactively affected fish physiology at sublethal levels, i.e. zero mortality and also no effects on Fulton condition were registered." (Page 10, Lines 14-19)

**Technical corrections:**

**Technical correction #1:** *"Page 2 Lines 1-2: CO2 should be subscript"*

**Response:** Corrected.

**Technical correction #2:** *"Page 4 Line 2: m3 should be superscript"*

**Response:** Corrected.

**Technical correction #3:** *"Page 4 Line 10: CO2 should be subscript"*

**Response:** Corrected.

**Technical correction #4:** *"Page 5 Line 5: lenght3 check type error"*

**Response:** Corrected.

***Technical correction #5:*** *"Page 6 Line 12: mg-1 should be superscript"*

**Response:** Corrected.

***Technical correction #6:*** *"Page 6 Line 23: mg-2 should be superscript"*

**Response:** Corrected.

***Technical correction #7:*** *"Pag 10 Line 20: H+ should be superscript"*

**Response:** Corrected.

***Technical correction #8:*** *"Page 9 line 17: the word non-lethal could be replaced by sublethal, which is more often used in toxicity studies"*

**Response:** We have changed the terms.

***Technical correction #9:*** *"Page 9 line 19: A. regius should be written in italic"*

**Response:** Changed.

**Ocean acidification dampens warming and contamination effects on the physiological stress response of a commercially important fish**

Eduardo Sampaio[1#*], Ana R Lopes[1#], Sofia Francisco[1], Jose R Paula[1], Marta Pimentel[1], Ana L Maulvault[1,2,3], Tiago Repolho[1], Tiago F Grilo[1], Pedro Pousão-Ferreira[2], António Marques[2,3], Rui Rosa[1]

[1]MARE- Marine Environmental Sciences Centre & Laboratório Marítimo da Guia, Faculdade de Ciências, Universidade de Lisboa, Av. Nossa Senhora do Cabo 939, Cascais 2750-374, Portugal

[2]Divisão de Aquacultura e Valorização (DivAV), Instituto Português do Mar e da Atmosfera (IPMA, I.P.), Av. Brasília, Lisboa 1449-006, Portugal

[3]Interdisciplinary Centre of Marine and Environmental Research (CIIMAR), University of Porto, Rua das Bragas, 289, 4050-123 Porto, Portugal

*Correspondence to*: Eduardo Sampaio (easampaio@fc.ul.pt)

**equally contributed**

**Abstract.** Increases in carbon dioxide ($CO_2$) and other greenhouse gases emissions are leading to changes in ocean temperature and carbonate chemistry, the so-called ocean warming and acidification phenomena, respectively. Methylmercury (MeHg) is the most abundant form of mercury (Hg), well-known for its toxic effects on biota and environmental persistency.  uture interactive effects between  contaminants and climate change stressors are still largely unknown, even though such interactions will play a key role in shaping the ecophysiology of marine organisms. Here we assessed organ-dependent Hg accumulation (gills, liver and muscle) within a warming ($\Delta T = 4$ ºC) and acidification ($\Delta pCO_2 = 1100$ µatm) context, and the respective phenotypic responses of molecular chaperone and antioxidant enzymatic machineries, in a commercially important fish (the meagre *Argyrosomus regius*). After 30 days of exposure, although no mortalities were observed in any treatments, Hg concentration was  enhanced under warming conditions, especially in the liver. On the other hand, increased $CO_2$ decreased Hg accumulation and, despite negative effects prompted as a sole stressor, consistently elicited an  opposing effect relatively to warming and contamination on oxidative stress (catalase, superoxide dismutase and glutathione-S-tranferase activities) and heat shock (Hsp70 levels) responses. Together with $CO_2$-promoted removal of damaged proteins and enzymes, We argue that  simultaneous increase in  hydrogen ($H^+$) and reactive oxygen species (e.g. $O_2^-$)  radicals is partially compensated through chemical reaction equilibrium balancing. Additional multi-stressor experiments are needed to

understand such biochemical mechanisms and further disentangle interactive (additive, synergistic or antagonistic) stressor effects on fish ecophysiology in the oceans of tomorrow.

**1 Introduction**

Atmospheric carbon dioxide ($CO_2$) concentrations have been increasing since the preindustrial era ($\simeq$400 $CO_2$ µatm nowadays), and are expected to reach approximately 1000 $CO_2$ µatm by the year 2100 (IPCC, 2014). Moreover, conjointly with other "greenhouse" gase, increased $CO_2$ has trigger a continuous rise in mean ocean temperatures (nowadays increased by 0.76 ºC from pre-industrial values), and predictions point to a further a 0.3-4.8 °C  increase  by the end of the century (IPCC, 2014). Atmospheric $CO_2$ dissolves in the ocean, altering seawater carbonate chemistry. Carbon dioxide uptake increases hydrogen ion ($H^+$) availability, leading to a concomitant decrease of 0.13-0.42 units in mean ocean pH by the year 2100, i.e. ocean acidification (IPCC, 2014). Due to naturally frequent variations in seawater physicochemical properties (e.g. upwelling events, significant carbon input from river basins), a more accentuated $CO_2$ input will occur in coastal areas, easily reaching pCO$_2$ values beyond 1500 µatm (Melzner, 2013). The combined occurrence of ocean warming and acidification imposes ecophysiological challenges to marine organisms, eliciting interactive negative effects on survival, growth and overall physiological fitness (Harvey et al., 2013; Kroeker et al., 2010; Pimentel et al., 2015).

In addition to global warming and ocean acidification, marine biota will also deal with an additional major stressor: contamination. One of the most concerning and persistent metal contaminants is mercury (Hg) and its ubiquitous environmental compound, methylmercury (MeHg) (Korbas et al., 2011). Inorganic mercury is methylated into organic MeHg by bacteria present in the sediment of estuaries and coastal areas (Dijkstra et al., 2013), augmenting Hg bioavailability, bioaccumulation and biomagnification in marine organisms throughout the food web (Campbell et al., 2005; Evers et al., 2011). In teleost fish, MeHg accumulates preferentially in organ tissue, producing site-specific structural and functional damage (Gonzalez et al., 2005), and comprises around 90–95% of total mercury (HgT) in the organism (Burger et al., 2003; Gray et al., 2000). Mercury accumulation can cause deleterious effects, such as physiological distress, i.e. activation of antioxidant and xenobiotic defense (Gonzalez et al., 2005; Mieiro et al., 2010), behavioural and organ functionality impairments (Berntssen et al., 2003; Sampaio et al., 2016) and ultimately mortality (Coccini et al., 2000).

Contaminant uptake and its impacts are potentially shaped by increased temperature or $CO_2$ and vice-versa (Noyes et al., 2009). Specifically, interactions between temperature and heavy metal contamination influence the physiological tolerance to both stress factors (Sokolova and Lannig, 2008) while exacerbating biological responses (Dorts et al., 2014; Lapointe et al., 2011; Sappal et al., 2014). Consequently, MeHg accumulation is augmented and propagation throughout the food chain is strengthened, until metabolic thresholds are reached (Dijkstra et al., 2013). In parallel, severe acidification (pH < 7)

increases metal availability (Wiener et al., 1990) and toxicity (Han et al., 2014). However,  such effect may be offset by $CO_2$-linked  decrease in mercury accumulation (Sampaio et al., 2016; Schiedek et al., 2007; Wang et al., 2017). Under environmental stressor exposure, a general deleterious biochemical pathway triggered is the formation of oxygen reactive species (ROS) in the organism's cells.

5 Although there is some proof linking ROS production to hypercapnic scenarios (Pimentel et al., 2015), such is particularly true for increased temperature and mercury contamination (Berntssen et al., 2003; Portner, 2002). Increasing ROS concentrations cause protein damage and lipid peroxidation, i.e. oxidative stress, cascading in augmented malondialdehyde content (MDA), one of the final products of lipid peroxidation (Lesser, 2006). As a physiological defense response, ROS production elicits antioxidant activity in the organism. Specifically, a battery of enzymes is activated to eliminate ROS

10 and prevent MDA build-up: superoxide dismutase (SOD), which converts superoxide ($O_2^-$) into hydrogen peroxide ($H_2O_2$); catalase (CAT) which converts $H_2O_2$ into water ($H_2O$) and oxygen ($O_2$); and glutathione S-transferase (GST), which is involved in the protection against xenobiotics and linked to antioxidant defense (Lesser, 2006; Wang et al., 2000). Moreover, tissue-specific heat shock protein (Hsp70) production are also correlated with thermal stress, i.e. high temperatures (Repolho et al., 2014; Rosa et al., 2012, 2014a) and metal contamination (Rajeshkumar and Munuswamy, 2011; Williams et

15 al., 1996). Heat shock proteins help repair, refold and eliminate damaged or denatured proteins, as well as protect and control ROS formation (Sokolova et al., 2011). Given their wide scope, these constituents of the antioxidant enzymatic and protein chaperone machineries are widely used as biomarkers in ecotoxicology to assess fish physiological stress response (e.g. Anacleto et al., 2014; Fonseca et al., 2011; Rosa et al., 2014b).

Despite the inevitability of marine organisms having to cope with simultaneous effects of ocean warming, acidification and

20 persistent contamination (MeHg), no studies have focused on how the interactive effects between these three stressors will challenge  fish ecophysiology. Due to its coastal distribution, the meagre *(Argyrosomus regius)* is particularly susceptible to MeHg accumulation, especially when they migrate towards  estuaries to spawn (Durrieu et al., 2005). Understanding how this commercially important species will deal with the predicted climate change scenarios may provide valuable information on future stock population conditions and potential impacts on coastal food-webs. Within this context,

25 here we performed a 30-day acclimation experiment to investigate organ-dependent Hg accumulation (gills, liver and muscle) under a warming ($\Delta T = 4$ ºC) and acidification ($\Delta CO_2 = 1100$ µatm) context, as well as the respective phenotypic responses of molecular chaperone (Hsp70) and antioxidant enzymatic (SOD, CAT and GST) machineries, in commercially important fish (*A. regius*). The direct consequences at organism (survival rates and condition index) and cellular (lipid peroxidation, MDA) levels were also evaluated.

**2 Material and Methods**

**2.1 Experimental setup and incubation**

Juvenile _Argyrosomus regius_ (n $\simeq$100; Fig. 5) (mean ± SD; total weight: 4.26 ± 2.8 g; total length: 6.30 ± 1.2 cm) from EPPO - IPMA (Estação Piloto de Piscicultura de Olhão – Instituto Português do Mar e da Atmosfera, Portugal) where fish were maintained under standard summer season environmental parameters (pH = 8.0 and 19 ºC). In August 2014, fish were transported to the facilities of Laboratório Marítimo da Guia (LMG, MARE, Faculdade de Ciências, Universidade de Lisboa) . Fish were randomly placed in twenty-four 50l tanks (n = 3-4 per tank) with individual recirculating aquaculture systems (RAS) equipped with glass wool (physical filtration), bio-balls (Fernando Ribeiro Lda) and protein skimmers (biological filtration, ReefSkimPro 850, TMC Iberia), as well as additional UV disinfection (Vecton 120, TMC Iberia) to maintain superior water quality. Natural seawater was pumped directly from the ocean into an 8 m$^3$ storage tank, and subsequently filtered (0.35 µm filters, Fernando Ribeiro Lda) and UV-sterilized (Vecton600, TMC Iberia), before pumping into mixing (n = 24) and respective experimental (n = 24, 50 l) tanks/RAS. To prevent fluctuations in environmental parameters, each RAS worked as a semi-closed system, with constant low flow external water input (flux $\geq$ 2 l h$^{-1}$; 50 l tank turnover rate = 24 h). Consequently, ammonia (NH$_3$/NH$_4^+$), nitrite (NO$_2^-$) and nitrate (NO$_3^-$) concentrations were daily checked (Colourimetric kits, Aquamerk, Germany), and kept below detectable levels (i.e. NH$_3$/NH$_4^+$ < 0.25 mg l$^{-1}$; NO$_2^-$ < 0.10 mg l$^{-1}$; NO$_3^-$ < 0.20 mg l$^{-1}$), and Salinity was kept at 35.0 ± 1.0 (V2 Refractometer, TMC Iberia, Portugal).  Temperature  and pH (multiparametric probe, Multi3420 SET G, WTW) were  measured daily, directly in the holding tanks. Photoperiod was fixed at 12 h light : 12 h dark.

As per experimental conditions, temperature in the tanks was down-regulated using chillers (± 0.1 ºC, Frimar, Fernando Ribeiro Lda), and up-regulated by submerged 200 W heaters (V2Therm, TMC Iberia). Seawater carbonate chemistry was altered through CO$_2$-enriched air input, with pH (8.0 and 7.5) used as proxy measurement.  We used a Profilux system (± 0.1, Profilux 3.1N, GHL) as pH controller, connected to each tank by individual pH probes. Within each RAS, pH was down-regulated by injection of the certified CO$_2$-enriched air (Air Liquide), and up-regulated by injection of atmospheric air. Seawater carbonate system speciation (Table S1) was calculated once every week from pH$_{total\ scale}$ (pH$_T$) and total alkalinity  . pH$_T$ was quantified via a Metrohm pH meter (826 pH mobile, Metrohm, Filderstadt, Germany) connected to a glass electrode (Schott IoLine, SI analytics, ± 0.001) and calibrated against TRIS–HCl (TRIS) and 2-aminopyridine-HCl (AMP; Mare, Liège, Belgium) seawater buffers (Dickson et al., 2007). Total alkalinity was measured spectrophotometrically (wavelength = 595 nm; UV-1800 Shimadzu, Japan) through base neutralization by formic acid and a pH sensitive dye (bromophenol blue), following Sarazin et al. (1999). Total dissolved

inorganic carbon ($C_T$), $pCO_2$ and aragonite saturation were calculated using CO2SYS software (Lewis and Wallace, 1998), with dissociation constants from Mehrbach et al. (1973) as refitted by Dickson and Millero (1987). The non-contaminated and contaminated fish were fed similar diets, differing only on MeHg content. Contaminated diet was fortified with MeHg (inserted in the form of MeHg(II) chloride, CH$_3$ClHg, 99.8 %, Sigma-Aldrich, solubilized previously in ethanol). Given our dietary option, ecologically relevant MeHg concentrations wasere chosen based on levels (low contamination, ~0.12 mg kg$^{-1}$ wet weight (ww); and high contamination, ~1.6 mg kg$^{-1}$ ww found in common *A. regius* prey species from contaminated coastal areas (Cardoso et al., 2014; Kannan et al., 1998; Nunes et al., 2008).). The pellets given to fish allocated to non-contaminated and contaminated treatments had approximately 0.60 ± 0.01 mg kg$^{-1}$ dry weight (dw) and 8.02 ± 0.01 mg kg$^{-1}$ dry weight (dw) of MeHg, respectively, The pellet given to the fish allocated toin the contaminated treatment had approximately 8.02 ± 0.01 mg kg$^{-1}$ dw of MeHg and 8.28 ± 0.01 mg kg$^{-1}$ dw of HgTwhich were considered to mimickthe concentrations found in the field (see Maulvault et al., 2016, 2017). Feed composition, manufacturing and MeHg spiking processeswere executed as described by (Maulvault et al.,(2016). Given our dietary option, MeHg concentration was chosen based on levels foundin common *A. regius* prey species from contaminated coastal areas(Cardoso et al., 2014; Nunes et al., 2008). An ecologically relevant concentration was chosen, indicated by previous studies on contaminated coastal areas (Nunes et al., 2008). MeHg exposure occurred via feed intake. Fish were fed two to three times a day and total feeood amountquantity provided per day was approximately 1% (standard calculation for aquaculture) of animal weight. (at the end of 30 days,each fish was given approximately 0.0106 mg of HgT). Selected fFeed quantity was calculated toalso minimized food remains, which, in case of existing, were siphoned together with fish faeces one hour after feeding.

[revised manuscript text omitted]
 physiology at sublethal levels, i.e. zero mortality and also no effects on Fulton condition were registered.

25     The fact that the meagre (*A. regius*) is a very resilient species and easily adapts to environmental alterations (Monfort, 2010) may explain the absence of deleterious effects at an organism level,after 30 days of exposure .

Affinity for metal accumulation varied between fish tissues with increasing Hg accumulation as follows: muscle < gills < liver. These results are supported by previous reports on mercury tissue preferential accumulation. The muscle is an organ

30    tissue generally characterized for its low metal affinity (Jezierska and Witeska, 2006) compared to the liver, where

metals accumulate at higher levels, due to its key role in metal accumulation and detoxification (Gbem et al., 2001; Wagner and Boman, 2003). Furthermore, as a result of increased blood supply, gills are organs likewise known to possess higher Hg affinity than the muscle (Jezierska and Witeska, 2006; Vergilio et al., 2012).

**4.2 Environmental influence on mercury accumulation**

Mercury accumulation in fish is known to depend on the water physicochemical properties (e.g. temperature, pH, alkalinity) (Harris and Bodaly, 1998; Ponce and Bloom, 1991; Wren et al., 1991). Indeed, we also showed a consistent increase in Hg accumulation under the warming scenario. However, when both temperature and $CO_2$ stressors were present, Hg accumulation was  decreased. Temperature increases Hg bioaccumulation in fish due to enhanced metabolism and consequent higher intake of MeHg-contaminated prey (Dijkstra et al., 2013; MacLeod and Pessah, 1973). Despite previous evidence that lowered pH (< 7.0 units) increases Hg accumulation in freshwater fish (Haines et al., 1992; Ponce and Bloom, 1991), the current findings do not reflect this pattern, arguably due to the magnitude of pH decrease (here we used pH 7.5). Instead, our results support recent studies  demonstrating that  hypercapnia dampens Hg accumulation in marine organisms (Li et al., 2017; Sampaio et al., 2016; Wang et al., 2017). There are several possible reasons which may underpin such an interaction, encompassing digestive (reduced digestive efficiency, reduced uptake through the gut membrane, reduced appetite, increased Hg depuration) and molecular (competition between Hg and $H^+$ ions for binding sites, impacts on Hg plasma transport, lower phospholipidic membrane permeability) mechanisms (Li et al., 2017). A recent study has also found that the lysosome-autophagy pathway was up-regulated by combined exposure to Hg and increased $CO_2$, enabling better animal fitness which may potentially reduceHg accumulation and toxicity (Wang et al., 2017). In addition,  Thus, taking  into account that the occurrence of both  warming and acidification  changes physiological  thresholds (Christensen et al., 2011; Harley et al., 2006; Rosa et al., 2013; Rosa and Seibel, 2008),  a  degree of metabolic depression  may also played a  role on decreasing HgT  accumulation (Dijkstra et al., 2013; Sampaio et al., 2016). From a consumer perspective, our study showed that the counteracting $CO_2$ effect (hampering warming-stimulated Hg accumulation) was consistent in the muscle, the main tissue ingested by human population. Since thisis the most relevant tissue for commercialization, such results constitute an important finding in the area of seafood safety, worthy of further research.

**4.3 Oxidative stress under a multi-stressor environment**

Exposure to MeHg contamination, ocean warming and acidification potentiated significant changes in meagre physiology. As expected, lipid peroxidation and consequent MDA build-up was higher under MeHg contamination (Berntssen et al., 2003; Vieira et al., 2009). The fact that contamination and warming per se elicited only small MDA build-up, is likely due to the fact that *A. regius* is a highly resilient estuarine species, i.e. great tolerance to environmental stressors (Monfort, 2010). Moreover, tTo cope with oxidative stress, *A. regius* displayed enhanced CAT, SOD and GST activities under contaminated and warming scenarios, which is in line with previous studies reporting an enhanced anti oxidative stress response in fish (Maulvault et al., 2017; Pimentel et al., 2015; Vieira et al., 2009). While it is worth mentioning that increased $CO_2$ played a minor role in CAT activity (non-significant, p = 0.116), regarding the other enzymes, hypercapnia as a sole stressor significantly augmented antioxidant activity. However when combined with other stressors, elevated $CO_2$ antagonized the co-ocurring stressor's effect (i.e. contamination and/or warming). Increased $CO_2$ (co-occuring with Hg contamination) may elicit the up-regulation of the lysosome-autophagy pathway, which is responsible for removing damaged proteins and organelles, effectively reducing oxidative stress (Wang et al., 2017). This potential mechanism may contribute to alleviate not only Hg induced stress, but also warming-related oxidative stress. We also argue that such this antagonistic relation can be partially explained by the dramatica $CO_2$-related increase of $H^+$ ion concentrations in the blood and cellular surroundings, counterbalanced by bicarbonate increase (acid-base compensation) to normalize pH levels stemming from increased $CO_2$ (Heuer and Grosell, 2014; Michaelidis et al., 2007)(Michaelidis et al., 2007). By itself, the presence of excessive $H^+$ ions activates free radical neutralizing defenses (Tiedke et al., 2013), which is in line with the present findings when hypercapnia was the sole stressor. Hhowever, the production of $O_2^-$ and further complementary ROS free radicals (e.g. $OH^-$) by other stressors may result in facilitated $H_2O$ and $H_2O_2$ formation, due to chemical reactions balancing equilibrium (e.g. $H^+ + OH^- \rightleftharpoons H_2O$), thus eliminating free radicals and decreasing activity of antioxidant enzymes to basal standards.

**4.4 Protein chaperone functioning under a multi-stressor environment**

Hsp70 response was tissue-dependent, showing a pattern similar to HgT tissue preferential accumulation (see first section). Higher liver expression is not unexpected given the fact that this organ plays a key role in metal accumulation and detoxification (Gbem et al., 2001; Wagner and Boman, 2003). More importantly, as observed in antioxidant stress enzymatic machinery, hypercapnia revealed the same antagonistic relationship with other stressor's effects: increased $CO_2$ down-regulated heat shock response in the livers of contaminated fish and in the muscle of fish under warming. As such, this study confirms that Hsp70 expression is closely correlated with other forms of antioxidant response, such as CAT, SOD and GST (Iwama et al., 1998; Rosa et al., 2012, 2014a). More so than for oxidative stress, the enhanced removal of damaged proteins and enzymes indirectly promoted by increased $CO_2$ (via up-regulated lysosome-autophagy) may have especially contributed to subside protein chaperone production. GMoreover, given that Hsp70 production can also be stimulated by extreme high

ionic (e.g. $H^+$) concentrations (Feder and Hofmann, 1999), we reason that the same additional mechanism by which hypercapnia potentially modulates oxidative stress can be applied for heat shock response . Enhanced $CO_2$ leads to increased $H^+$ concentration triggering physiological stress responses, while the facilitated conversion of free ions and radicals ($H^+$ and O-associated molecules) into $H_2O$ and $H_2O_2$ leads to reduced stress input by warming, contamination (and hypercapnia itself).

**5 Conclusions**

In this study, we observed that sublethal MeHg contamination is organ selective (accumulating to higher levels in the liver) and found that future abiotic conditions modulate its accumulation throughout the organism. In general, warming conditions enhanced MeHg accumulation but $CO_2$-linked impacts countered this effect. In fact, despite negative effects prompted as a sole stressor, acidification consistently elicited antagonistic responses to temperature and contamination effects on oxidative stress (including heat shock response), which may be explained by stimulated removal of damaged proteins and organelles (Wang et al., 2017). Moreover, we also argue that the mechanistic interactions found are coadjuvanted by the coinciding increase of hydrogen ($H^+$) and radical reactive oxygen species (e.g. $O_2^-$, $OH^-$), which subsequently nullify each other due to the spontaneous equilibrium of chemical reactions (e.g. $H^+ + OH^- \rightleftharpoons H_2O$).

In the future, it is important to deepen our understanding on this mechanism and evaluate if this antagonistic relationship is conservative throughout other less-resilient species (e.g. non-estuarine ones). Further knowledge on climate change and contamination impacts on fish ecophysiology (and biochemical stress-coping mechanisms) will help towards better comprehension of future fish stocks' health condition and tissue-dependent contaminant accumulation, consequently forecasting socio-ecological consequences in the oceans of tomorrow. Another pertinent knowledge gap that has been scarcely addressed is how oxidative stress and lipid peroxidation modify the nutritional value and general palatability of seafood, particularly fish. Thus, further multi-stressor studies on seafood safety and biochemical changes should be performed with the intent of helping stakeholders and regulatory authorities define future consumption recommendations and legislation.

**6 Code availability**

R code used in the analysis is available as Supplemental material.

**7 Data availability**

The full dataset is made available as Supplemental material.

**8 Competing interests**

The authors declare no conflict of interest.

**9 Author contributions**

ES, ARL, SF, AM, PP and RR designed the study. JRP, MP, TR and TFG assisted during the experiment and sampling. AL and SF quantified HgT accumulation. ARL, SF, MP and JRP quantified the enzymes. ES, TR, TFG and RR performed the statistical analysis. ES and ARL wrote the paper, for which all authors contributed with discussion and earlier drafts.

**Acknowledgments**

We thank Kenneth Storey for the helpful discussion and IPMA-Olhão for providing juvenile meagre specimens for the trials. This work was supported by Fundação para a Ciência e Tecnologia (FCT): PhD (ARL, SFRH/BD/97070/2013;JRP, SFRH/BD/111153/2015; ALM, SFRH/BD/103569/2014) and post-doctoral (TFG, SFRH/BPD/98590/2013; TR, SFRH/BPD/98590/2013) scholarships, as well as RR and AM in the framework of the IF 2013 and IF2014 programs.

[revised manuscript text omitted]